# Improving Generalization and Convergence by Enhancing Implicit Regularization

**Mingze Wang**[1,3,†]    **Jinbo Wang**[1,3]    **Haotian He**[1,3]    **Zilin Wang**[1]    **Guanhua Huang**[5,6]
**Feiyu Xiong**[3]    **Zhiyu Li**[3]    **Weinan E**[1,2,3,4]    **Lei Wu**[1,2,†]

[1]School of Mathematical Sciences, Peking University
[2]Center for Machine Learning Research, Peking University
[3]Institute for Advanced Algorithms Research (Shanghai)    [4]AI for Science Institute
[5]School of Data Science, University of Science and Technology of China    [6]ByteDance Research

{mingzewang, wangjinbo, haotianhe, wangzilin}@stu.pku.edu.cn
guanhuahuang@mail.ustc.edu.cn    {xiongfy, lizy}@iaar.ac.cn
{weinan, leiwu}@math.pku.edu.cn

## Abstract

In this work, we propose an Implicit Regularization Enhancement (IRE) framework to accelerate the discovery of flat solutions in deep learning, thereby improving generalization and convergence. Specifically, IRE decouples the dynamics of flat and sharp directions, which boosts the sharpness reduction along flat directions while maintaining the training stability in sharp directions. We show that IRE can be practically incorporated with *generic base optimizers* without introducing significant computational overload. Experiments show that IRE consistently improves the generalization performance for image classification tasks across a variety of benchmark datasets (CIFAR-10/100, ImageNet) and models (ResNets and ViTs). Surprisingly, IRE also achieves a $2\times$ *speed-up* compared to AdamW in the pre-training of Llama models (of sizes ranging from 60M to 229M) on datasets including Wikitext-103, Minipile, and Openwebtext. Moreover, we provide theoretical guarantees, showing that IRE can substantially accelerate the convergence towards flat minima in sharpness-aware minimization (SAM).

## 1  Introduction

Deep learning has achieved remarkable success across a variety of fields, including computer vision, scientific computing, and artificial intelligence. The core challenge in deep learning lies in how to train deep neural networks (DNNs) efficiently to achieve superior performance. Understanding and improving the generalization and convergence of commonly-used optimizers, such stochastic gradient descent (SGD) (Robbins and Monro, 1951; Rumelhart et al., 1986), in deep learning is crucial for both theoretical research and practical applications.

Notably, optimizers often exhibit a preference for certain solutions in training DNNs. For instance, SGD and its variants consistently converge to solutions that generalize well, even when DNNs are highly over-parameterized and there are many solutions that generalize poorly. This phenomenon is referred to as *implicit regularization* in the literature (Neyshabur et al., 2014; Zhang et al., 2017).

The most popular explanation for implicit regularization is that SGD and its variants tend to converge to flat minima (Keskar et al., 2016; Wu et al., 2017), and flat minima generalize better (Hochreiter and

---

† Correspondence to: Mingze Wang and Lei Wu.

38th Conference on Neural Information Processing Systems (NeurIPS 2024).

Schmidhuber, 1997; Jiang et al., 2019). However, the process of this *implicit sharpness regularization* occurs at a very slow pace, as demonstrated in works such as Blanc et al. (2020), Li et al. (2022), and Ma et al. (2022). Consequently, practitioners often use a large learning rate (LR) and extend the training time even when the loss no longer decreases, ensuring the convergence to flatter minima (He et al., 2016; Goyal et al., 2017; Hoffer et al., 2017). Nevertheless, the largest allowable LR is constrained by the need to maintain training stability. In addition, Foret et al. (2021) proposed SAM, which aims to explicitly regularize sharpness during training and has achieved superior performance across a variety of tasks.

**Our contributions** can be summarized as follows:

- We propose an Implicit Regularization Enhancement (IRE) framework to speed up the convergence towards flatter minima. As suggested by works like Blanc et al. (2020), Li et al. (2022) and Ma et al. (2022), the implicit sharpness reduction often occurs at a very slow pace, along flat directions. Inspired by this picture, IRE particularly accelerates the dynamics along flat directions, while keeping sharp directions' dynamics unchanged. As such, IRE can boost the implicit sharpness reduction substantially without hurting training stability. For a detailed illustration of this mechanism, we refer to Section 2.

- We then provide a practical IRE framework, which can be efficiently incorporated with generic base optimizers. We evaluate the performance of this practical IRE in both vision and language tasks. For vision tasks, IRE consistently improves the generalization performance of popular optimizers like SGD, Adam, and SAM in classifying the CIFAR-10/100 and ImageNet datasets with ResNets (He et al., 2016) and vision transformers (ViTs) (Dosovitskiy et al., 2020). For language modelling, we consider the pre-training of Llama models (Touvron et al., 2023) of various sizes, finding that IRE surprisingly can accelerate the pre-training convergence. Specifically, we observe a remarkable $2.0\times$ speedup compared to AdamW in the scenarios we examined, despite IRE being primarily motivated to speed up the convergence to flat solutions.

- Lastly, we provide theoretical guarantees showing that IRE can achieves a $\Theta(1/\rho)$-time acceleration over the base SAM algorithm in minimizing the trace of Hessian, where $\rho \in (0, 1)$ is a small hyperparameter in SAM.

## 1.1 Related works

**Implicit sharpness regularization.** There have been extensive attempts to explain the mystery of implicit regularization in deep learning (see the survey by Vardi (2023) and references therein). Here, we focus on works related to implicit sharpness regularization. Wu et al. (2018; 2022) and Ma and Ying (2021) provided an explanation of implicit sharpness regularization from a dynamical stability perspective. Moreover, in-depth analysis of SGD dynamics near global minima shows that the SGD noise (Blanc et al., 2020; Li et al., 2022; Ma et al., 2022; Damian et al., 2021) and the edge of stability (EoS)-driven (Wu et al., 2018; Cohen et al., 2021) oscillations (Even et al., 2024) can drive SGD/GD towards flatter minima. Additional studies explored how training components, including learning rate and batch size (Jastrzębski et al., 2017), normalization (Lyu et al., 2022), cyclic LR (Wang and Wu, 2023), influence this sharpness regularization. Furthermore, some works have provided theoretical evidence explaining the superior generalization of flat minima for neural networks (Ma and Ying, 2021; Mulayoff et al., 2021; Wu and Su, 2023; Gatmiry et al., 2023; Wen et al., 2023b). Our work is inspired by this line of research, aiming to boost implicit sharpness regularization by decoupling the dynamics along flat and sharp directions.

**Sharpness-aware minimization.** IRE shares the same motivation as SAM in enhancing sharpness regularization, although their specific approaches differ significantly. It is worth noting that the per-step computational cost of SAM is twice that of base optimizers. Consequently, there have been numerous attempts to reduce the computational cost of SAM (Kwon et al., 2021; Liu et al., 2022; Du et al., 2021; Mi et al., 2022; Mueller et al., 2024). In contrast, the per-step computational cost of IRE is only approximately 1.1 times that of base optimizers (see Table 5). Moreover, we provide both theoretical and experimental evidence demonstrating that the mechanism of IRE in boosting sharpness regularization is nearly orthogonal to that of SAM.

**Optimizers for large language model (LLM) pre-training.** (Momentum) SGD (Sutskever et al., 2013; Nesterov, 1983) and its adaptive variants like Adagrad (Duchi et al., 2011), RMSProp (Tieleman, 2012), and Adam (Kingma and Ba, 2014) have been widely used in DNN training. Despite the efforts

in designing better adaptive gradient methods (Liu et al., 2019a; Luo et al., 2019; Heo et al., 2020; Zhuang et al., 2020; Xie et al., 2022b;a), AdamW(Adam+decoupled weight decay) (Loshchilov and Hutter, 2017) has become the default optimizer in LLM pre-training. Recently, Chen et al. (2024) discovered Lion by searching the space of adaptive first-order optimizers; Liu et al. (2024) introduced Sophia, a scalable second-order optimizer. In this paper, we instead empirically demonstrate that IRE can accelerate the convergence of AdamW in the pre-training of Llama models.

## 1.2 Notations

Throughout this paper, let $\mathcal{L} : \mathbb{R}^p \mapsto \mathbb{R}_{\geqslant 0}$ be the function of total loss, where $p$ denotes the number of model parameters. For a $\mathcal{C}^2$-submanifold $\mathcal{M}$ in $\mathbb{R}^p$, we denote the tangent space of $\mathcal{M}$ at $\boldsymbol{\theta} \in \mathcal{M}$ as $\mathcal{T}_{\boldsymbol{\theta}}\mathcal{M}$, which is a linear subspace in $\mathbb{R}^p$. For $f \in \mathcal{C}^1(\mathcal{M})$ and $\boldsymbol{\theta} \in \mathcal{M}$, let $\nabla_{\mathcal{M}} f(\boldsymbol{\theta}) := \mathcal{Q}_{\mathcal{T}_{\boldsymbol{\theta}}\mathcal{M}} \nabla f(\boldsymbol{\theta})$ denote the Riemannian gradient, where $\mathcal{Q}_{\mathcal{T}_{\boldsymbol{\theta}}\mathcal{M}} : \mathbb{R}^p \mapsto \mathbb{R}^p$ denotes the orthogonal projection to $\mathcal{T}_{\boldsymbol{\theta}}\mathcal{M}$. For a symmetric matrix $A \in \mathbb{R}^{p \times p}$, its eigen pairs are denoted as $\{(\lambda_i, \boldsymbol{u}_i)\}_{i \in [p]}$ with the order $\lambda_1 \geqslant \cdots \geqslant \lambda_p$. We use $P_{i:j}(A) = \sum_{k=i}^{j} \boldsymbol{u}_k \boldsymbol{u}_k^\top$ to denote the projection operator onto $\text{span}\{\boldsymbol{u}_i, \ldots, \boldsymbol{u}_j\}$. Denote $\mathcal{N}(\boldsymbol{\mu}, \Sigma)$ as the Gaussian distribution with mean $\boldsymbol{\mu}$ and covariance matrix $\Sigma$, and $\mathbb{U}(\mathcal{S})$ as the uniform distribution over a set $\mathcal{S}$. Given a vector $\boldsymbol{h} = (h_1, \ldots, h_p)$, let $|\boldsymbol{h}| = (|h_1|, \ldots, |h_p|)$. We denote by $\mathbf{1}$ the all-ones vector. We will use standard big-O notations like $\mathcal{O}(\cdot), \Omega(\cdot)$, and $\Theta(\cdot)$ to hide constants.

## 2 An Illustrative Example Motivating IRE

In this section, we provide an illustration of how the dynamics along flat directions can *reduce the sharpness* (curvatures along sharp directions) and how IRE can accelerate this sharpness reduction. To this end, we consider the following phenomenological problem:

$$\min_{\boldsymbol{\theta} \in \mathbb{R}^p} \mathcal{L}(\boldsymbol{\theta}) := \boldsymbol{v}^\top H(\boldsymbol{u}) \boldsymbol{v} / 2, \tag{1}$$

where $\boldsymbol{v} \in \mathbb{R}^m$, $\boldsymbol{u} \in \mathbb{R}^{p-m}$, and $\boldsymbol{\theta} = \text{vec}(\boldsymbol{u}, \boldsymbol{v}) \in \mathbb{R}^p$. We assume $H(\cdot) \in \mathcal{C}^2(\mathbb{R}^{p-m})$ and $\inf_{\boldsymbol{u}} \lambda_{\min}(H(\boldsymbol{u})) > 0$. Then, the minimizers of $\mathcal{L}(\cdot)$ form a $m$-dim manifold $\mathcal{M} = \{(\boldsymbol{u}, \boldsymbol{v}) : \boldsymbol{v} = \mathbf{0}\}$ and the Hessian at any $\boldsymbol{\theta} \in \mathcal{M}$ is given by $\nabla^2 \mathcal{L}(\boldsymbol{\theta}) = \begin{pmatrix} \mathbf{0} & \mathbf{0} \\ \mathbf{0} & H(\boldsymbol{u}) \end{pmatrix}$. For clarity, we shall call $\boldsymbol{u}$ and $\boldsymbol{v}$ the flat and sharp directions, respectively.

**Example 2.1.** *The loss landscape of fitting zero labels with two-layer neural networks (2LNNs) exhibits exactly the form* (1). *Let $f(\boldsymbol{x}; \boldsymbol{\theta}) = \boldsymbol{a}^\top \phi(\boldsymbol{x}; W)$ be a 2LNN with $\boldsymbol{\theta} = \text{vec}(W, \boldsymbol{a})$. Then $\mathcal{L}(\boldsymbol{\theta}) = \mathbb{E}_{(\boldsymbol{x}, y)}[(f(\boldsymbol{x}; \boldsymbol{\theta}) - y)^2]/2 = \boldsymbol{a}^\top \mathbb{E}_{\boldsymbol{x}}[\phi(\boldsymbol{x}; W)\phi(\boldsymbol{x}; W)^\top]\boldsymbol{a}/2 =: \boldsymbol{a}^\top H(W)\boldsymbol{a}/2.$*

For breviety, we further assume $H(\boldsymbol{u}) = \text{diag}(\boldsymbol{\lambda}(\boldsymbol{u}))$ with $\boldsymbol{\lambda}(\boldsymbol{u}) = (\lambda_1(\boldsymbol{u}), \cdots, \lambda_m(\boldsymbol{u}))$. In this case, the GD dynamics can be naturally decomposed into the flat and sharp directions as follows

$$\begin{aligned} \boldsymbol{u}_{t+1} &= \boldsymbol{u}_t - \frac{\eta}{2} \sum_{i=1}^{m} v_{t,i}^2 \nabla \lambda_i(\boldsymbol{u}_t), \\ \boldsymbol{v}_{t+1} &= (\mathbf{1} - \eta\boldsymbol{\lambda}(\boldsymbol{u}_t)) \odot \boldsymbol{v}_t, \end{aligned} \tag{2}$$

where $\odot$ denotes the element-wise multiplication of two vectors.

**The implicit sharpness regularization.** From Eq. (2), we can see that 1) the flat direction $\boldsymbol{u}_t$'s dynamics monotonically reduces the sharpness $\boldsymbol{\lambda}(\boldsymbol{u})$ as long as $\boldsymbol{v}_t$ is nonzero; 2) the sharp direction $\boldsymbol{v}_t$'s dynamics determines the speed of sharpness reduction. The larger $|\boldsymbol{v}_t|$ is, the faster the curvature $\boldsymbol{\lambda}(\boldsymbol{u})$ decreases. Particularly, when near convergence, we have $|\boldsymbol{v}_t| = o(1)$ and thus the implicit sharpness reduction is *very slow* during the late phase of GD. Figure 1a provides a visualization of this slow implicit sharpness reduction.

**Accelerating the sharpness reduction.** Inspired by the above analysis, we can accelerate the sharpness reduction by speeding up the flat directions' dynamics. To this end, there are two approaches:

- **Naively increasing the global learning rate $\eta$ (fail).** Increasing $\eta$ accelerates the dynamics of $\boldsymbol{u}_t$, but the largest allowed $\eta$ is constrained by curvatures of sharpest directions. In GD (2), to

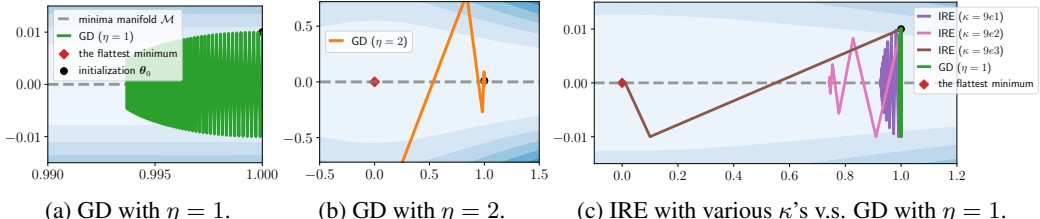

(a) GD with $\eta = 1$.  (b) GD with $\eta = 2$.  (c) IRE with various $\kappa$'s v.s. GD with $\eta = 1$.

Figure 1: A 2-d example of (1): $\mathcal{L}(u, v) = (1 + u^2)v^2/2$. The gray arrows denote to the minima manifold $\mathcal{M} = \{(u, v) : v = 0\}$, where the smaller the $|u|$, the flatter the minimizer. The red marker highlights the flattest minimizer $(0, 0)$. (a) The dynamics of GD $(\eta = 1)$, which moves *slowly* towards flatter minima as it converges. (b) The dynamics of GD $(\eta = 2)$, which diverges due to the excessively large $\eta$. (c) The behavior of our IRE approach with varying $\kappa$'s v.s. GD $(\eta = 1)$. Is is shown that IRE can significantly accelerate the $u_t$'s dynamics, almost reaching the flattest minimum $(0, 0)$ when taking a very large $\kappa$.

maintain training stability, $\eta$ must be smaller than $2/\max_i \lambda_i(\boldsymbol{u}_t)$. Otherwise, $\boldsymbol{v}_t$'s dynamics will blow up. As illustrated in Figure 1b, setting $\eta = 2$ leads to divergence, whereas $\eta = 1$ ensures convergence.

- **Increasing only the flat directions' learning rate (our approach, IRE)**. Specifically, for GD (2), the GD-IRE dynamics is given by

$$\boldsymbol{u}_{t+1} = \boldsymbol{u}_t - (1 + \kappa)\frac{\eta}{2}\sum_{i=1}^{m} v_{t,i}^2 \nabla \lambda_i(\boldsymbol{u}_t),$$

$$\boldsymbol{v}_{t+1} = (\boldsymbol{1} - \eta\boldsymbol{\lambda}(\boldsymbol{u}_t)) \odot \boldsymbol{v}_t,$$

(3)

where $\kappa > 0$ controls the enhancement strength. In GD-IRE (3), $\boldsymbol{u}_t$'s dynamics is $(1 + \kappa)$ **faster** than that of GD (2). Notably, the sharp directions' dynamics $(\boldsymbol{v}_t)$ are unchanged. The choice of $\kappa$ only needs to maintain the stability of flat directions' dynamics, for which, we can always take a significantly large $\kappa$ to enhance the sharpness regularization. As demonstrated in Figure 1c, IRE with larger $\kappa$ always find flatter minima.

**Remark 2.2** (The generality). It is worth noting that similar implicit sharpness regularization also holds for SGD (Ma et al., 2022; Li et al., 2022) and SAM (Wen et al., 2023a). In this section, we focus on the above toy model and GD mainly for illustration. In Appendix B, we provide an analogous illustrative analysis of how IRE accelerates the sharpness reduction of SGD. In Section 5, we further provide theoretical evidence to show that IRE can boost the implicit sharpness regularization of SAM.

## 3 A Practical Framework of Implementing IRE

Although the preceding illustration of IRE is for GD, in practice, we can incorporate IRE with *any base optimizers*. Specifically, for a generic update: $\boldsymbol{\theta}_{t+1} = \boldsymbol{\theta}_t - \eta\boldsymbol{g}_t$, the corresponding IRE modification is given by

$$\boldsymbol{\theta}_{t+1} = \boldsymbol{\theta}_t - \eta(\boldsymbol{g}_t + \kappa\mathcal{P}_t\boldsymbol{g}_t),$$

(4)

where $\kappa$ denotes the enhancement strength and $\mathcal{P}_t : \mathbb{R}^p \to \mathbb{R}^p$ projects $\boldsymbol{g}_t$ into the *flat directions* of the landscape. The flat directions and corresponding projection operator $\mathcal{P}_t$ can be estimated using the Hessian information.

However, estimating the full Hessian matrix $\nabla^2\mathcal{L}(\boldsymbol{\theta}_t) \in \mathbb{R}^{p \times p}$ is computationally infeasible. Consequently, we propose to use only the *diagonal Hessian* $\mathrm{diag}(\nabla^2\mathcal{L}(\boldsymbol{\theta}_t)) \in \mathbb{R}^p$ to estimate $\mathcal{P}_t$. Let $\boldsymbol{h}_t \in \mathbb{R}^p$ be an estimate of the diagonal Hessian. Then, we perform the projection as follows

$$\mathcal{P}_t\boldsymbol{g}_t = \boldsymbol{n}_t \odot \boldsymbol{g}_t, \text{ with } (\boldsymbol{n}_t)_i = \begin{cases} 1 & \text{if } (|\boldsymbol{h}_t|)_i \leqslant \mathrm{Top}_{\mathrm{small}}(|\boldsymbol{h}_t|, \gamma) \\ 0 & \text{otherwise} \end{cases},$$

(5)

where $\gamma \in (0, 1)$ and $\mathrm{Top}_{\mathrm{small}}(|\boldsymbol{h}_t|, \gamma)$ returns the $\lfloor p \cdot \gamma \rfloor$-th smallest value in $|\boldsymbol{h}_t|$. Note that $\boldsymbol{n}_t \in \mathbb{R}^p$ denotes a mask vector and the above approximate projection essentially masks the top-$(1 - \gamma)$ sharp coordinates out. As such, the projection (5) will retain the *top-$\gamma$ flat coordinates*. Noticing that in DNNs, there are much more flat directions than sharp directions (Yao et al., 2020), we thus often use $\gamma > 0.5$ in practice.

**Algorithm 1: Practical IRE** (A practical framework of implementing IRE)

---

**Input:** $\boldsymbol{\theta}_0$, $T$, $K$, learning rate $\{\eta_t\}_t$, warm-up time $T_{\mathrm{w}}$, IRE hyperparams: $\kappa \geqslant 0$ and $\gamma \in (0.5, 1)$;

**for** $t = T_{\mathrm{w}}, \cdots, T-1$ **do**

    Compute the original update direction $\boldsymbol{g}_t$ according to the base optimizer;

    **if** $(t - T_{\mathrm{w}}) \bmod K = 0$ **then**

        Estimate the diagonal Hessian $\boldsymbol{h}_t \in \mathbb{R}^p$ using Eq. (6);

        Update the mask $\boldsymbol{n}_t \in \mathbb{R}^p$ using Eq. (5);

    **else**

        $\boldsymbol{n}_t = \boldsymbol{n}_{t-1}$

    $\boldsymbol{\theta}_{t+1} = \boldsymbol{\theta}_t - \eta_t \big(\boldsymbol{g}_t + \kappa \boldsymbol{n}_t \odot \boldsymbol{g}_t \big)$;

**Output:** $\boldsymbol{\theta}_T$.

---

**A light-weight estimator of the diagonal Hessian.** Let $\ell(\cdot, \cdot)$ be the cross-entropy loss. Given an input data $\boldsymbol{x} \in \mathbb{R}^{d_x}$ and label $\boldsymbol{y} \in \mathbb{R}^{d_y}$, let the model's prediction be $f(\boldsymbol{x}; \boldsymbol{\theta}) \in \mathbb{R}^{d_y}$. The Fisher (Gauss-Newton) matrix $F(\boldsymbol{\theta})$ is widely acknowledged to be a good approximation of the Hessian, particularly near minima. Thus, we can estimate the diagonal Hessian by $\boldsymbol{h}_t = \mathrm{diag}(F(\boldsymbol{\theta}_t))$, which has been widely used in deep learning optimization (Martens and Grosse, 2015; Grosse and Martens, 2016; George et al., 2018; Mi et al., 2022; Liu et al., 2024). Given an input batch $\{(\boldsymbol{x}_b, \boldsymbol{y}_b)\}_{b=1}^B$, the empirical diagonal Fisher is given by $\mathrm{diag}(\hat{F}(\boldsymbol{\theta})) = \frac{1}{B} \sum_{b=1}^{B} \nabla \ell(f(\boldsymbol{x}_b; \boldsymbol{\theta}); \hat{\boldsymbol{y}}_b) \odot \nabla \ell(f(\boldsymbol{x}_b; \boldsymbol{\theta}); \hat{\boldsymbol{y}}_b)$, where $\hat{\boldsymbol{y}}_b \sim \mathrm{softmax}(f(\boldsymbol{\theta}; \boldsymbol{x}_b))$. However, as noted by Liu et al. (2024), implementing this estimator is computationally expensive due to the need to calculate $B$ single-batch gradients. Liu et al. (2024) proposed a more convenient estimator $\mathrm{diag}(\hat{F}_{\mathrm{eff}}(\boldsymbol{\theta}))$, only requires computing the mini-batch gradient $\nabla \hat{\mathcal{L}}_B(\boldsymbol{\theta}) = \frac{1}{B} \sum_{b=1}^{B} \nabla \ell(f(\boldsymbol{x}_b; \boldsymbol{\theta}); \hat{\boldsymbol{y}}_b)$ with $\hat{\boldsymbol{y}}_b \sim \mathrm{softmax}(f(\boldsymbol{x}_b; \boldsymbol{\theta}))$:

$$\boldsymbol{h}_t = \mathrm{diag}(\hat{F}_{\mathrm{eff}}(\boldsymbol{\theta})) = B \cdot \nabla \hat{\mathcal{L}}_B(\boldsymbol{\theta}) \odot \nabla \hat{\mathcal{L}}_B(\boldsymbol{\theta}). \tag{6}$$

According to Liu et al. (2024), this estimator is an unbiased estimate of the empirical diagonal Fisher, i.e., $\mathbb{E}_{\hat{\boldsymbol{y}}}[\mathrm{diag}(\hat{F}_{\mathrm{eff}}(\boldsymbol{\theta}))] = \mathbb{E}_{\hat{\boldsymbol{y}}}[\mathrm{diag}(\hat{F}(\boldsymbol{\theta}))]$. For more discussions on the efficiency of this estimator, please refer to (Liu et al., 2024, Section 2). Additionally, for squared loss, one can simply use Fisher as the estimator (Liu et al., 2024).

**The practical IRE and computational efficiency.** The practical IRE is summarized in Algorithm 1, which is notably lightweight. The estimation of $\boldsymbol{h}_t$ using (6) requires computational resources roughly equivalent to one back-propagation. Consequently, by setting $K = 10$ in Algorithm 1 (estimating the projection every 10 steps), the average per-step computational load of IRE is *only* 1.1 *times* that of the base optimizer. This claim can be empirically validated as shown in Table 5.

## 4 Experiments

In this section, we evaluate how IRE performs when incorporating with various base optimizers. Specifically, we examine the incorporation of IRE with SGD (SGD-IRE), SAM (SAM-IRE), and AdamW (AdmIRE) across vision and language tasks.

### 4.1 Image classification

#### 4.1.1 Validating our motivation

To show that IRE can accelerate the sharpness reduction, we train WideResNet-16-8 (Zagoruyko and Komodakis, 2016) on CIFAR-10 dataset (Krizhevsky and Hinton, 2009) by SAM-IRE (with $K = 10$, varying $\kappa$ and $\gamma$). Here, we incoporate IRE into SAM starting from the 30-th epochs. We vary $\gamma \in \{0.8, 0.9, 0.95\}$ and $\kappa \in \{0, 2, 5, 10\}$. Regarding the learning rate (LR), both constant LR and decayed LR are considered. The sharpness is measured by $\mathrm{Tr}(\nabla^2 \mathcal{L}(\boldsymbol{\theta}))$. Further experimental details can be found in Appendix C.

As depicted in Fig. 2(a), SAM-IRE (with constant LR) consistently finds flatter solutions compared to SAM and higher $\kappa$ always leads to flatter minima. Additionally, SAM-IRE also shows robustness to

variations of $\gamma$. For SAM-IRE with decayed LR (Fig. 2(b)), SAM-IRE still consistently finds flatter solutions than SAM. Notably, flatter solutions correlate positively with lower training loss and higher test accuracy (Fig. 2(c,d)).

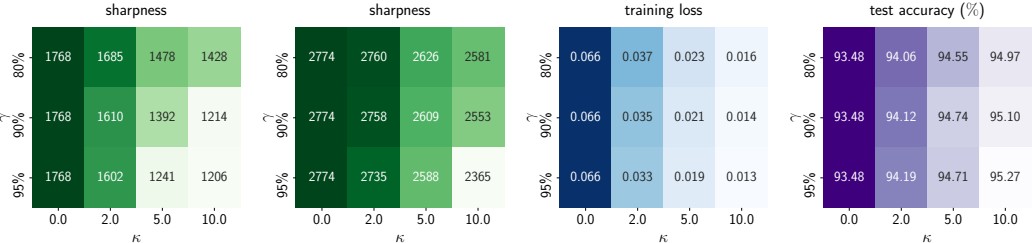

(a) sharpness, constant LR. (b) sharpness, decayed LR. (c) train loss, decayed LR. (d) test acc, decayed LR.

Figure 2: Training WRN-16-8 on CIFAR-10 by SAM-IRE with varying $\gamma, \kappa$. Particularly, the case of $\kappa = 0$ correspond to the standard SAM.

### 4.1.2 IRE can consistently improve generalization

**Convolutional Neural Networks (CNNs).** In this experiment, we first consider the classification of CIFAR-{10,100} with WideResNet-28-10 (Zagoruyko and Komodakis, 2016) and ResNet-56 (He et al., 2016). Both SGD and SAM optimizers are adopted. All the experiments use base data augmentation and label smoothing. For SGD-IRE/SAM-IRE, we fix $K = 10$, and tune hyperparameters $\gamma$ and $\kappa$ via a grid search over $\gamma \in \{0.99, 0.9, 0.8\}$ and $\kappa \in \{1, 2\}$. The total epochs are set to 100 for CIFAR-10 and 200 for CIFAR-100, and we switch from SGD/SAM to SGD-IRE/SAM-IRE when the training loss approaches 0.1. The other experimental details are deferred to Appendix C and the results are shown in Table 1.

Secondly, we evaluate IRE for training ResNet-50 on ImageNet (Deng et al., 2009). The experimental details are deferred to Appendix C and the results are shown in Table 2.

**Vision Transformers (ViTs).** We also examine the impact of IRE on generalization of ViT-T and ViT-S (Dosovitskiy et al., 2020) on CIFAR100. The default optimizers used are AdamW and SAM (Mueller et al., 2024). Strong data augmentations (basic + AutoAugment) are utilized. The total epochs are set to 200 and we switch from AdamW/SAM to AdmIRE/SAM-IRE when the training loss approaches 0.5. For AdmIRE/SAM-IRE, we fix $K = 10$, and tune hyperparameters $\gamma$ and $\kappa$ via a grid search over $\gamma \in \{0.99, 0.9, 0.8\}$ and $\kappa \in \{20, 50\}$. Other experimental details are deferred to Appendix C. The results are shown in Table 3.

Additionally, we evaluate IRE for training ViT-S on ImageNet. The experimental details are deferred to Appendix C and the results are shown in Table 4.

As demonstrated in Table 1, 2, 3, and 4, IRE *consistently improves* generalization of SGD, AdamW and SAM across all settings examined.

Table 1: WRN-28-10/ResNet-56 on CIFAR-10/100.

|  | WRN-28-10 | | ResNet-56 | |
| --- | --- | --- | --- | --- |
|  | CIFAR-10 | CIFAR-100 | CIFAR-10 | CIFAR-100 |
| SGD | 95.84 | 80.81 | 93.49 | 72.81 |
| SGD-IRE | **96.24 (+0.40)** | **81.49 (+0.68)** | **93.78 (+0.29)** | **73.78 (+0.97)** |
| SAM | 96.58 | 83.05 | 94.05 | 75.54 |
| SAM-IRE | **96.70 (+0.12)** | **83.50 (+0.45)** | **94.46 (+0.41)** | **75.86 (+0.32)** |

Table 2: ResNet-50 on ImageNet.

|  | Top-1 | Top-5 |
| --- | --- | --- |
| SGD | 76.81 | 93.31 |
| SGD-IRE | **77.04 (+0.23)** | **93.58 (+0.27)** |
| SAM | 77.47 | 93.90 |
| SAM-IRE | **77.92 (+0.45)** | **94.12 (+0.22)** |

Table 3: ViT-T/S on CIFAR-100.

|  | ViT-T | ViT-S |
| --- | --- | --- |
| AdamW | 63.90 | 65.43 |
| AdmIRE | **67.05 (+3.15)** | **68.39 (+2.96)** |
| SAM | 64.25 | 66.93 |
| SAM-IRE | **67.33 (+3.08)** | **70.47 (+3.54)** |

Table 4: ViT-S on ImageNet.

|  | Top-1 | Top-5 |
|---|---|---|
| AdamW | 78.7 | 94.0 |
| AdmIRE ($\kappa = 2, \gamma = 0.6$) | **79.0 (+0.3)** | **94.3 (+0.3)** |
| AdmIRE ($\kappa = 2, \gamma = 0.8$) | **79.1 (+0.4)** | **94.2 (+0.2)** |

## 4.2 Large language model pre-training

We now evaluate IRE in the pre-training of decoder-only large language models (LLMs). Following the training protocol of Llama models, we employ the AdamW optimizer with hyperparameters $\beta_1 = 0.9, \beta_2 = 0.95$ and weight decay $\lambda = 0.1$ (Touvron et al., 2023). The learning rate strategy includes a warm-up phase followed by a cosine decay scheduler, capped at `lr_max`. In each experiment, we tune `lr_max` only for AdamW and use it also for AdmIRE, for which the IRE is activated at the end of warm-up phase.

### 4.2.1 Computational efficiency and hyperparameter robustness

The first experiment is conducted to verify both the computational efficiency and the robustness of hyperparameters $(\gamma, \kappa)$ in IRE for pre-training tasks. Specifically, we train a 2-layer decoder-only Transformer (8M) on the Wikitext-2 dataset (4.3M) (Merity et al., 2016) by AdamW and AdmIRE (with $K = 10$ and varying $\gamma, \kappa$). The total training duration is 100k steps, including a 3k-step warm-up phase.

First, we tune `lr_max` in AdamW, identifying the optimal `lr_max=6e-4`. Subsequently, we train both AdamW and AdmIRE using this `lr_max`.

Table 5: Wall-clock time on 1 A800.

| Algorithm | time (/step) |
|---|---|
| AdamW | 0.165s |
| AdmIRE | 0.185s |

**Computational efficiency.** As shown in Table 5, AdmIRE with $K = 10$ (estimating the projection mask every 10 steps) is computationally efficient: the average time per step of AdmIRE is only 1.12 times that of AdamW, corresponding to the theoretical estimation (1.1 times).

**Robustness to hyperparameters.** Figure 3 shows that AdmIRE, with varying $\gamma$ and $\kappa$, consistently speeds up the pre-training. Remarkably, with the best configuration, AdmIRE can achieves **5.4× speedup** than well-tuned AdamW.

More experimental details and results are deferred to Appendix C.

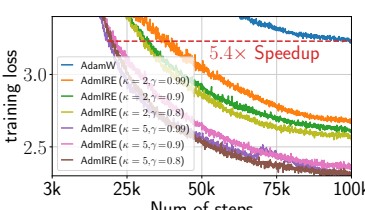

Figure 3: Transformer on wikitext-2.

### 4.2.2 Experiments on Llama models

Llama (Touvron et al., 2023), a popular open LLM, exhibits remarkable capabilities across general domains. In this section, we examine the performance of AdmIRE in training Llama models of various sizes across various datasets:

- **Llama (60M) on wikitext-103 (0.5G).** Wikitext-103 (Merity et al., 2016) serves as a standard language modeling benchmark for pre-training, which contains 103M training tokens from 28K articles, with an average length of 3.6K tokens per article.

- **Llama (119M) on minipile (6G).** Minipile (Kaddour, 2023), a 6GB subset of the deduplicated Pile (825GB) (Gao et al., 2020) presents a highly diverse text corpus. Given its diversity, training on minipile poses more challenges and potential instabilities for optimizers compared to Wikitext-103.

- **Llama (229M) on openwebtext (38G).** Openwebtext (Gokaslan and Cohen, 2019), an open-source recreation of the WebText corpus, is extensively utilized for LLM pre-training such as RoBERTa (Liu et al., 2019b) and GPT-2 (Radford et al., 2019).

Additionally, gradient clipping is adopted to maintain the training stability (Pascanu et al., 2012). First, we tune `lr_max` in AdamW for each of the three experiments, separately. The optimal `lr_max`

identified for these three experiments is all `6e-4`. Then, both AdamW and AdmIRE are trained using this optimal `lr_max`. For more details, please refer to Appendix C.

**AdmIRE is $2\times$ faster than AdamW.** The results are reported in Figure 4. We can see that AdmIRE consistently achieves a $2.1\times$ speedup compared with well-tuned AdamW for all three cases.

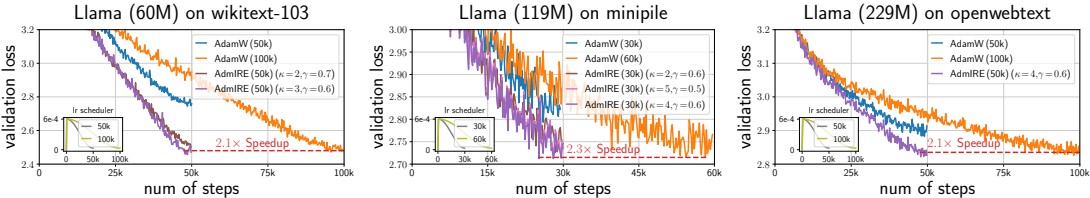

Figure 4: AdmIRE outperforms AdamW in the pre-training of Llama models.

Notice that the primary motivation behind IRE is to speed up the sharpness reduction, which only requires to increase learning rate along completely flat (zero-curvature) directions. However, practical implementation may also increase the learning rate along directions with small but non-zero curvatures, which can further speed up loss convergence. A thorough explanation for the significant acceleration provided by this approach is left for future research.

We further assess the sharpness reduction capability of IRE for LLM pre-training. Specifically, we compare the sharpness of solutions, $\mathrm{Tr}(\nabla^2 \mathcal{L}(\boldsymbol{\theta}))$, found by AdamW/AdmIRE during pre-training of Llama (60M) on wiki-103 dataset (corresponding to Figure 4 (left)). The results shown in Table 6 demonstrate that AdamIRE not only achieves the same loss in *only half the iterations* required by AdamW, but also the solutions found by AdmIRE are *significantly flatter* than that found by AdamW.

Table 6: Comparison of the sharpness of the solutions found by AdamW/AdmIRE.

|  | AdamW | AdmIRE |
|---|---|---|
| training steps | 100k | 50k |
| final $\mathcal{L}(\boldsymbol{\theta})$ | 2.47 | 2.47 |
| final $\mathrm{Tr}(\nabla^2 \mathcal{L}(\boldsymbol{\theta}))$ | 120.41 | 88.86 |

Recently, Liu et al. (2023) revealed a strong correlation between the sharpness and downstream task performance, suggesting that for models with the same pre-training loss, flatter solutions yield better performance on downstream tasks. Based on this, we hypothesize that the solutions found by IRE may also have better performance in downstream tasks, which we leave to future work.

## 5 Theoretical Guarantees for IRE on SAMs

Both empirical (Foret et al., 2021) and theoretical (Wen et al., 2023a) studies have validated that SAM algorithms exhibit superior sharpness regularization compared to (S)GD. In this section, we provide a theoretical analysis demonstrating that IRE can further enhance the sharpness regularization of SAM algorithms substantially.

### 5.1 Theoretical setups

Recall that $\mathcal{L}(\boldsymbol{\theta}) := \frac{1}{n} \sum_{i=1}^{n} \mathcal{L}_i(\boldsymbol{\theta})$ denote the total loss, where $\mathcal{L}_i(\boldsymbol{\theta})$ is the loss on the $i$-th data. Without loss of generality, we assume $\min_{\boldsymbol{\theta}} \mathcal{L}(\boldsymbol{\theta}) = 0$. We further make the following assumption:

**Assumption 5.1** (Manifold of minimizers)**.** Assume that $\mathcal{L} \in \mathcal{C}^4(\mathbb{R}^p)$, $\mathcal{M} := \arg\min_{\boldsymbol{\theta}} \mathcal{L}(\boldsymbol{\theta})$ is a $(p-m)$-dim $\mathcal{C}^2$-submanifold in $\mathbb{R}^p$ for some $m \in [p]$, and $\mathrm{rank}(\nabla^2 \mathcal{L}(\boldsymbol{\theta})) = m$ for any $\boldsymbol{\theta} \in \mathcal{M}$.

The above connectivity assumption on the manifold of minimizers $\mathcal{M}$ has been empirically verified in works such as Draxler et al. (2018) and Garipov et al. (2018), and theoretically supported in Cooper (2018). This assumption is also widely used in the theoretical analysis of implicit regularization (Fehrman et al., 2020; Li et al., 2022; Arora et al., 2022; Wen et al., 2023a).

Besides, we introduce the following definitions to characterize the dynamics of gradient flow (GF) near the minima manifold $\mathcal{M}$, which is also used in the related works above.

**Definition 5.2** (Limiting map of GF)**.** Consider the GF: $\frac{d\boldsymbol{\theta}(t)}{dt} = -\nabla\mathcal{L}(\boldsymbol{\theta}(t))$ starting from $\boldsymbol{\theta}(0) = \boldsymbol{\theta}$. Denote by $\Phi(\boldsymbol{\theta}) := \lim_{t\to\infty} \boldsymbol{\theta}(t)$ the limiting map of this GF.

**Definition 5.3** (Attraction set of $\mathcal{M}$)**.** Let $U$ be the attraction set of $\mathcal{M}$ under GF, i.e., GF starting in $U$ converges to some point in $\mathcal{M}$. Formally, $U := \{\boldsymbol{\theta} \in \mathbb{R}^p : \Phi(\boldsymbol{\theta}) \in \mathcal{M}\}$.

As proven in (Arora et al., 2022, Lemma B.15), Assumption 5.1 ensures that $U$ (in Definition 5.3) is open and $\Phi(\cdot)$ (in Definition 5.2) is $\mathcal{C}^2$ on $U$.

## 5.2 Theoretical results

The stochastic SAM (Foret et al., 2021) is given by

$$\text{standard SAM:} \quad \boldsymbol{\theta}_{t+1} = \boldsymbol{\theta}_t - \eta\nabla\mathcal{L}_{i_t}\left(\boldsymbol{\theta}_t + \rho\frac{\nabla\mathcal{L}_{i_t}(\boldsymbol{\theta}_t)}{\|\nabla\mathcal{L}_{i_t}(\boldsymbol{\theta}_t)\|}\right), \text{ where } i_t \sim \mathbb{U}([n]). \quad (7)$$

The generalization capability of standard SAM can be bounded by the average sharpness, $\mathcal{L}^{\text{avg}}(\boldsymbol{\theta}) := \mathbb{E}_{\boldsymbol{\xi}\sim\mathcal{N}(\mathbf{0},I)}\mathcal{L}(\boldsymbol{\theta} + \rho\boldsymbol{\xi}/\|\boldsymbol{\xi}\|)$ (Foret et al., 2021). This leads researchers to also explore average SAM (Wen et al., 2023a; Zhu et al., 2023; Ujváry et al., 2022), which minimizes $\mathcal{L}^{\text{avg}}$:

$$\text{average SAM:} \quad \boldsymbol{\theta}_{t+1} = \boldsymbol{\theta}_t - \eta\nabla\mathcal{L}(\boldsymbol{\theta}_t + \rho\boldsymbol{\xi}_t/\|\boldsymbol{\xi}_t\|), \text{ where } \boldsymbol{\xi}_t \sim \mathcal{N}(\mathbf{0},I). \quad (8)$$

**Two-phase algorithms.** Our theoretical focus is on how IRE accelerates the sharpness reduction of SAM (7) and (8) *near the minima manifold* $\mathcal{M}$. Thus, we analyze the two-phase algorithms. Specifically, let the initialization $\boldsymbol{\theta}_0 \in U$. In **Phase I** ($t \leqslant T_{\text{I}}$), we employ GF $\frac{d\boldsymbol{\theta}_t}{dt} = -\nabla\mathcal{L}(\boldsymbol{\theta}_t)$ to ensure that the loss decreases sufficiently; then in **Phase II** ($T_{\text{I}} < t \leqslant T_{\text{I}} + T_{\text{II}} := T$), we incorporate IRE into the standard / average SAM.

**Effective dynamics: sharpness regularization.** The implicit regularization of SAMs can be modeled using effective dynamics. In Phase II, $\boldsymbol{\theta}_t$ are close the manifold of minimizers $\mathcal{M}$ and let $\boldsymbol{z}_t := \Phi(\boldsymbol{\theta}_t) \in \mathcal{M}$. Then, the effective dynamics is given by $\{\boldsymbol{z}_t\}_{t=T_{\text{I}}+1}^T$, revealing how SAMs explore the manifold of minimizers $\mathcal{M}$. Particularly, Wen et al. (2023a) showed that the effective dynamics of standard/average SAM are both

$$\mathbb{E}[\boldsymbol{z}_{t+1}] = \boldsymbol{z}_t - \eta_{\text{eff}}\nabla_{\mathcal{M}}\text{Tr}\left[\nabla^2\mathcal{L}(\boldsymbol{z}_t)/2\right] + o(\eta_{\text{eff}}), \quad (9)$$

which minimizes the trace of Hessian on $\mathcal{M}$. The difference between the standard SAM (7) and average SAM (8) lies in the effective learning rate (LR) $\eta_{\text{eff}}$'s. A visual illustration of some quantities in (9) is provided in the figure above.

**Summary of our theoretical results.** In this section, we show that incorporating IRE into SAMs can significantly increase the effective LR $\eta_{\text{eff}}$ in (9) while maintaining the same training stability as SAMs. In Table 7, we present the effective LR for SAMs and the SAM-IREs. We see clearly that IRE can accelerate the sharpness reduction by a non-trivial factor for both standard and average SAM.

Table 7: Comparison of the implicit regularization strength of SAMs w/o IRE.

| Algorithm | Effective LR: $\eta_{\text{eff}}$ |
|---|---|
| average SAM (8) | $\eta^2/p$ (Thm 5.5) |
| IRE + average SAM (8) | $\eta^{1.5}/p$ (Thm 5.5) |
| standard SAM (7) | $\eta\rho^2$ (Thm 5.6; Wen et al. (2023a)) |
| IRE + standard SAM (7) | $\eta\rho$ (Thm 5.6) |

**Remark 5.4** (The mechanism of IRE's success)**.** The success of SAM-IRE follows the same mechanism illustrated in Section 2. The key fact that IRE only increases the LR along flat directions has two implications: 1) It does not change the trend of implicit regularization in Eq. (9) but accelerates SAMs' effective dynamics by a factor of $(1 + \kappa)$; 2) Since the LR is only increased along flat directions, $\kappa$ can be set substantially large without hurting the training stability, because the dynamics in sharp directions remain unchanged. Specifically, we theoretically justify in SAM-IRE, $\kappa$ can be selected as large as $1/\rho$.

### 5.2.1 IRE on average SAM: An $\Omega(1/\eta^{0.5})$ acceleration

We first consider IRE on average SAM. Let $T_{\mathrm{I}}$ be the hitting time: $T_{\mathrm{I}} := \inf\{t \geqslant 0 : \|\boldsymbol{\theta}_t - \Phi(\boldsymbol{\theta}_t)\| = \mathcal{O}(\sqrt{\eta}\rho)\}$. When running GF in Phase I, Definition 5.3 guarantees $T_{\mathrm{I}} < \infty$. Thus, at the starting of Phase II, $\|\boldsymbol{\theta}_t - \Phi(\boldsymbol{\theta}_t)\| = \mathcal{O}(\sqrt{\eta}\rho)$. Furthermore, the following result holds for Phase II.

**Theorem 5.5** (IRE on average SAM). *Suppose Assumption 5.1 holds. If $\eta = \mathcal{O}(1)$ and $\rho = \mathcal{O}(\sqrt{\eta})$ in SAM (8), $\kappa \leqslant 1/\rho$, and $\mathcal{P}_t = P_{m+1:p}(\nabla^2 \mathcal{L}(\boldsymbol{\theta}_t))$ in IRE (4), then with high probability at least $1 - T_{\mathrm{II}}^2 \exp\left(-\Omega\left(1/\left(\eta + p^{-1}\right)\right)\right)$, $\|\boldsymbol{\theta}_t - \Phi(\boldsymbol{\theta}_t)\| = \mathcal{O}(\sqrt{\eta}\rho)$ holds for all $T_{\mathrm{I}} \leqslant t \leqslant T$. Furthermore, the effective dynamics of $\boldsymbol{z}_t := \Phi(\boldsymbol{\theta}_t) \in \mathcal{M}$ satisfies:*

$$\mathbb{E}_{\boldsymbol{\xi}_t}[\boldsymbol{z}_{t+1}] = \boldsymbol{z}_t - \frac{(1+\kappa)\eta\rho^2}{p}\nabla_{\mathcal{M}}\operatorname{Tr}\left[\nabla^2\mathcal{L}(\boldsymbol{z}_t)/2\right] + \mathcal{O}(\eta^{3/2}\rho^2).$$

Note that $\rho = \mathcal{O}(\sqrt{\eta})$ and $\kappa$ can be as large as $1/\rho$. Consequently, the effective LR of minimizing the trace of Hessian can be selected as large as $\eta_{\mathrm{eff}} = (\kappa + 1)\eta\rho^2/p = \mathcal{O}(\eta^{1.5}/p)$. In contrast, that of average SAM is at most $\mathcal{O}(\eta^2/p)$. The proof of Theorem 5.5 can be found in Appendix D.

### 5.2.2 IRE on standard SAM: An $\Omega(1/\rho)$ acceleration

This subsection delves into IRE on standard SAM (7), which is more widely used and often yields better performance than average SAM (8). However, since standard SAM (7) requires stochastic gradients $\nabla \mathcal{L}_i(\boldsymbol{\theta})$ $(i \in [n])$, we need an additional assumption regarding the features on the manifold (see Setting E.1), which is commonly used in the literature (Du et al., 2018; 2019; Li et al., 2022; Arora et al., 2022; Wen et al., 2023a). We defer it to Appendix E due to space constraints. Under this Setting, Assumption 5.1 holds naturally with $m = n$.

During Phase I of GF, Definition 5.3 ensures that there exists $t < \infty$ such that $\|\boldsymbol{\theta}_t - \Phi(\boldsymbol{\theta}_t)\| = \mathcal{O}(\eta^{1-\alpha}\rho)$ for any $\alpha \in [0, 1)$. We define $T_{\mathrm{I}}$ as the hitting time: $T_{\mathrm{I}} := \inf\{t \geqslant 0 : \|\boldsymbol{\theta}_t - \Phi(\boldsymbol{\theta}_t)\| = \mathcal{O}(\eta^{1-\alpha}\rho)\}$. Then the following result holds for Phase II, whose proof can be founded in Appendix E.

**Theorem 5.6** (IRE on standard SAM). *Under Setting E.1, if $\eta, \rho = \mathcal{O}(1)$ in SAM (7), $\kappa \leqslant 1/\rho$, and $\mathcal{P}_t = P_{n+1:p}(\nabla^2 \mathcal{L}(\boldsymbol{\theta}_t))$ in IRE (4), then with high probability at least $1 - T_{\mathrm{II}}^2 \exp\left(-\Omega(1/\eta^{\alpha})\right)$, $\|\boldsymbol{\theta}_t - \Phi(\boldsymbol{\theta}_t)\| = \mathcal{O}(\eta^{1-\alpha}\rho)$ holds for all $T_{\mathrm{I}} \leqslant t \leqslant T$. Moreover, the effective dynamics $\boldsymbol{z}_t = \Phi(\boldsymbol{\theta}_t) \in \mathcal{M}$ satisfies:*

$$\mathbb{E}_{i_t}[\boldsymbol{z}_{t+1}] = \boldsymbol{z}_t - (1+\kappa)\eta\rho^2\nabla_{\mathcal{M}}\operatorname{Tr}\left[\nabla^2\mathcal{L}(\boldsymbol{z}_t)/2\right] + \mathcal{O}\left((\kappa+1)\eta\rho^2(\rho + \eta^{1-\alpha})\right).$$

Taking $\kappa = 0$ and $\alpha = 0$ recovers the result established in Wen et al. (2023a). However, $\kappa$ can be as large as $1/\rho$, where IRE provides a $\Theta(1/\rho)$-time acceleration over the standard SAM.

## 6 Conclusion

In this work, we propose a novel IRE framework to enhance the implicit sharpness regularization of base optimizers. Experiments demonstrate that IRE not only consistently improves generalization but also accelerates loss convergence in the pre-training of Llama models of various sizes. The code is available at `https://github.com/wmz9/IRE-algorithm-framework`.

For future work, there are two urgent directions: 1) understanding why IRE can accelerate convergence, which may require studying the interplay between IRE and the Edge of Stability (EoS) (Wu et al., 2018; Jastrzębski et al., 2017; Cohen et al., 2021); and 2) conducting a larger-scale investigation into the acceleration of AdmIRE compared to AdamW in LLM pre-training, as well as the downstream performance of the LLMs pre-trained by AdmIRE.

## Acknowledgments

Lei Wu is supported by the National Key R&D Program of China (No. 2022YFA1008200) and National Natural Science Foundation of China (No. 12288101). Mingze Wang is supported in part by the National Key Basic Research Program of China (No. 2015CB856000). We thank Dr. Hongkang Yang, Liu Ziyin, Liming Liu, Zehao Lin, Hao Wu, and Kai Chen for helpful discussions and anonymous reviewers for their valuable suggestions.

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

# Appendix

## A  Other Related Works

**Other Implicit Biases.** Beyond the implicit sharpness regularization, numerous other attempts have explored implicit biases in deep learning algorithms Vardi (2023). Among these, a popular research line is the *max-margin bias*, which suggests that optimizers favor the solutions with large margin, which generalizes well. Soudry et al. (2018) showed that GD converges to max-margin solutions under exponentially-tailed loss on linearly separable data, albeit with an extremely slow rate $\mathcal{O}(1/\log t)$. Furthermore, Nacson et al. (2019c) studied this bias for SGD, Ji and Telgarsky (2019b) investigated linearly non-separable data, and Ji et al. (2020) analyzed the effects of the tail behavior of loss functions.

To achieve faster margin maximization, Nacson et al. (2019b); Ji and Telgarsky (2021) demonstrated that GD with aggressively loss-scaled step sizes can achieve a faster polynomial rate of $\mathcal{O}(1/t)$. Building on this, Ji et al. (2021); Wang et al. (2022) proposed momentum-based gradient methods which achieve a rate of $\mathcal{O}(1/t^2)$. Recently, Wang et al. (2024) established that the polynomial rates for most previous algorithms are tight, and proposed a progressive rescaling gradient descent method that achieves margin maximization exponentially fast $\mathcal{O}(e^{-\Omega(t)})$.

The margin-maximization bias has also been studied for nonlinear models. Ji and Telgarsky (2019a); Gunasekar et al. (2018) examined deep linear networks, while Chizat and Bach (2020) focused on wide two-layer ReLU networks. Notably, Nacson et al. (2019a); Lyu and Li (2019); Ji and Telgarsky (2020) demonstrated that for general homogeneous networks, Gradient Flow (GF) and GD converge to solutions corresponding the KKT point of the max-margin problem. Recently, Kunin et al. (2023) extended this analysis to quasi-homogeneous networks. For two-layer (leaky-)ReLU neural networks, Lyu et al. (2021); Vardi et al. (2022); Wang and Ma (2023) examined whether the convergent KKT point of GF correspond to global optima of the max-margin problem.

Future work could investigate whether the IRE framework can also enhance the margin maximization bias, although it is primarily designed for enhancing the implicit sharpness regularization.

Additionally, Woodworth et al. (2020); Pesme et al. (2021); Nacson et al. (2022); Pesme and Flammarion (2023); Even et al. (2023) conducted fine-grained analyses of training dynamics, examining how initialization and step size impact (S)GD's minima selection in linear diagonal networks.

# B  Proofs for SGD in Section 2

For the example in Section 2, we have studied the implicit sharpness regularization of GD dynamics and how IRE enhances the implicit regularization of GD. In this Section, we illustrate that, for this example, similar results hold for SGD dynamics.

**SDE Modelling of SGD.** We consider SGD approximated by SDE (Li et al., 2017; 2019; 2022) with noise covariance $\Sigma$: $\mathrm{d}\boldsymbol{\theta}_t = -\nabla\mathcal{L}(\boldsymbol{\theta}_t)\mathrm{d}t + \sqrt{\eta\Sigma(\boldsymbol{\theta}_t)}\mathrm{d}W_t$. We consider that the noise covariance aligns with the Hessian near minima, i.e., $\Sigma(\boldsymbol{\theta}) = \begin{pmatrix} \mathbf{0}_d & 0 \\ 0 & \sigma^2 h(\boldsymbol{u}) \end{pmatrix}$ (where $\sigma > 0$ is a scalar), such as the label noise (Damian et al., 2021; Li et al., 2022). Then, the SDE above can be rewritten as

$$\mathrm{d}\begin{pmatrix} \boldsymbol{u}_t \\ v_t \end{pmatrix} = -\begin{pmatrix} v_t^2 \nabla h(\boldsymbol{u}_t)/2 \\ v_t h(\boldsymbol{u}_t) \end{pmatrix}\mathrm{d}t + \begin{pmatrix} 0 \\ \sqrt{\eta\sigma^2 h(\boldsymbol{u}_t)}\mathrm{d}W_t \end{pmatrix}. \tag{10}$$

**Implicit Sharpness Regularization.** Intuitively, when $v_t$ is close to 0, the speed of $\boldsymbol{u}_t$ is much slower than $v_t$ due to $v_t^2 \ll v_t$. Following Ma et al. (2022), when this speed separation is large, $v_t$ is always at equilibrium given $\boldsymbol{u}_t$. Solving the Ornstein–Uhlenbeck process about $v_t$, we know the equilibrium distribution of $v_t$ is $v_\infty \sim \mathcal{N}(0, \eta\sigma^2)$, and hence the dynamics $\boldsymbol{u}_t$ (along the manifold) is

$$\mathrm{d}\boldsymbol{u}_t/\mathrm{d}t = -\mathbb{E}_{v_\infty}\left[v_\infty^2 \nabla h(\boldsymbol{u}_t)/2\right] = -\nabla h(\boldsymbol{u}_t)/2. \tag{11}$$

This derivation clearly shows the slow "effective dynamics" $\boldsymbol{u}_t$ along the manifold is a *gradient flow minimizing the sharpness $h(\cdot)$*. When SGD minimizes the loss, it also minimzes the sharpness implicitly, that is to say, SGD has the following *implicit sharpness regularization*: $\min_{\boldsymbol{\theta}} \mathrm{Tr}(\nabla^2 \mathcal{L}(\boldsymbol{\theta}))$ s.t. $\mathcal{L}(\boldsymbol{\theta}) \approx 0$.

**Generalization and Optimization benefits** of the sharpness regularization. In terms of generalization, as discussed in related works, a common view is that *flat minima generalize well*, which has been proved in a large number of previous works. In addition, in terms of optimization, after SGD reaches the equilibrium $v_\infty \sim \mathcal{N}(0, \eta\sigma^2)$, the loss near the flat minimum is smaller because $\mathbb{E}_{v_\infty}[\mathcal{L}(\boldsymbol{\theta})] = \mathbb{E}_{v_\infty}[h(\boldsymbol{u})v_\infty^2/2] = \eta\sigma^2 h(\boldsymbol{u})/2 \propto \mathrm{Tr}(\nabla^2 \mathcal{L}(\boldsymbol{u}))$.

**Q.** *How can we enhance the implicit sharpness regularization of SGD?*
**A.** *Accelerating SGD's slow "effective dynamics" $\boldsymbol{u}_t$ along the minima manifold.*

**Implicit Regularization Enhancement (IRE)** by accelerating the effective dynamics along minima manifold. First, it is worth noting that naively increasing the learning rate $\eta$ cannot achieve our aim, because increasing $\eta$ will influence the dynamic stability of $v_t$ and the equilibrium $v_\infty$. Therefore, we need to accelerate the effective dynamics $\boldsymbol{u}_t$ without affecting the dynamics of $v_t$. Another main point is that SGD's effective dynamics $\boldsymbol{u}_t$ can naturally minimize the sharpness implicitly, so we only need to enhance this property. To achieve this, we only need to correct the update direction in (10) from $-\nabla\mathcal{L}(\boldsymbol{\theta}_t)$ to $-(\nabla\mathcal{L}(\boldsymbol{\theta}_t) + \kappa P_\mathcal{M}\nabla\mathcal{L}(\boldsymbol{\theta}_t))$, where $P_\mathcal{M}$ is the projection matrix to the manifold $\mathcal{M}$ and $\kappa$ is a scalar. Using this new algorithm, the SDE dynamics corresponds to

$$\mathrm{d}\begin{pmatrix} \boldsymbol{u}_t \\ v_t \end{pmatrix} = -\begin{pmatrix} (1+\kappa)v_t^2 \nabla h(\boldsymbol{u}_t)/2 \\ v_t h(\boldsymbol{u}_t) \end{pmatrix}\mathrm{d}t + \begin{pmatrix} 0 \\ \sqrt{\eta\sigma^2 h(\boldsymbol{u}_t)}\mathrm{d}W_t \end{pmatrix}. \tag{12}$$

Comparing (10) and (12), the dynamics of $v_t$ are the same, so they attain the same equilibrium distribution $v_\infty \sim \mathcal{N}(0, \eta\sigma^2)$. As for the effective dynamics along manifold, (10) corresponds to the form:

$$\mathrm{d}\boldsymbol{u}_t/\mathrm{d}t = -\mathbb{E}_{v_\infty}\left[(1+\kappa)v_\infty^2 \nabla h(\boldsymbol{u}_t)/2\right] = -(1+\kappa)\nabla h(\boldsymbol{u}_t)/2. \tag{13}$$

$(1+\kappa)$ **times Enhancement.** Comparing (13) and (11), it is clear that our new algorithm can enhance implicit sharpness regularization $(1+\kappa)$ times faster than the original SGD.

## C  Experimental Details

This section describes the experimental details in Section 4.

### C.1  Experimental details in Section 4.1

#### C.1.1  Experimental details in Section 4.1.1

We train WideResNet-16-8 on CIFAR-10 dataset by SAM-IRE (with $K = 10$, varying $\kappa$ and $\gamma$). The experiments employ basic data augmentations and 0.1 label smoothing, as outlined by Foret et al. (2021). The mini-batch size is set to 128, the weight decay is set to 5e-4, and the $\rho$ in SAM is to 0.05, as in Foret et al. (2021). To evaluate the implicit flatness regularization of SAM itself, the momentum is set to 0.0. Regarding the learning rate (lr), we evaluate for both constant lr (within our theoretical framework) and decayed lr (common in practice though not covered by our theory). In the experiment in Fig 2 (a), a fixed lr 0.1 is used. In the experiment in Fig 2 (b)(c)(d), a step-decayed lr schedule is employed, starting at 0.1 and reducing lr by a factor of 5 at epoch 20, 50, 80. We transit from SAM to SAM-IRE at the 30th epoch out of 100 total epochs. We test $\gamma$ in 0.8, 0.9, 0.95, and $\kappa$ in 0 (original SAM), 2, 5, 10.

The flatness measure, $\mathrm{Tr}(\nabla^2 \mathcal{L}(\boldsymbol{\theta}))$, is approximated by the trace of Fisher $\mathrm{Tr}(\boldsymbol{F}(\boldsymbol{\theta}))$. Specifically, we use $\mathrm{diag}(\hat{F}_{\mathrm{eff}}(\boldsymbol{\theta}))$ in (6) for the estimate because $\mathbb{E}_{\hat{\boldsymbol{y}}}[\mathrm{diag}(\hat{F}_{\mathrm{eff}}(\boldsymbol{\theta}))] = \mathbb{E}_{\hat{\boldsymbol{y}}}[\mathrm{diag}(\hat{F}(\boldsymbol{\theta}))]$ and thus,

$$\mathrm{Tr}(\nabla^2 \mathcal{L}(\boldsymbol{\theta})) \approx \mathbb{E}_{\hat{\boldsymbol{y}}}[\mathrm{Tr}(\hat{F}(\boldsymbol{\theta}))] = \mathbb{E}_{\hat{\boldsymbol{y}}}[\mathrm{Tr}(\mathrm{diag}(\hat{F}(\boldsymbol{\theta})))] = \mathbb{E}_{\hat{\boldsymbol{y}}}[\mathrm{diag}(\hat{F}_{\mathrm{eff}}(\boldsymbol{\theta}))].$$

Moreover, the first "$\approx$" above takes "=" when $\mathcal{L}(\boldsymbol{\theta}) = 0$.

In this section, all experiments were conducted using a single A800 GPU.

#### C.1.2  Experimental details in Section 4.1.2

**Experiments for CNNs on CIFAR-10/CIFAR-100.** We first evaluate the impact of IRE on generalization of baseline models (WideResNet-28-10 and ResNet-56) and default optimizers (SGD and SAM) on CIFAR-{10,100}. For the base optimizers, SGD and SAM, cosine learning rate decay is adopted with an initial lr 0.1. For other training components, we follow the settings in Foret et al. (2021): basic data augmentations and 0.1 label smoothing; for both SGD and SAM, the momentum is set to 0.9, the batch size is set to 128, and the weight decay is set to 5e-4; for SAM, $\rho$ is set to 0.05 for CIFAR-10 and 0.1 for CIFAR-100. The total epochs is set to 100 for CIFAR-10 and 200 for CIFAR-100, and we switch from SGD/SAM to SGD-IRE/SAM-IRE when the training loss approaches 0.1. For SGD-IRE/SAM-IRE, we fix $K = 10$, and tune hyperparameters $\gamma$ and $\kappa$ via a grid search over $\gamma \in \{0.99, 0.9, 0.8\}$ and $\kappa \in \{1, 2\}$. The results are reported in Table 1.

**Experiments without finely tuned hyperparameters.** A high sensitivity to the choice of hyperparameters would make a method less practical. To demonstrate that our IRE performs *even when $\kappa, \gamma$ are not finely tuned*, we conduct experiments using fixed $\gamma = 0.99, \kappa = 1$, under the same settings as described above.. The results (averaged over 3 random seeds) are reported in Table 8.

Table 8: Results for SGD-IRE and SAM-IRE on {WideResNet-28-10, ResNet-56} on CIFAR-{10, 100}, using fixed $\gamma = 0.99, \kappa = 1$ in IRE.

| | WideResNet-28-10 | | ResNet-56 | |
| --- | --- | --- | --- | --- |
| | CIFAR-10 | CIFAR-100 | CIFAR-10 | CIFAR-100 |
| SGD | 95.93 | 80.77 | 93.80 | 72.72 |
| SGD-IRE | **96.13 (+0.20)** | **81.12 (+0.35)** | **93.94(+0.14)** | **72.93(+0.21)** |
| SAM | 96.73 | 83.22 | 94.58 | 75.25 |
| SAM-IRE | **96.75 (+0.02)** | **83.40 (+0.19)** | **94.65 (+0.07)** | **75.49 (+0.24)** |

**Experiments for CNNs on ImageNet.** We also examine the impact of IRE on generalization of ResNet-50 and default optimizers (SGD and SAM) on on ImageNet. Following Foret et al. (2021) and Kwon et al. (2021), we use basic data augmentations and 0.1 label smoothing. For the base optimizers, SGD and SAM, we also follow the settings in Kwon et al. (2021): the momentum is set to 0.9; cosine learning rate decay is adopted with an initial lr 0.2; the batch size is set to 1024; the weight decay is set to 1e-4; for SAM, $\rho$ is set to 0.05. The total epochs is set to 200, and we switch from SGD/SAM to SGD-IRE/SAM-IRE when the training loss approaches 1.5. For SGD-IRE/SAM-IRE, we fix $K = 10$, and tune hyperparameters $\gamma$ and $\kappa$ via a grid search over $\gamma \in \{0.8, 0.6\}$ and $\kappa \in \{2, 4\}$. The results are reported in Table 2.

**Experiments for ViTs on CIFAR-100.** We examine the impact of IRE on generalization of ViT-T and ViT-S on CIFAR-100. We follow the settings in Mueller et al. (2024): the default optimizers used are AdamW and SAM, with cosine lr decay to 0 starting from an initial lr 1e-4; the weight decay is 5e-4; batch size is 64; strong data augmentations (basic + AutoAugment) are utilized; $\rho = 0.1$ for SAM. The total epochs are set to 200, and we switch from AdamW/SAM to AdmIRE/SAM-IRE when the training loss approaches 0.5. For AdmIRE/SAM-IRE, we fix $K = 10$, and tune hyperparameters $\gamma$ and $\kappa$ via a grid search over $\gamma \in \{0.99, 0.9, 0.8\}$ and $\kappa \in \{20, 50\}$. The results are reported in Table 3.

**Experiments for ViTs on ImageNet.** We also examine the impact of IRE on generalization of ViT-S/16 on ImageNet. We follow the settings in Chen et al. (2024): RandAugment and Mixup with $\alpha = 0.5$ are utilized; the default optimizer used is AdamW with hyperparameters $\beta_1 = 0.9, \beta_2 = 0.999$ and weight decay $\lambda = 1.0$; the learning rate strategy integrates a warm-up phase followed by a cosine decay scheduler with `lr_max=3e-3`; batch size is 4096; the total training duration is 300 epochs, including 30 warm-up epochs; For AdmIRE, we switch from AdamW to AdmIRE at epoch 100 and examine different $\gamma, \kappa$. The results are reported in Table 4.

In this section, the experiments on CIFAR-10/CIFAR-100 were conducted using a single A800 GPU, and the experiments on ImageNet were conducted using 4 A800 GPUs.

## C.2 Experimental details in Section 4.2

### C.2.1 Experimental details in Section 4.2.1

We train a 2-layer decoder-only Transformer (8M parameters) using absolute positional encodings (Vaswani et al., 2017), with 8 heads in each layer and a hidden size of 128, on the `wikitext-2` dataset (4.3M) by AdamW and AdmIRE (with $K = 10$ and varying $\gamma, \kappa$). The (max) sequence length is set to 512, and the batch size is set to 32. The experiments in this section are conducted on 1 A800.

For the optimizer AdamW, we use the hyperparameters $\beta_1 = 0.9, \beta_2 = 0.95$ and weight decay $\lambda = 0.1$, as suggested in Touvron et al. (2023). The total training duration is 100,000 steps, including 3,000 warm-up steps followed by a cosine decay scheduler with `lr_max` and `lr_min=lr_max/20`.

First, we tune `lr_max` in AdamW from $\{$`1.5e-4, 3e-4, 6e-4, 1.2e-3, 1.8e-3, 3e-3`$\}$. The results, shown in Figure 5 (left), identify the optimal `lr_max=6e-4`. We also use the optimal `lr_max` of `6e-4` for AdmIRE, for which the IRE is enable at the end of warm-up phase.

The results are reported in Figure 3.

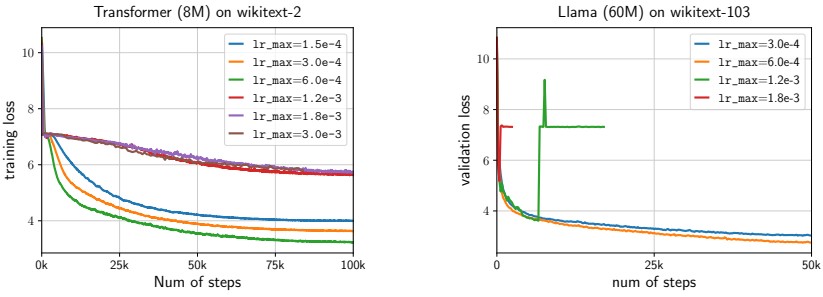

Figure 5: The results for tuning `lr_max` in AdamW.

### C.2.2 Experimental details in Section 4.2.2

Llama (Touvron et al., 2023) is a decode-only Transformer architecture using Rotary Positional Encoding (RoPE) (Su et al., 2024), Swish-Gated Linear Unit (SwiGLU) (Shazeer, 2020), and Root mean square layer normalization (RMSNorm) (Zhang and Sennrich, 2019). The experiments in this section examine the performance of AdmIRE in training Llama models with different sizes. For implementation, we utilize the Llama code available on huggingface. The experiments are conducted on 4 H800.

For the optimizer AdamW, we use the well-tuned hyperparameters $\beta_1 = 0.9, \beta_2 = 0.95$ and weight decay $\lambda = 0.1$ for LLama (Touvron et al., 2023). The learning rate strategy integrates a warm-up phase followed by a cosine decay scheduler with `lr_max` and `lr_min=lr_max/20`. Additionally, it is used with gradient clipping 1.0 to maintain the training stability.

In each experiment, we tune the optimal `lr_max` for AdamW and then use it also for AdmIRE, for which the IRE is enable at the end of warm-up phase.

**Llama (60M) on wikitext-103 (0.5G).** We train a 16-layer Llama model, with 10 heads in each layer and a hidden size of 410, on the wikitext-103 dataset. The (max) sequence length is set to 150, and the batch size is set to 240, following Dai et al. (2019). The total training duration is 50,000 or 100,000 steps, including 500 warm-up steps. First, we tune `lr_max` in AdamW from {3e-4, 6e-4, 1.2e-3, 1.8e-3}, identifying the optimal `lr_max` 6e-4. The experimental results are very similar to Figure 5 (right), so we will not show them again. Then, both AdamW and AdmIRE are trained using the optimal `lr_max`.

**Llama (119M) on minipile (6G).** We train a 6-layer Llama model, with 12 heads in each layer and a hidden size of 768, on the minipile dataset. The (max) sequence length is set to 512, and the batch size is set to 300. The total training duration is 30,000 or 60,000 steps, including 300 warm-up steps. First, we tune `lr_max` in AdamW from {3e-4, 6e-4, 1.2e-3, 1.8e-3}, identifying the optimal `lr_max` 6e-4. (The results are very similar to Figure 5 (right), so we do not show them repeatly.) Then, both AdamW and AdmIRE are trained using the optimal `lr_max`.

**Llama (229M) on openwebtext (38G).** We train a 16-layer Llama model, with 12 heads in each layer and a hidden size of 768, on the openwebtext dataset. The (max) sequence length is set to 1024, and the batch size is set to 480, following nanoGPT and Liu et al. (2024). The total training duration is 50,000 or 100,000 steps, including 1,000 warm-up steps. First, we tune `lr_max` in AdamW from {3e-4, 6e-4, 1.2e-3, 1.8e-3}, identifying the optimal `lr_max` 6e-4. (The results are very similar to Figure 5 (right), so we do not show them repearly.) Then, both AdamW and AdmIRE are trained using the optimal `lr_max`.

## D Proofs in Section 5.2.1

**Additional Notations.** For the proofs in Section 5, some additional notations are used. For any set $K \subset \mathbb{R}^p$ and a constant $R > 0$, we denote $\mathbb{B}(K; R) := \{\boldsymbol{\theta} \in \mathbb{R}^p : \mathrm{dist}(\boldsymbol{\theta}; K) \leqslant R\}$. $\langle \cdot, \cdot \rangle$ represents the standard Euclidean inner product between two vectors. $\|\cdot\|$ denotes the $\ell_2$ norm of a vector or the spectral norm of a matrix, whereas $\|\cdot\|_F$ denotes the Frobenius norm of a matrix.

### D.1 Preliminary Lemmas

**Lemma D.1** (Arora et al. (2022), Lemma B.2). *Under Assumption 5.1, for any compact set $K \subset \Gamma$, there exist absolute constants $R_1, \mu > 0$ such that*

- *(i) $\mathbb{B}(K; R_1) \subset U$;*

- *(ii) $\mathcal{L}(\cdot)$ is $\mu$-PL (defined in Def F.1) on $\mathbb{B}(K; R_1)$;*

- *(iii) $\inf_{\boldsymbol{\theta} \in \mathbb{B}(K; R_1)} \lambda_m \left( \nabla^2 \mathcal{L}(\boldsymbol{\theta}) \right) \geqslant \mu$.*

*We further define the following absolute constants on $\mathbb{B}(K; R_1)$:*

$$\beta_2 := \sup_{\boldsymbol{\theta} \in \mathbb{B}(K;R_1)} \left\| \nabla^2 \mathcal{L}(\boldsymbol{\theta}) \right\|; \quad \beta_3 := \sup_{\boldsymbol{\theta} \in \mathbb{B}(K;R_1)} \left\| \nabla^3 \mathcal{L}(\boldsymbol{\theta}) \right\|; \quad \beta_4 := \sup_{\boldsymbol{\theta} \in \mathbb{B}(K;R_1)} \left\| \nabla^4 \mathcal{L}(\boldsymbol{\theta}) \right\|;$$

$$\nu := \inf_{\boldsymbol{\theta} \in \mathbb{B}(K;R_1)} \lambda_m \left( \nabla^2 \mathcal{L}(\boldsymbol{\theta}) \right); \quad \zeta_\Phi := \sup_{\boldsymbol{\theta} \in \mathbb{B}(K;R_1)} \left\| \nabla^2 \Phi(\boldsymbol{\theta}) \right\|.$$

**Lemma D.2** (Key properties of $\Phi(\cdot)$ (Arora et al., 2022)). *Under Assumption 5.1,*

- *For any $\boldsymbol{\theta} \in U$, $\partial\Phi(\boldsymbol{\theta})\nabla\mathcal{L}(\boldsymbol{\theta}) = \boldsymbol{0}$.*

- *For any $\boldsymbol{\theta} \in \Gamma$, $\partial\Phi(\boldsymbol{\theta}) = P_{m+1:p}(\nabla^2\mathcal{L}(\boldsymbol{\theta}))$.*

**Lemma D.3** (Continuity of $P_{m+1:p}$). *Under Assumption 5.1, there exists absolute constants $R_2, \zeta_P > 0$ such that for any $\boldsymbol{\theta} \in \mathbb{B}(K; R_2)$,*

$$\left\| P_{m+1:p}(\nabla^2\mathcal{L}(\boldsymbol{\theta})) - P_{m+1:p}(\nabla^2\mathcal{L}(\Phi(\boldsymbol{\theta}))) \right\| \leqslant \zeta_P \left\| \boldsymbol{\theta} - \Phi(\boldsymbol{\theta}) \right\|.$$

*Proof of Lemma D.3.*
Let the orthogonal decomposition of $\nabla^2\mathcal{L}(\boldsymbol{\theta})$ and $\nabla^2\mathcal{L}(\Phi(\boldsymbol{\theta}))$ be $\nabla^2\mathcal{L}(\boldsymbol{\theta}) = \sum_{k=1}^p \lambda_k \boldsymbol{u}_k \boldsymbol{u}_k^\top$ ($\lambda_1 \geqslant \cdots \geqslant \lambda_p$) and $\nabla^2\mathcal{L}(\Phi(\boldsymbol{\theta})) = \sum_{k=1}^p \mu_k \boldsymbol{v}_k \boldsymbol{v}_k^\top$ ($\mu_1 \geqslant \cdots \geqslant \mu_p$), respectively.

By Lemma D.1, for any $\boldsymbol{\theta} \in \mathbb{B}(K; R_1)$, it holds that $\left\| \nabla^2\mathcal{L}(\boldsymbol{\theta}) - \nabla^2\mathcal{L}(\Phi(\boldsymbol{\theta})) \right\| \leqslant \beta_3 \left\| \boldsymbol{\theta} - \Phi(\boldsymbol{\theta}) \right\|$. We choose $R_2 := R_1 \wedge \frac{\mu}{4\beta_3}$. Then for any $\boldsymbol{\theta} \in \mathbb{B}(K; R_2)$,

$$\left\| \nabla^2\mathcal{L}(\boldsymbol{\theta}) - \nabla^2\mathcal{L}(\Phi(\boldsymbol{\theta})) \right\| \leqslant \beta_3 \left\| \boldsymbol{\theta} - \Phi(\boldsymbol{\theta}) \right\| \leqslant \frac{\mu}{4}.$$

Consequently, by Lemma F.2, we can bound the gap of eigenvalues: for any $k \in [p]$,

$$|\lambda_k - \mu_k| \leqslant \left\| \nabla^2\mathcal{L}(\boldsymbol{\theta}) - \nabla^2\mathcal{L}(\Phi(\boldsymbol{\theta})) \right\| \leqslant \frac{\mu}{4}.$$

Noticing $\Phi(\boldsymbol{\theta}) \in \Gamma$, by Lemma D.1, it holds that $\mu_1 \geqslant \mu_m \geqslant \mu$ and $\mu_{m+1} = \cdots = \mu_p = 0$. Thus, we can obtain the bounds of $\{\lambda_k\}_k$:

$$\text{for all } k \leqslant m, \quad \lambda_k \geqslant \lambda_m \geqslant \mu_m - |\lambda_m - \mu_m| \geqslant \frac{3\mu}{4};$$

$$\text{for all } k \geqslant m+1, \quad \lambda_k \leqslant \lambda_{m+1} \leqslant \mu_{m+1} + |\lambda_{m+1} - \mu_{m+1}| \leqslant \frac{\mu}{4}.$$

For simplicity, we denote $\boldsymbol{U}_{\text{top}} := (\boldsymbol{u}_1, \cdots, \boldsymbol{u}_m)$, $\boldsymbol{U}_{\text{bottom}} := (\boldsymbol{u}_{m+1}, \cdots, \boldsymbol{u}_p)$, $\boldsymbol{V}_{\text{top}} := (\boldsymbol{v}_1, \cdots, \boldsymbol{v}_m)$, $\boldsymbol{V}_{\text{bottom}} := (\boldsymbol{v}_{m+1}, \cdots, \boldsymbol{v}_p)$.

By Lemma F.3, we can bound the gap between the subspaces:

$$\left\| \boldsymbol{U}_{\text{bottom}}^\top \boldsymbol{V}_{\text{top}} \right\|_{\text{F}} \leqslant \frac{\left\| \boldsymbol{U}_{\text{bottom}}^\top (\nabla^2\mathcal{L}(\boldsymbol{\theta}) - \nabla^2\mathcal{L}(\Phi(\boldsymbol{\theta}))) \boldsymbol{V}_{\text{top}} \right\|_{\text{F}}}{\frac{3\mu}{4} - \frac{\mu}{4}}$$

$$\overset{\text{Lemma F.6}}{\leqslant} \begin{cases} \frac{2}{\mu} \left\| \boldsymbol{U}_{\text{bottom}}^\top \right\| \left\| \nabla^2\mathcal{L}(\boldsymbol{\theta}) - \nabla^2\mathcal{L}(\Phi(\boldsymbol{\theta})) \right\| \left\| \boldsymbol{V}_{\text{top}} \right\|_{\text{F}} \\ \frac{2}{\mu} \left\| \boldsymbol{U}_{\text{bottom}}^\top \right\|_{\text{F}} \left\| \nabla^2\mathcal{L}(\boldsymbol{\theta}) - \nabla^2\mathcal{L}(\Phi(\boldsymbol{\theta})) \right\| \left\| \boldsymbol{V}_{\text{top}} \right\| \end{cases}$$

$$= \frac{2\left(m \wedge (p-m)\right)}{\mu} \left\| \nabla^2\mathcal{L}(\boldsymbol{\theta}) - \nabla^2\mathcal{L}(\Phi(\boldsymbol{\theta})) \right\|.$$

According to the definition of $P_{m+1:p}(\cdot)$, it holds that

$$P_{m+1:p}(\nabla^2\mathcal{L}(\boldsymbol{\theta})) = \sum_{k=m+1}^p \boldsymbol{u}_k \boldsymbol{u}_k^\top = \boldsymbol{U}_{\text{bottom}} \boldsymbol{U}_{\text{bottom}}^\top,$$

$$P_{m+1:p}(\nabla^2\mathcal{L}(\Phi(\boldsymbol{\theta}))) = \sum_{k=m+1}^p \boldsymbol{v}_k \boldsymbol{v}_k^\top = \boldsymbol{V}_{\text{bottom}} \boldsymbol{V}_{\text{bottom}}^\top.$$

Noticing the relationship

$$\left\|P_{m+1:p}(\nabla^2\mathcal{L}(\boldsymbol{\theta})) - P_{m+1:p}(\nabla^2\mathcal{L}(\Phi(\boldsymbol{\theta})))\right\|^2 \leqslant \left\|P_{m+1:p}(\nabla^2\mathcal{L}(\boldsymbol{\theta})) - P_{m+1:p}(\nabla^2\mathcal{L}(\Phi(\boldsymbol{\theta})))\right\|_{\mathrm{F}}^2$$

$$= \left\|\boldsymbol{U}_{\mathrm{bottom}}\boldsymbol{U}_{\mathrm{bottom}}^\top - \boldsymbol{V}_{\mathrm{bottom}}\boldsymbol{V}_{\mathrm{bottom}}^\top\right\|_{\mathrm{F}}^2 = 2(p-m) - 2\operatorname{Tr}\left(\boldsymbol{U}_{\mathrm{bottom}}\boldsymbol{U}_{\mathrm{bottom}}^\top\boldsymbol{V}_{\mathrm{bottom}}\boldsymbol{V}_{\mathrm{bottom}}^\top\right)$$

$$= 2\operatorname{Tr}\left(\boldsymbol{U}_{\mathrm{bottom}}\boldsymbol{U}_{\mathrm{bottom}}^\top\left(\boldsymbol{I} - \boldsymbol{V}_{\mathrm{bottom}}\boldsymbol{V}_{\mathrm{bottom}}^\top\right)\right) = 2\operatorname{Tr}\left(\boldsymbol{U}_{\mathrm{bottom}}\boldsymbol{U}_{\mathrm{bottom}}^\top\boldsymbol{V}_{\mathrm{top}}\boldsymbol{V}_{\mathrm{top}}^\top\right)$$

$$= 2\left\|\boldsymbol{U}_{\mathrm{bottom}}^\top\boldsymbol{V}_{\mathrm{top}}\right\|_{\mathrm{F}}^2,$$

we obtain the bound:

$$\left\|P_{m+1:p}(\nabla^2\mathcal{L}(\boldsymbol{\theta})) - P_{m+1:p}(\nabla^2\mathcal{L}(\Phi(\boldsymbol{\theta})))\right\| \leqslant \sqrt{2}\left\|\boldsymbol{U}_{\mathrm{bottom}}^\top\boldsymbol{V}_{\mathrm{top}}\right\|_{\mathrm{F}}$$

$$\leqslant \frac{2\sqrt{2}\,(m\wedge(p-m))}{\mu}\left\|\nabla^2\mathcal{L}(\boldsymbol{\theta}) - \nabla^2\mathcal{L}(\Phi(\boldsymbol{\theta}))\right\| \leqslant \frac{2\sqrt{2}\,(m\wedge(p-m))\,\beta_3}{\mu}\left\|\boldsymbol{\theta} - \Phi(\boldsymbol{\theta})\right\|.$$

To summarize, we only need to choose the constants $R_2 = R_1 \wedge \frac{\mu}{4\beta_3}$ and $\zeta_P = \frac{2\sqrt{2}(m\wedge(p-m))\beta_3}{\mu}$ to ensure this lemma holds. $\qquad\square$

**Proof Notations.** Now we introduce some additional useful notations in the proof in this section.

First, we choose $R := (R_1 \wedge R_2)/2$, where $R_1$ is defined in Lemma D.1 and $R_2$ is defined in Lemma D.3. Let $\mu$ be the PL constant on $\mathbb{B}(K;R)$. Moreover, we use the following notations:

$$\beta_2 := \sup_{\boldsymbol{\theta}\in\mathbb{B}(K;R)}\left\|\nabla^2\mathcal{L}(\boldsymbol{\theta})\right\|;\quad \beta_3 := \sup_{\boldsymbol{\theta}\in\mathbb{B}(K;R)}\left\|\nabla^3\mathcal{L}(\boldsymbol{\theta})\right\|;\quad \beta_4 := \sup_{\boldsymbol{\theta}\in\mathbb{B}(K;R)}\left\|\nabla^4\mathcal{L}(\boldsymbol{\theta})\right\|;$$

$$\nu := \inf_{\boldsymbol{\theta}\in\mathbb{B}(K;R)}\lambda_m\left(\nabla^2\mathcal{L}(\boldsymbol{\theta})\right);\quad \zeta_\Phi := \sup_{\boldsymbol{\theta}\in\mathbb{B}(K;R)}\left\|\nabla^2\Phi(\boldsymbol{\theta})\right\|;\qquad\qquad (14)$$

$$\zeta_P := \sup_{\boldsymbol{\theta}\in\mathbb{B}(K;R)-\Gamma}\frac{\left\|P_{m+1:p}(\nabla^2\mathcal{L}(\boldsymbol{\theta})) - P_{m+1:p}(\nabla^2\mathcal{L}(\Phi(\boldsymbol{\theta})))\right\|}{\left\|\boldsymbol{\theta} - \Phi(\boldsymbol{\theta})\right\|}.$$

Ensured by Lemma D.1 and D.3, these quantities are all absolute constants in $(0, +\infty)$. Moreover, without loss of generality, we can assume that $\beta_1, \beta_2, \beta_3, \zeta_\Phi, \zeta_P > 1$ and $\mu \leqslant \nu < 1$.

**Lemma D.4** (Connections between para norm, grad norm, and loss). *For any $\boldsymbol{\theta}\in\mathbb{B}(K;R) > 0$, it holds that:*

- *(para norm v.s. grad norm)* $\mu\left\|\nabla\mathcal{L}(\boldsymbol{\theta})\right\| \leqslant \left\|\boldsymbol{\theta} - \Phi(\boldsymbol{\theta})\right\| \leqslant \beta_2\left\|\nabla\mathcal{L}(\boldsymbol{\theta})\right\|$;

- *(grad norm v.s. loss)* $2\mu\mathcal{L}(\boldsymbol{\theta}) \leqslant \left\|\nabla\mathcal{L}(\boldsymbol{\theta})\right\|^2 \leqslant \frac{2\beta_2^2}{\mu}\mathcal{L}(\boldsymbol{\theta})$;

- *(loss v.s. para norm)* $\frac{\mu}{2}\left\|\boldsymbol{\theta} - \Phi(\boldsymbol{\theta})\right\|^2 \leqslant \mathcal{L}(\boldsymbol{\theta}) \leqslant \frac{\beta_2^2}{2\mu}\left\|\boldsymbol{\theta} - \Phi(\boldsymbol{\theta})\right\|^2$.

*Proof of Lemma D.4.* This lemma is a corollary of local PL and smoothness (Lemma D.1). For the three lower bounds, please refer to the proof of Lemma B.6 in Arora et al. (2022). Then utilizing these lower bounds and the smoothness $\beta_2$, the upper bounds hold naturally. $\qquad\square$

**Lemma D.5.** *For all $\boldsymbol{\theta}\in\mathbb{B}(K;R)$,*

- $\left\|P_{m+1:p}(\nabla^2\mathcal{L}(\boldsymbol{\theta}))\nabla^2\mathcal{L}(\boldsymbol{\theta})\right\| \leqslant \mathcal{O}\left(\left\|\boldsymbol{\theta} - \Phi(\boldsymbol{\theta})\right\|\right)$;

- $\left\|P_{m+1:p}(\nabla^2\mathcal{L}(\boldsymbol{\theta}))\nabla\mathcal{L}(\boldsymbol{\theta})\right\| \leqslant \mathcal{O}\left(\left\|\boldsymbol{\theta} - \Phi(\boldsymbol{\theta})\right\|^2\right)$;

- *Let $\rho > 0$ and $\boldsymbol{v}\in\mathbb{S}^{p-1}$. If $\boldsymbol{\theta} + \rho\boldsymbol{v} \in \mathbb{B}(K;R)$, then*

$$\left\|\nabla\mathcal{L}(\boldsymbol{\theta} + \rho\boldsymbol{v})\right\| \leqslant \left\|\nabla\mathcal{L}(\boldsymbol{\theta})\right\| + \rho\beta_2;$$

$$\left\|P_{m+1:p}(\nabla^2\mathcal{L}(\boldsymbol{\theta}))\nabla\mathcal{L}(\boldsymbol{\theta} + \rho\boldsymbol{v})\right\| \leqslant \mathcal{O}\left(\left\|\boldsymbol{\theta} - \Phi(\boldsymbol{\theta})\right\|^2\right) + \mathcal{O}\left(\rho\left\|\boldsymbol{\theta} - \Phi(\boldsymbol{\theta})\right\|\right) + \frac{\rho^2\beta_3}{2}.$$

*Proof of Lemma D.5.*

$$\left\|P_{m+1:p}(\nabla^2\mathcal{L}(\boldsymbol{\theta}))\nabla^2\mathcal{L}(\boldsymbol{\theta})\right\| \leqslant \left\|P_{m+1:p}(\nabla^2\mathcal{L}(\Phi(\boldsymbol{\theta})))\nabla^2\mathcal{L}(\Phi(\boldsymbol{\theta}))\right\| + \zeta_P\beta_2\left\|\boldsymbol{\theta}-\Phi(\boldsymbol{\theta})\right\| + \beta_3\left\|\boldsymbol{\theta}-\Phi(\boldsymbol{\theta})\right\|$$
$$= 0 + \mathcal{O}\left(\left\|\boldsymbol{\theta}-\Phi(\boldsymbol{\theta})\right\|\right);$$

$$\left\|P_{m+1:p}(\nabla^2\mathcal{L}(\boldsymbol{\theta}))\nabla\mathcal{L}(\boldsymbol{\theta})\right\| \leqslant \left\|P_{m+1:p}(\nabla^2\mathcal{L}(\boldsymbol{\theta}))\nabla^2\mathcal{L}(\Phi(\boldsymbol{\theta}))(\Phi(\boldsymbol{\theta})-\boldsymbol{\theta})\right\| + \mathcal{O}\left(\left\|\boldsymbol{\theta}-\Phi(\boldsymbol{\theta})\right\|^2\right)$$
$$\leqslant \left\|P_{m+1:p}(\nabla^2\mathcal{L}(\Phi(\boldsymbol{\theta})))\nabla^2\mathcal{L}(\Phi(\boldsymbol{\theta}))(\Phi(\boldsymbol{\theta})-\boldsymbol{\theta})\right\| + \mathcal{O}\left(\left\|\boldsymbol{\theta}-\Phi(\boldsymbol{\theta})\right\|^2\right) = \mathcal{O}\left(\left\|\boldsymbol{\theta}-\Phi(\boldsymbol{\theta})\right\|^2\right);$$

$$\left\|\nabla\mathcal{L}(\boldsymbol{\theta}+\rho\boldsymbol{v})\right\| \leqslant \left\|\nabla\mathcal{L}(\boldsymbol{\theta})\right\| + \rho\beta_2;$$

$$\left\|P_{m+1:p}(\nabla^2\mathcal{L}(\boldsymbol{\theta}))\nabla\mathcal{L}(\boldsymbol{\theta}+\rho\boldsymbol{v})\right\|$$
$$\leqslant \left\|P_{m+1:p}(\nabla^2\mathcal{L}(\boldsymbol{\theta}))\nabla\mathcal{L}(\boldsymbol{\theta})\right\| + \rho\left\|P_{m+1:p}(\nabla^2\mathcal{L}(\boldsymbol{\theta}))\nabla^2\mathcal{L}(\boldsymbol{\theta})\boldsymbol{v}\right\| + \frac{\rho^2\beta_3}{2}$$
$$\leqslant \mathcal{O}\left(\left\|\boldsymbol{\theta}-\Phi(\boldsymbol{\theta})\right\|^2\right) + \mathcal{O}\left(\rho\left\|\boldsymbol{\theta}-\Phi(\boldsymbol{\theta})\right\|\right) + \frac{\rho^2\beta_3}{2}.$$

$\square$

## D.2 Proof of Theorem 5.5

### D.2.1 Proof of Keeping Moving Near Minimizers for SAM

We first give the proof for SAM about "keeping moving near minimizers", which provides important insights into the proof for SAM-IRE.

Recalling (8), the update rule of average SAM is:

$$\boldsymbol{\theta}_{t+1} = \boldsymbol{\theta}_t - \eta\nabla\mathcal{L}\left(\boldsymbol{\theta}_t + \rho\frac{\boldsymbol{\xi}_t}{\|\boldsymbol{\xi}_t\|}\right), \quad \text{where } \boldsymbol{\xi}_t \sim \mathcal{N}(\boldsymbol{0}, I).$$

Let the $\rho$ in SAM satisfy:

$$\rho = \mathcal{O}(\sqrt{\eta}).$$

For simplicity, we denote

$$\boldsymbol{v}_t := \frac{\boldsymbol{\xi}_t}{\|\boldsymbol{\xi}_t\|}, \quad C_1 := \frac{4\beta_2^3}{\mu}, \quad C_2 := \beta_2^3.$$

Notice $\mathcal{L}(\boldsymbol{\theta}_{T_\mathrm{I}}) < C_1\eta\rho^2$, we have the following upper bound for the probability

$$\mathbb{P}\left(\exists\, t \in [T_\mathrm{I}, T_\mathrm{I}+T_\mathrm{II}], \mathcal{L}(\boldsymbol{\theta}_t) \geqslant 2C_1\eta\rho^2\right)$$
$$\leqslant \sum_{t=T_\mathrm{I}}^{T_\mathrm{I}+T_\mathrm{II}-1} \mathbb{P}\left(\mathcal{L}(\boldsymbol{\theta}_{t+1}) \geqslant 2C_1\eta\rho^2;\ \forall\, s \in [T_\mathrm{I}, t], \mathcal{L}(\boldsymbol{\theta}_s) < 2C_1\eta\rho^2\right).$$

For each term $t \in [T_\mathrm{I}, T_\mathrm{I}+T_\mathrm{II}-1]$, it can be bounded by:

$$\mathbb{P}\left(\mathcal{L}(\boldsymbol{\theta}_{t+1}) \geqslant 2C_1\eta\rho^2;\ \forall\, s \in [T_\mathrm{I}, t], \mathcal{L}(\boldsymbol{\theta}_s) < 2C_1\eta\rho^2\right)$$
$$\leqslant \mathbb{P}\left(\mathcal{L}(\boldsymbol{\theta}_{t+1}) \geqslant 2C_1\eta\rho^2; \mathcal{L}(\boldsymbol{\theta}_t) < C_1\eta\rho^2\right)$$
$$+ \sum_{s=T_\mathrm{I}}^{t-1} \mathbb{P}\left(\mathcal{L}(\boldsymbol{\theta}_{t+1}) \geqslant 2C_1\eta\rho^2; \mathcal{L}(\boldsymbol{\theta}_s) < C_1\eta\rho^2;\ \forall\, \tau \in [s+1,t], C_1\eta\rho^2 \leqslant \mathcal{L}(\boldsymbol{\theta}_\tau) < 2C_1\eta\rho^2\right)$$
$$\leqslant \mathbb{P}\left(\mathcal{L}(\boldsymbol{\theta}_{t+1}) \geqslant 2C_1\eta\rho^2\Big|\mathcal{L}(\boldsymbol{\theta}_t) < C_1\eta\rho^2\right)$$
$$+ \sum_{s=T_\mathrm{I}}^{t-1} \mathbb{P}\left(\mathcal{L}(\boldsymbol{\theta}_{t+1}) \geqslant 2C_1\eta\rho^2;\ \forall\, \tau \in [s+1,t], C_1\eta\rho^2 \leqslant \mathcal{L}(\boldsymbol{\theta}_\tau) < 2C_1\eta\rho^2\Big|\mathcal{L}(\boldsymbol{\theta}_s) < C_1\eta\rho^2\right).$$

For simplicity, we denote

$$\mathbb{P}_{t+1,t} := \mathbb{P}\left(\mathcal{L}(\boldsymbol{\theta}_{t+1}) \geqslant 2C_1\eta\rho^2 \big| \mathcal{L}(\boldsymbol{\theta}_t) < C_1\eta\rho^2\right),$$

$$\mathbb{P}_{t+1,s} := \mathbb{P}\left(\mathcal{L}(\boldsymbol{\theta}_{t+1}) \geqslant 2C_1\eta\rho^2; \ \forall \ \tau \in [s+1, t], C_1\eta\rho^2 \leqslant \mathcal{L}(\boldsymbol{\theta}_\tau) < 2C_1\eta\rho^2 \big| \mathcal{L}(\boldsymbol{\theta}_s) < C_1\eta\rho^2\right), \ s \in [T_{\mathrm{I}}, t-1].$$

- Step I. Bounding $\mathbb{P}_{t+1,t}$.

  From $\mathcal{L}(\boldsymbol{\theta}_t) \leqslant C_1\eta\rho^2$, we have $\|\boldsymbol{\theta}_t - \Phi(\boldsymbol{\theta}_t)\| \leqslant \sqrt{\frac{2}{\mu}\mathcal{L}(\boldsymbol{\theta}_t)} = \mathcal{O}(\frac{\sqrt{\eta}\rho}{\mu})$, thus

  $$\|\boldsymbol{\theta}_t + \rho\boldsymbol{v}_t - \Phi(\boldsymbol{\theta}_t)\| \leqslant \|\boldsymbol{\theta}_t - \Phi(\boldsymbol{\theta}_t)\| + \rho = \mathcal{O}(\sqrt{\eta}\rho) + \mathcal{O}(\rho) < R,$$

  which means $\boldsymbol{\theta}_t + \rho\boldsymbol{v}_t \in \mathbb{B}(K; R)$. Furthermore,

  $$\|\boldsymbol{\theta}_{t+1} - \Phi(\boldsymbol{\theta}_t)\| \leqslant \|\boldsymbol{\theta}_t - \Phi(\boldsymbol{\theta}_t)\| + \eta\|\nabla\mathcal{L}(\boldsymbol{\theta}_t + \rho\boldsymbol{v}_t)\|$$
  $$\leqslant \mathcal{O}(\sqrt{\eta}\rho) + \eta\|\nabla\mathcal{L}(\boldsymbol{\theta}_t + \rho\boldsymbol{v}_t)\| \leqslant \mathcal{O}(\sqrt{\eta}\rho) + \eta\|\nabla\mathcal{L}(\boldsymbol{\theta}_t)\| + \beta_2\eta\rho$$

  $$\leqslant \mathcal{O}(\sqrt{\eta}\rho) + \eta\sqrt{\frac{2\beta_2^2}{\mu}\mathcal{L}(\boldsymbol{\theta}_t)} + \beta_2\eta\rho \leqslant \mathcal{O}(\sqrt{\eta}\rho) + \mathcal{O}(\eta^{3/2}\rho) + \mathcal{O}(\eta\rho) \leqslant R,$$

  which implies $\boldsymbol{\theta}_{t+1} \in \mathbb{B}(K; R)$. Consequently, we have the following quadratic upper bound:

  $$\mathcal{L}(\boldsymbol{\theta}_{t+1}) = \mathcal{L}(\boldsymbol{\theta}_t - \eta\nabla\mathcal{L}(\boldsymbol{\theta}_t + \rho\boldsymbol{v}_t))$$

  $$\leqslant \mathcal{L}(\boldsymbol{\theta}_t) - \eta\langle\nabla\mathcal{L}(\boldsymbol{\theta}_t), \nabla\mathcal{L}(\boldsymbol{\theta}_t + \rho\boldsymbol{v}_t)\rangle + \frac{\eta^2\beta_2}{2}\|\nabla\mathcal{L}(\boldsymbol{\theta}_t + \rho\boldsymbol{v}_t)\|^2$$

  $$\leqslant \mathcal{L}(\boldsymbol{\theta}_t) - \eta\|\nabla\mathcal{L}(\boldsymbol{\theta}_t)\|^2 - \eta\rho\langle\nabla\mathcal{L}(\boldsymbol{\theta}_t), \nabla^2\mathcal{L}(\boldsymbol{\theta}_t)\boldsymbol{v}_t\rangle + \frac{\eta\rho^2\beta_3}{2}\|\nabla\mathcal{L}(\boldsymbol{\theta})\| + \frac{\eta^2\beta_2}{2}(\|\nabla\mathcal{L}(\boldsymbol{\theta}_t)\| + \rho\beta_2)^2$$

  $$\leqslant \mathcal{L}(\boldsymbol{\theta}_t) - \eta\|\nabla\mathcal{L}(\boldsymbol{\theta}_t)\|^2 - \eta\rho\langle\nabla\mathcal{L}(\boldsymbol{\theta}_t), \nabla^2\mathcal{L}(\boldsymbol{\theta}_t)\boldsymbol{v}_t\rangle + \frac{\eta\rho^2\beta_3}{2}\|\nabla\mathcal{L}(\boldsymbol{\theta})\| + \eta^2\beta_2\left(\|\nabla\mathcal{L}(\boldsymbol{\theta}_t)\|^2 + \rho^2\beta_2^2\right)$$

  $$\leqslant \mathcal{L}(\boldsymbol{\theta}_t) + \eta\rho\|\nabla^2\mathcal{L}(\boldsymbol{\theta}_t)\nabla\mathcal{L}(\boldsymbol{\theta}_t)\| + \frac{\eta\rho^2\beta_3}{2}\|\nabla\mathcal{L}(\boldsymbol{\theta}_t)\| + \eta^2\beta_2\rho^2\beta_2^2$$

  $$\leqslant \mathcal{L}(\boldsymbol{\theta}_t) + \left(\eta\rho\beta_2 + \frac{\eta\rho^2\beta_3}{2}\right)\|\nabla\mathcal{L}(\boldsymbol{\theta}_t)\| + \mathcal{O}(\eta^2\rho^2)$$

  $$\leqslant \mathcal{L}(\boldsymbol{\theta}_t) + \left(\eta\rho\beta_2 + \frac{\eta\rho^2\beta_3}{2}\right)\sqrt{\frac{2\beta_2^2}{\mu}}\sqrt{\mathcal{L}(\boldsymbol{\theta}_t)} + \mathcal{O}(\eta^2\rho^2)$$

  $$\leqslant C_1\eta\rho^2 + \mathcal{O}(\eta^{3/2}\rho^2) + \mathcal{O}(\eta^2\rho^2) < 3C_1\eta\rho^2/2.$$

  Thus, we obtain

  $$\mathbb{P}_{t+1,t} = \mathbb{P}\left(\mathcal{L}(\boldsymbol{\theta}_{t+1}) \geqslant 2C_1\eta\rho^2 \big| \mathcal{L}(\boldsymbol{\theta}_t) < C_1\eta\rho^2\right) = 0.$$

- Step II. Bounding $\mathbb{P}_{t+1,s}$ for $s \in [T_{\mathrm{I}}, t-1]$.

  We prove this step under the condition $\mathcal{L}(\boldsymbol{\theta}_s) < C_1\eta\rho^2$. Define a process $\{X_\tau\}_{\tau=s}^{t+1}$: $X_{s+1} = \mathcal{L}(\boldsymbol{\theta}_{s+1})$,

  $$X_{\tau+1} = \begin{cases} \mathcal{L}(\boldsymbol{\theta}_{\tau+1}), & \text{if } C_1\eta\rho^2 \leqslant X_\tau = \mathcal{L}(\boldsymbol{\theta}_\tau) \leqslant 2C_1\eta\rho^2 \\ X_\tau - C_2\eta^2\rho^2, & \text{else} \end{cases}.$$

  It is clear that

  $$\mathbb{P}_{t+1,s} \leqslant \mathbb{P}\left(X_{t+1} \geqslant 2C_1\eta\rho^2\right).$$

  Then our key step is to prove the following two claims about the process $\{X_\tau\}$.

  - Claim I. $X_\tau - C_2\tau\eta\rho^2$ is a super-martingale. From the definition of $X_\tau$, we only need to prove that if $C_1\eta\rho^2 \leqslant \mathcal{L}(\boldsymbol{\theta}_\tau) \leqslant 2C_1\eta\rho^2$, then $\mathbb{E}\left[\mathcal{L}(\boldsymbol{\theta}_{\tau+1})\right] \leqslant \mathcal{L}(\boldsymbol{\theta}_\tau) - C_2\eta^2\rho^2$.

    If $C_1\eta\rho^2 \leqslant \mathcal{L}(\boldsymbol{\theta}_\tau) \leqslant 2C_1\eta\rho^2$, similar to Step I, it is clear that $\boldsymbol{\theta}_{\tau+1} \in \mathbb{B}(K; R)$. Applying the quadratic upper bound, it holds that:

    $$\mathcal{L}(\boldsymbol{\theta}_{\tau+1}) = \mathcal{L}(\boldsymbol{\theta}_\tau - \eta\nabla\mathcal{L}(\boldsymbol{\theta}_\tau + \rho\boldsymbol{v}_\tau))$$

$$\leqslant \mathcal{L}(\boldsymbol{\theta}_\tau) - \eta \left\langle \nabla \mathcal{L}(\boldsymbol{\theta}_\tau), \nabla \mathcal{L}\left(\boldsymbol{\theta}_\tau + \rho \boldsymbol{v}_\tau\right)\right\rangle + \frac{\eta^2 \beta_2}{2}\left\| \nabla \mathcal{L}\left(\boldsymbol{\theta}_\tau + \rho \boldsymbol{v}_\tau\right)\right\|^2$$

$$\leqslant \mathcal{L}(\boldsymbol{\theta}_\tau) - \eta \left\| \nabla \mathcal{L}(\boldsymbol{\theta}_\tau)\right\|^2 - \eta\rho \left\langle \nabla \mathcal{L}(\boldsymbol{\theta}_\tau), \nabla^2 \mathcal{L}(\boldsymbol{\theta}_\tau)\boldsymbol{v}_t\right\rangle$$
$$+ \frac{\eta\rho^2 \beta_3}{2}\left\| \nabla \mathcal{L}(\boldsymbol{\theta}_\tau)\right\| + \eta^2 \beta_2 \left(\left\| \nabla \mathcal{L}(\boldsymbol{\theta}_\tau)\right\|^2 + \rho^2 \beta_2^2\right).$$

Taking the expectation, we have:

$$\mathbb{E}\left[\mathcal{L}(\boldsymbol{\theta}_{\tau+1})\right] \leqslant \mathcal{L}(\boldsymbol{\theta}_\tau) - \eta \left\| \nabla \mathcal{L}(\boldsymbol{\theta}_\tau)\right\|^2 + \frac{\eta\rho^2 \beta_3}{2}\left\| \nabla \mathcal{L}(\boldsymbol{\theta}_\tau)\right\| + \eta^2 \beta_2 \left(\left\| \nabla \mathcal{L}(\boldsymbol{\theta}_\tau)\right\|^2 + \rho^2 \beta_2^2\right)$$

$$\leqslant \mathcal{L}(\boldsymbol{\theta}_\tau) - \frac{3}{4}\eta \left\| \nabla \mathcal{L}(\boldsymbol{\theta}_\tau)\right\|^2 + \frac{\eta\rho^2 \beta_3}{2}\left\| \nabla \mathcal{L}(\boldsymbol{\theta}_\tau)\right\| + \eta^2 \rho^2 \beta_2^3$$

$$\leqslant \mathcal{L}(\boldsymbol{\theta}_\tau) - \eta \left\| \nabla \mathcal{L}(\boldsymbol{\theta}_\tau)\right\| \left(\frac{3}{4}\left\| \nabla \mathcal{L}(\boldsymbol{\theta}_\tau)\right\| - \frac{\rho^2 \beta_3}{2}\right) + \eta^2 \rho^2 \beta_2^3.$$

From $\mathcal{L}(\boldsymbol{\theta}_\tau) \geqslant C_1 \eta \rho^2$ and $\rho = \mathcal{O}(\sqrt{\eta})$, it holds that

$$\left\| \nabla \mathcal{L}(\boldsymbol{\theta}_\tau)\right\| \geqslant \sqrt{2\mu\mathcal{L}(\boldsymbol{\theta}_t)} \geqslant \sqrt{2C_1\mu}\sqrt{\eta}\rho \geqslant 4\beta_3 \rho^2.$$

Therefore, we have:

$$\mathbb{E}\left[\mathcal{L}(\boldsymbol{\theta}_{\tau+1})\right] \leqslant \mathcal{L}(\boldsymbol{\theta}_\tau) - \frac{5}{8}\eta \left\| \nabla \mathcal{L}(\boldsymbol{\theta}_\tau)\right\|^2 + \eta^2 \rho^2 \beta_2^3$$

$$\leqslant \mathcal{L}(\boldsymbol{\theta}_\tau) - \frac{10}{8}\eta\mu\mathcal{L}(\boldsymbol{\theta}_\tau) + \eta^2 \rho^2 \beta_2^3 \leqslant \mathcal{L}(\boldsymbol{\theta}_\tau) - \frac{10}{8}C_1\mu\eta^2 \rho^2 + \eta^2 \rho^2 \beta_2^3$$

$$\leqslant \mathcal{L}(\boldsymbol{\theta}_\tau) - \beta_2^3 \eta^2 \rho^2 = \mathcal{L}(\boldsymbol{\theta}_\tau) - C_2 \eta^2 \rho^2.$$

- Claim II. $X_{\tau+1} - X_\tau + C_2 \eta^2 \rho^2$ is $\mathcal{O}(\eta^2 \rho^2 + \eta^{3/2}\rho^2/p^{1/2})$-sub-Gaussian. From the definition of $X_\tau$, we only need to prove for the case $C_1 \eta \rho^2 \leqslant \mathcal{L}(\boldsymbol{\theta}_\tau) \leqslant 2C_1 \eta \rho^2$.
If $C_1 \eta \rho^2 \leqslant \mathcal{L}(\boldsymbol{\theta}_\tau) \leqslant 2C_1 \eta \rho^2$, then $X_\tau = \mathcal{L}(\boldsymbol{\theta}_\tau)$ and $X_{\tau+1} = \mathcal{L}(\boldsymbol{\theta}_{\tau+1})$. Similar to Step I, it is clear that $\boldsymbol{\theta}_{\tau+1} \in \mathbb{B}(K;R)$.
Applying the smoothness, we have:

$$\mathcal{L}(\boldsymbol{\theta}_{\tau+1}) = \mathcal{L}\left(\boldsymbol{\theta}_\tau - \eta\nabla \mathcal{L}\left(\boldsymbol{\theta}_\tau + \rho \boldsymbol{v}_\tau\right)\right)$$

$$= \mathcal{L}(\boldsymbol{\theta}_\tau) - \eta \left\langle \nabla \mathcal{L}(\boldsymbol{\theta}_\tau), \nabla \mathcal{L}\left(\boldsymbol{\theta}_\tau + \rho \boldsymbol{v}_\tau\right)\right\rangle + \mathcal{O}\left(\eta^2 \left\| \nabla \mathcal{L}\left(\boldsymbol{\theta}_\tau + \rho \boldsymbol{v}_\tau\right)\right\|^2\right)$$

$$= \mathcal{L}(\boldsymbol{\theta}_\tau) - \eta \left\| \nabla \mathcal{L}(\boldsymbol{\theta}_\tau)\right\|^2 - \eta\rho \left\langle \nabla \mathcal{L}(\boldsymbol{\theta}_\tau), \nabla^2 \mathcal{L}(\boldsymbol{\theta}_\tau)\boldsymbol{v}_t\right\rangle$$
$$+ \mathcal{O}\left(\eta\rho^2 \left\| \nabla \mathcal{L}(\boldsymbol{\theta}_\tau)\right\|\right) + \mathcal{O}\left(\eta^2 \left\| \nabla \mathcal{L}(\boldsymbol{\theta}_\tau)\right\|^2 + \eta^2 \rho^2\right)$$

$$= \mathcal{L}(\boldsymbol{\theta}_\tau) + \mathcal{O}\left(\eta^2 \rho^2\right) - \eta\rho \left\langle \nabla \mathcal{L}(\boldsymbol{\theta}_\tau), \nabla^2 \mathcal{L}(\boldsymbol{\theta}_\tau)\boldsymbol{v}_t\right\rangle + \mathcal{O}\left(\eta\rho^2 \sqrt{\eta}\rho\right) + \mathcal{O}\left(\eta^3 \rho^2 + \eta^2 \rho^2\right)$$

$$= \mathcal{L}(\boldsymbol{\theta}_\tau) - \eta\rho \left\langle \nabla \mathcal{L}(\boldsymbol{\theta}_\tau), \nabla^2 \mathcal{L}(\boldsymbol{\theta}_\tau)\boldsymbol{v}_t\right\rangle + \mathcal{O}(\eta^2 \rho^2),$$

which implies:

$$\left\| X_{\tau+1} - X_\tau - C_2 \eta^2 \rho^2\right\|_{\psi_2} \leqslant \left\| \mathcal{L}(\boldsymbol{\theta}_{\tau+1}) - \mathcal{L}(\boldsymbol{\theta}_\tau)\right\|_{\psi_2} + \left\| C_2 \eta^2 \rho^2\right\|_{\psi_2}$$

$$\leqslant \left\| \eta\rho \left\langle \nabla \mathcal{L}(\boldsymbol{\theta}_\tau), \nabla^2 \mathcal{L}(\boldsymbol{\theta}_\tau)\boldsymbol{v}_t\right\rangle\right\|_{\psi_2} + \mathcal{O}(\eta^2 \rho^2)$$

$$\leqslant \eta\rho \left\| \nabla^2 \mathcal{L}(\boldsymbol{\theta}_\tau)\nabla \mathcal{L}(\boldsymbol{\theta}_\tau)\right\| \left\| \left\langle \frac{\nabla^2 \mathcal{L}(\boldsymbol{\theta}_\tau)\nabla \mathcal{L}(\boldsymbol{\theta}_\tau)}{\|\nabla^2 \mathcal{L}(\boldsymbol{\theta}_\tau)\nabla \mathcal{L}(\boldsymbol{\theta}_\tau)\|}, \boldsymbol{v}_t\right\rangle\right\|_{\psi_2} + \mathcal{O}(\eta^2 \rho^2)$$

$$\leqslant \mathcal{O}\left(\eta^{3/2}\rho^2 \left\| \left\langle \frac{\nabla^2 \mathcal{L}(\boldsymbol{\theta}_\tau)\nabla \mathcal{L}(\boldsymbol{\theta}_\tau)}{\|\nabla^2 \mathcal{L}(\boldsymbol{\theta}_\tau)\nabla \mathcal{L}(\boldsymbol{\theta}_\tau)\|}, \boldsymbol{v}_t\right\rangle\right\|_{\psi_2}\right) + \mathcal{O}(\eta^2 \rho^2)$$

$$\overset{\text{Lemma F.5}}{\leqslant} \mathcal{O}\left(\eta^{3/2}\rho^2/\sqrt{p}\right) + \mathcal{O}(\eta^2 \rho^2).$$

With the preparation of Claim I and Claim II, we can use the Azuma-Hoeffding inequality (Lemma F.4 (ii)): for any $Q > 0$, it holds that

$$\mathbb{P}\left(X_{t+1} - X_{s+1} + (t-s)C_2 \eta^2 \rho^2 > Q\right) \leqslant 2\exp\left(-\frac{Q^2}{2(t-s)\left(\mathcal{O}(\eta^{3/2}\rho^2/p^{1/2} + \eta^2 \rho^2)\right)^2}\right).$$

As proved in Claim I, $X_{s+1} = \mathcal{L}(\boldsymbol{\theta}_{s+1}) \leqslant \frac{3}{2}C_1\eta\rho^2$ due to $\mathcal{L}(\boldsymbol{\theta}_s) \leqslant C_1\eta\rho^2$. Therefore, by choosing $Q = (t-s)C_2\eta^2\rho^2 - \frac{3}{2}C_1\eta\rho^2 + 2C_1\eta\rho^2 = (t-s)C_2\eta^2\rho^2 + \frac{1}{2}C_1\eta\rho^2$, we have

$$\mathbb{P}_{t+1,s} \leqslant \mathbb{P}\left(X_{t+1} \geqslant 2C_1\eta\rho^2\right)$$

$$\leqslant \mathbb{P}\left(X_{t+1} - X_{s+1} + (t-s)C_2\eta^2\rho^2 > (t-s)C_2\eta^2\rho^2 + \frac{1}{2}C_1\eta\rho^2\right)$$

$$\leqslant 2\exp\left(-\frac{\left((t-s)C_2\eta^2\rho^2 + \frac{1}{2}C_1\eta\rho^2\right)^2}{2(t-s)\left(\mathcal{O}(\eta^{3/2}\rho^2/p^{1/2} + \eta^2\rho^2)\right)^2}\right)$$

$$\leqslant 2\exp\left(-\frac{4(t-s)C_2\eta^2\rho^2 \cdot \frac{1}{2}C_1\eta\rho^2}{4(t-s)\left(\mathcal{O}(\eta^3\rho^4/p + \eta^4\rho^4)\right)}\right) \leqslant 2\exp\left(-\Omega\left(\frac{1}{\eta+p^{-1}}\right)\right).$$

Therefore, we obtain the union bound:

$$\mathbb{P}\left(\exists\, t \in [T_\mathrm{I}, T_\mathrm{I} + T_\mathrm{II}], \mathcal{L}(\boldsymbol{\theta}_t) \geqslant 2C_1\eta\rho^2\right) \leqslant \sum_{t=T_I}^{T_\mathrm{I}+T_\mathrm{II}-1}\left(\mathbb{P}_{t+1,t} + \sum_{s=T_\mathrm{I}}^{t-1}\mathbb{P}_{t+1,s}\right)$$

$$\leqslant \sum_{t=T_I}^{T_\mathrm{I}+T_\mathrm{II}-1}\sum_{s=T_\mathrm{I}}^{t-1}\mathbb{P}_{t+1,s} \leqslant T_\mathrm{II}^2\exp\left(-\Omega\left(\frac{1}{\eta+p^{-1}}\right)\right).$$

Hence, with probability at least $1 - T_\mathrm{II}^2\exp\left(-\Omega\left(\frac{1}{\eta+p^{-1}}\right)\right)$, for any $t \in [T_\mathrm{I}, T_\mathrm{I} + T_\mathrm{II}]$,

$$\|\boldsymbol{\theta}_t - \Phi(\boldsymbol{\theta}_t)\| \leqslant \sqrt{\frac{2}{\mu}\mathcal{L}(\boldsymbol{\theta}_t)} \leqslant 2\sqrt{\frac{C_1}{\mu}}\sqrt{\eta}\rho = \frac{4\beta_2^{3/2}}{\mu}\sqrt{\eta}\rho = \mathcal{O}(\sqrt{\eta}\rho).$$

### D.2.2 Proof of Moving Near Minimizers for SAM-IRE

We prove "keeping moving near minimizers" for SAM-IRE. The proof outline for SAM-IRE is the same as SAM. However, the key non-trivial difference is that the IRE term will hardly cause loss instability since IRE only perturbs the parameters in the flat directions.

Under the conditions in Theorem 5.5, the update rule of IRE on average SAM is:

$$\boldsymbol{\theta}_{t+1} = \boldsymbol{\theta}_t - \eta\nabla\mathcal{L}\left(\boldsymbol{\theta}_t + \rho\frac{\boldsymbol{\xi}_t}{\|\boldsymbol{\xi}_t\|}\right) - \eta\kappa P_{m+1:p}(\nabla^2\mathcal{L}(\boldsymbol{\theta}_t))\nabla\mathcal{L}\left(\boldsymbol{\theta}_t + \rho\frac{\boldsymbol{\xi}_t}{\|\boldsymbol{\xi}_t\|}\right), \quad \text{where } \boldsymbol{\xi}_t \sim \mathcal{N}(\mathbf{0}, I).$$

Let the $\rho$ in SAM and the $\kappa$ in IRE satisfy:

$$\rho = \mathcal{O}(\sqrt{\eta}), \quad \kappa \leqslant \frac{1}{\rho}. \tag{15}$$

For simplicity, we denote

$$\boldsymbol{v}_t := \frac{\boldsymbol{\xi}_t}{\|\boldsymbol{\xi}_t\|}, \quad P(\boldsymbol{\theta}_t) := P_{m+1:p}(\nabla^2\mathcal{L}(\boldsymbol{\theta}_t)), \quad C_1 = \frac{4\beta_2^3}{\mu} \vee \frac{4\beta_2\beta_3^2}{\mu}, \quad C_2 = \beta_2^3$$

Following the proof for SAM, we denote

$$\mathbb{P}_{t+1,t} := \mathbb{P}\left(\mathcal{L}(\boldsymbol{\theta}_{t+1}) \geqslant 2C_1\eta\rho^2 \Big| \mathcal{L}(\boldsymbol{\theta}_t) < C_1\eta\rho^2\right),$$

$$\mathbb{P}_{t+1,s} := \mathbb{P}\Big(\mathcal{L}(\boldsymbol{\theta}_{t+1}) \geqslant 2C_1\eta\rho^2;$$

$$\forall\, \tau \in [s+1, t], C_1\eta\rho^2 \leqslant \mathcal{L}(\boldsymbol{\theta}_\tau) < 2C_1\eta\rho^2 \Big| \mathcal{L}(\boldsymbol{\theta}_s) < C_1\eta\rho^2\Big), \quad s \in [T_\mathrm{I}, t-1].$$

and it holds that

$$\mathbb{P}\left(\exists\, t \in [T_\mathrm{I}, T_\mathrm{I} + T_\mathrm{II}], \mathcal{L}(\boldsymbol{\theta}_t) \geqslant 2C_1\eta\rho^2\right) \leqslant \sum_{t=T_\mathrm{I}}^{T_\mathrm{I}+T_\mathrm{II}-1}\left(\mathbb{P}_{t+1,t} + \sum_{s=T_\mathrm{I}}^{t-1}\mathbb{P}_{t+1,s}\right).$$

- Step I. Bounding $\mathbb{P}_{t+1,t}$.

  From $\mathcal{L}(\boldsymbol{\theta}_t) \leqslant C_1\eta\rho^2$, we have $\|\boldsymbol{\theta}_t - \Phi(\boldsymbol{\theta}_t)\| \leqslant \sqrt{\frac{2}{\mu}\mathcal{L}(\boldsymbol{\theta}_t)} = \mathcal{O}(\sqrt{\eta}\rho)$, thus

  $$\|\boldsymbol{\theta}_t + \rho\boldsymbol{v}_t - \Phi(\boldsymbol{\theta}_t)\| \leqslant \|\boldsymbol{\theta}_t - \Phi(\boldsymbol{\theta}_t)\| + \rho = \mathcal{O}(\sqrt{\eta}\rho) + \mathcal{O}(\rho) < R,$$

  which means $\boldsymbol{\theta}_t + \rho\boldsymbol{v}_t \in \mathbb{B}(K; R)$. Furthermore,

  $$\begin{aligned}
  &\|\boldsymbol{\theta}_{t+1} - \Phi(\boldsymbol{\theta}_t)\| \\
  &\leqslant \|\boldsymbol{\theta}_t - \Phi(\boldsymbol{\theta}_t)\| + \eta\|\nabla\mathcal{L}(\boldsymbol{\theta}_t + \rho\boldsymbol{v}_t)\| + \eta\kappa\|P(\boldsymbol{\theta}_t)\nabla\mathcal{L}(\boldsymbol{\theta}_t + \rho\boldsymbol{v}_t)\| \\
  &\overset{\text{Lemma D.5}}{\leqslant} \mathcal{O}(\sqrt{\eta}\rho) + \eta\|\nabla\mathcal{L}(\boldsymbol{\theta}_t)\| + \beta_2\eta\rho + \mathcal{O}(\eta\kappa\rho^2) \\
  &\leqslant \mathcal{O}(\sqrt{\eta}\rho) + \mathcal{O}(\eta^{3/2}\rho) + \mathcal{O}(\eta\rho) + \mathcal{O}(\eta\rho) \\
  &\leqslant \mathcal{O}(\sqrt{\eta}\rho) + \mathcal{O}(\eta^{3/2}\rho) + \mathcal{O}(\eta\rho) \leqslant R,
  \end{aligned}$$

  which implies $\boldsymbol{\theta}_{t+1} \in \mathbb{B}(K; R)$. Consequently, we have the following quadratic upper bound:

  $$\begin{aligned}
  \mathcal{L}(\boldsymbol{\theta}_{t+1}) &= \mathcal{L}(\boldsymbol{\theta}_t - \eta\nabla\mathcal{L}(\boldsymbol{\theta}_t + \rho\boldsymbol{v}_t) - \eta\kappa P(\boldsymbol{\theta}_t)\nabla\mathcal{L}(\boldsymbol{\theta}_t + \rho\boldsymbol{v}_t)) \\
  &\leqslant \mathcal{L}(\boldsymbol{\theta}_t) - \eta\langle\nabla\mathcal{L}(\boldsymbol{\theta}_t), \nabla\mathcal{L}(\boldsymbol{\theta}_t + \rho\boldsymbol{v}_t) + \kappa P(\boldsymbol{\theta}_t)\nabla\mathcal{L}(\boldsymbol{\theta}_t + \rho\boldsymbol{v}_t)\rangle \\
  &\quad + \frac{\eta^2\beta_2}{2}\|\nabla\mathcal{L}(\boldsymbol{\theta}_t + \rho\boldsymbol{v}_t) + \kappa P(\boldsymbol{\theta}_t)\nabla\mathcal{L}(\boldsymbol{\theta}_t + \rho\boldsymbol{v}_t)\|^2 \\
  &\leqslant \mathcal{L}(\boldsymbol{\theta}_t) - \eta\langle\nabla\mathcal{L}(\boldsymbol{\theta}_t), \nabla\mathcal{L}(\boldsymbol{\theta}_t + \rho\boldsymbol{v}_t)\rangle - \kappa\eta\langle\nabla\mathcal{L}(\boldsymbol{\theta}_t), P(\boldsymbol{\theta}_t)\nabla\mathcal{L}(\boldsymbol{\theta}_t + \rho\boldsymbol{v}_t)\rangle \\
  &\quad + \eta^2\beta_2\left(\|\nabla\mathcal{L}(\boldsymbol{\theta}_t + \rho\boldsymbol{v}_t)\|^2 + \kappa^2\|P(\boldsymbol{\theta}_t)\nabla\mathcal{L}(\boldsymbol{\theta}_t + \rho\boldsymbol{v}_t)\|^2\right) \\
  &\leqslant \mathcal{L}(\boldsymbol{\theta}_t) - \eta\|\nabla\mathcal{L}(\boldsymbol{\theta}_t)\|^2 + \eta\rho\beta_2\|\nabla\mathcal{L}(\boldsymbol{\theta}_t)\| - \kappa\eta\langle\nabla\mathcal{L}(\boldsymbol{\theta}_t), P(\boldsymbol{\theta}_t)\nabla\mathcal{L}(\boldsymbol{\theta}_t)\rangle \\
  &\quad - \kappa\eta\rho\langle\nabla\mathcal{L}(\boldsymbol{\theta}_t), P(\boldsymbol{\theta}_t)\nabla^2\mathcal{L}(\boldsymbol{\theta}_t)\boldsymbol{v}_t\rangle + \kappa\eta\frac{\beta_3\rho^2}{2}\|P(\boldsymbol{\theta}_t)\nabla\mathcal{L}(\boldsymbol{\theta}_t)\| \\
  &\quad + \eta^2\beta_2\left((\|\nabla\mathcal{L}(\boldsymbol{\theta}_t)\| + \rho\beta_2)^2 + \kappa^2\|P(\boldsymbol{\theta}_t)\nabla\mathcal{L}(\boldsymbol{\theta}_t + \rho\boldsymbol{v}_t)\|^2\right) \\
  &\leqslant \mathcal{L}(\boldsymbol{\theta}_t) + \eta\rho\beta_2\|\nabla\mathcal{L}(\boldsymbol{\theta}_t)\| + \kappa\eta\rho\|P(\boldsymbol{\theta}_t)\nabla^2\mathcal{L}(\boldsymbol{\theta}_t)\nabla\mathcal{L}(\boldsymbol{\theta}_t)\| \\
  &\quad + \frac{\kappa\eta\rho^2\beta_3}{2}\|P(\boldsymbol{\theta}_t)\nabla\mathcal{L}(\boldsymbol{\theta}_t)\| + \eta^2\beta_2\left((\|\nabla\mathcal{L}(\boldsymbol{\theta}_t)\| + \rho\beta_2)^2 + \kappa^2\|P(\boldsymbol{\theta}_t)\nabla\mathcal{L}(\boldsymbol{\theta}_t + \rho\boldsymbol{v}_t)\|^2\right) \\
  &\overset{\text{Lemma D.5}}{\leqslant} \mathcal{L}(\boldsymbol{\theta}_t) + \mathcal{O}(\eta\rho\|\nabla\mathcal{L}(\boldsymbol{\theta}_t)\|) + \mathcal{O}(\kappa\eta\rho\|\boldsymbol{\theta}_t - \Phi(\boldsymbol{\theta}_t)\|\|\nabla\mathcal{L}(\boldsymbol{\theta}_t)\|) \\
  &\quad + \mathcal{O}\left(\kappa\eta\rho^2\|\boldsymbol{\theta}_t - \Phi(\boldsymbol{\theta}_t)\|^2\right) + \mathcal{O}\left(\eta^2\rho^2\right) + \mathcal{O}\left(\eta^2\kappa^2\rho^4\right) \\
  &\leqslant C_1\eta\rho^2 + o(\eta\rho^2) \leqslant \frac{3}{2}C_1\eta\rho^2.
  \end{aligned}$$

  Thus, we obtain

  $$\mathbb{P}_{t+1,t} = \mathbb{P}\left(\mathcal{L}(\boldsymbol{\theta}_{t+1}) \geqslant 2C_1\eta\rho^2\,\Big|\,\mathcal{L}(\boldsymbol{\theta}_t) < C_1\eta\rho^2\right) = 0.$$

- Step II. Bounding $\mathbb{P}_{t+1,s}$ for $s \in [T_{\mathrm{I}}, t-1]$.

  We prove this step under the condition $\mathcal{L}(\boldsymbol{\theta}_s) < C_1\eta\rho^2$. Define a process $\{X_\tau\}_{\tau=s}^{t+1}$: $X_{s+1} = \mathcal{L}(\boldsymbol{\theta}_{s+1})$,

  $$X_{\tau+1} = \begin{cases} \mathcal{L}(\boldsymbol{\theta}_{\tau+1}), & \text{if } C_1\eta\rho^2 \leqslant X_\tau = \mathcal{L}(\boldsymbol{\theta}_\tau) \leqslant 2C_1\eta\rho^2 \\ X_\tau - C_2\eta^2\rho^2, & \text{else} \end{cases}.$$

  It is clear that

  $$\mathbb{P}_{t+1,s} \leqslant \mathbb{P}\left(X_{t+1} \geqslant 2C_1\eta\rho^2\right).$$

  Then our key step is to prove the following two claims about the process $\{X_\tau\}$.

  - Claim I. $X_\tau - C_2\tau\eta\rho^2$ is a super-martingale. From the definition of $X_\tau$, we only need to prove that if $C_1\eta\rho^2 \leqslant \mathcal{L}(\boldsymbol{\theta}_\tau) \leqslant 2C_1\eta\rho^2$, then $\mathbb{E}[\mathcal{L}(\boldsymbol{\theta}_{\tau+1})] \leqslant \mathcal{L}(\boldsymbol{\theta}_\tau) - C_2\eta^2\rho^2$.

If $C_1\eta\rho^2 \leqslant \mathcal{L}(\boldsymbol{\theta}_\tau) \leqslant 2C_1\eta\rho^2$, similar to Step I, it is clear that $\boldsymbol{\theta}_{\tau+1} \in \mathbb{B}(K; R)$. Applying the quadratic upper bound, it holds that:

$$\mathcal{L}(\boldsymbol{\theta}_{\tau+1}) = \mathcal{L}\left(\boldsymbol{\theta}_\tau - \eta\nabla\mathcal{L}\left(\boldsymbol{\theta}_\tau + \rho\boldsymbol{v}_\tau\right) - \eta\kappa P(\boldsymbol{\theta}_\tau)\nabla\mathcal{L}\left(\boldsymbol{\theta}_\tau + \rho\boldsymbol{v}_\tau\right)\right)$$

$$\leqslant \mathcal{L}(\boldsymbol{\theta}_\tau) - \eta\left\langle\nabla\mathcal{L}(\boldsymbol{\theta}_\tau), \nabla\mathcal{L}\left(\boldsymbol{\theta}_\tau + \rho\boldsymbol{v}_\tau\right) + \kappa P(\boldsymbol{\theta}_t)\nabla\mathcal{L}\left(\boldsymbol{\theta}_\tau + \rho\boldsymbol{v}_\tau\right)\right\rangle$$

$$+ \frac{\eta^2\beta_2}{2}\left\|\nabla\mathcal{L}\left(\boldsymbol{\theta}_\tau + \rho\boldsymbol{v}_\tau\right) + \kappa P(\boldsymbol{\theta}_\tau)\nabla\mathcal{L}\left(\boldsymbol{\theta}_\tau + \rho\boldsymbol{v}_\tau\right)\right\|^2$$

$$\leqslant \mathcal{L}(\boldsymbol{\theta}_\tau) - \eta\left\langle\nabla\mathcal{L}(\boldsymbol{\theta}_\tau), \nabla\mathcal{L}\left(\boldsymbol{\theta}_\tau + \rho\boldsymbol{v}_\tau\right)\right\rangle - \kappa\eta\left\langle\nabla\mathcal{L}(\boldsymbol{\theta}_\tau), P(\boldsymbol{\theta}_t)\nabla\mathcal{L}\left(\boldsymbol{\theta}_\tau + \rho\boldsymbol{v}_\tau\right)\right\rangle$$

$$+ \eta^2\beta_2\left(\left\|\nabla\mathcal{L}\left(\boldsymbol{\theta}_\tau + \rho\boldsymbol{v}_\tau\right)\right\|^2 + \kappa^2\left\|P(\boldsymbol{\theta}_\tau)\nabla\mathcal{L}\left(\boldsymbol{\theta}_\tau + \rho\boldsymbol{v}_\tau\right)\right\|^2\right)$$

$$\leqslant \mathcal{L}(\boldsymbol{\theta}_\tau) - \eta\left\|\nabla\mathcal{L}(\boldsymbol{\theta}_\tau)\right\|^2 - \eta\rho\left\langle\nabla\mathcal{L}(\boldsymbol{\theta}_\tau), \nabla^2\mathcal{L}(\boldsymbol{\theta}_t)\boldsymbol{v}_\tau\right\rangle + \frac{\eta\rho^2\beta_3}{2}\left\|\nabla\mathcal{L}(\boldsymbol{\theta}_\tau)\right\|$$

$$- \kappa\eta\left\langle\nabla\mathcal{L}(\boldsymbol{\theta}_\tau), P(\boldsymbol{\theta}_\tau)\nabla\mathcal{L}(\boldsymbol{\theta}_\tau)\right\rangle - \kappa\eta\rho\left\langle\nabla\mathcal{L}(\boldsymbol{\theta}_\tau), P(\boldsymbol{\theta}_\tau)\nabla^2\mathcal{L}(\boldsymbol{\theta}_\tau)\boldsymbol{v}_\tau\right\rangle + \kappa\eta\frac{\beta_3\rho^2}{2}\left\|P(\boldsymbol{\theta}_\tau)\nabla\mathcal{L}(\boldsymbol{\theta}_\tau)\right\|$$

$$+ \eta^2\beta_2\left(\left(\left\|\nabla\mathcal{L}(\boldsymbol{\theta}_\tau)\right\| + \rho\beta_2\right)^2 + \kappa^2\left\|P(\boldsymbol{\theta}_\tau)\nabla\mathcal{L}\left(\boldsymbol{\theta}_\tau + \rho\boldsymbol{v}_\tau\right)\right\|^2\right).$$

Taking the expectation, we have:

$$\mathbb{E}[\mathcal{L}(\boldsymbol{\theta}_{\tau+1})]$$

$$\leqslant \mathcal{L}(\boldsymbol{\theta}_\tau) - \eta\left\|\nabla\mathcal{L}(\boldsymbol{\theta}_\tau)\right\|^2 + \frac{\eta\rho^2\beta_3}{2}\left\|\nabla\mathcal{L}(\boldsymbol{\theta}_\tau)\right\| - \kappa\eta\left\langle\nabla\mathcal{L}(\boldsymbol{\theta}_\tau), P(\boldsymbol{\theta}_\tau)\nabla\mathcal{L}(\boldsymbol{\theta}_\tau)\right\rangle$$

$$+ \kappa\eta\frac{\beta_3\rho^2}{2}\left\|P(\boldsymbol{\theta}_\tau)\nabla\mathcal{L}(\boldsymbol{\theta}_\tau)\right\| + \eta^2\beta_2\left(\left(\left\|\nabla\mathcal{L}(\boldsymbol{\theta}_\tau)\right\| + \rho\beta_2\right)^2 + \kappa^2\left\|P(\boldsymbol{\theta}_\tau)\nabla\mathcal{L}\left(\boldsymbol{\theta}_\tau + \rho\boldsymbol{v}_\tau\right)\right\|^2\right)$$

$$\leqslant \mathcal{L}(\boldsymbol{\theta}_\tau) - \eta\left\|\nabla\mathcal{L}(\boldsymbol{\theta}_\tau)\right\|^2 + \frac{\eta\rho^2\beta_3}{2}\left\|\nabla\mathcal{L}(\boldsymbol{\theta}_\tau)\right\| + \kappa\eta\frac{\beta_3\rho^2}{2}\left\|P(\boldsymbol{\theta}_\tau)\nabla\mathcal{L}(\boldsymbol{\theta}_\tau)\right\|$$

$$+ 2\eta^2\beta_2\left\|\nabla\mathcal{L}(\boldsymbol{\theta}_\tau)\right\|^2 + 2\beta_2^3\eta^2\rho^2 + \beta_2\eta^2\kappa^2\left\|P(\boldsymbol{\theta}_\tau)\nabla\mathcal{L}\left(\boldsymbol{\theta}_\tau + \rho\boldsymbol{v}_\tau\right)\right\|^2$$

$$\overset{\text{Lemma D.5}}{\leqslant} \mathcal{L}(\boldsymbol{\theta}_\tau) - \frac{3\eta}{4}\left\|\nabla\mathcal{L}(\boldsymbol{\theta}_\tau)\right\|^2 + \frac{\eta\rho^2\beta_3}{2}\left\|\nabla\mathcal{L}(\boldsymbol{\theta}_\tau)\right\| + \mathcal{O}\left(\kappa\eta\rho^2 \cdot \eta\rho^2\right)$$

$$+ 2\beta_2^3\eta^2\rho^2 + \beta_2\eta^2\kappa^2\left(\mathcal{O}(\sqrt{\eta}\rho^2) + \frac{\beta_3}{2}\rho^2\right)^2$$

$$\leqslant \mathcal{L}(\boldsymbol{\theta}_\tau) - \frac{3\eta}{4}\left\|\nabla\mathcal{L}(\boldsymbol{\theta}_\tau)\right\|^2 + \frac{\eta\rho^2\beta_3}{2}\left\|\nabla\mathcal{L}(\boldsymbol{\theta}_\tau)\right\| + 2\beta_2^3\eta^2\rho^2 + \beta_2\beta_3^2\eta^2\kappa^2\rho^4 + o(\eta^2\rho^2).$$

From $C_1\eta\rho^2 \leqslant \mathcal{L}(\boldsymbol{\theta}_\tau) \leqslant 2C_1\eta\rho^2$ and $\rho = \mathcal{O}(\sqrt{\eta})$, it holds that

$$\left\|\nabla\mathcal{L}(\boldsymbol{\theta}_\tau)\right\| \geqslant \sqrt{2\mu\mathcal{L}(\boldsymbol{\theta}_\tau)} \geqslant \sqrt{2C_1\mu}\sqrt{\eta}\rho \geqslant 4\beta_3\rho^2.$$

Moreover, recall $\kappa \leqslant 1/\rho$. Therefore, we have the upper bound:

$$\mathbb{E}[\mathcal{L}(\boldsymbol{\theta}_{\tau+1})]$$

$$\leqslant \mathcal{L}(\boldsymbol{\theta}_\tau) - \frac{3\eta}{4}\left\|\nabla\mathcal{L}(\boldsymbol{\theta}_\tau)\right\|^2 + \frac{\eta\rho^2\beta_3}{2}\left\|\nabla\mathcal{L}(\boldsymbol{\theta}_\tau)\right\| + 2\beta_2^3\eta^2\rho^2 + \beta_2\beta_3^2\eta^2\kappa^2\rho^4 + o(\eta^2\rho^2)$$

$$\leqslant \mathcal{L}(\boldsymbol{\theta}_\tau) - \frac{5\eta}{8}\left\|\nabla\mathcal{L}(\boldsymbol{\theta}_\tau)\right\|^2 + 2\beta_2^3\eta^2\rho^2 + \beta_2\beta_3^2\eta^2\kappa^2\rho^4 + o(\eta^2\rho^2)$$

$$\leqslant \mathcal{L}(\boldsymbol{\theta}_\tau) - \frac{10}{8}\eta\mu\mathcal{L}(\boldsymbol{\theta}_\tau) + 2\beta_2^3\eta^2\rho^2 + \beta_2\beta_3^2\eta^2\rho^2 + o(\eta^2\rho^2).$$

$$\leqslant \mathcal{L}(\boldsymbol{\theta}_\tau) - \frac{10}{2}\left(\frac{\beta_2^3}{\mu} \vee \frac{\beta_2\beta_3^2}{\mu}\right)\eta^2\rho^2 + 2\beta_2^3\eta^2\rho^2 + \beta_2\beta_3^2\eta^2\rho^2 + o(\eta^2\rho^2)$$

$$\leqslant \mathcal{L}(\boldsymbol{\theta}_\tau) - \beta_2^3\eta^2\rho^2 = \mathcal{L}(\boldsymbol{\theta}_\tau) - C_2\eta^2\rho^2.$$

- Claim II. $X_{\tau+1} - X_\tau + C_2\eta^2\rho^2$ is $\mathcal{O}(\eta^2\rho^2 + \eta^{3/2}\rho^2/p^{1/2})$-sub-Gaussian. From the definition of $X_\tau$, we only need to prove for the case $C_1\eta\rho^2 \leqslant \mathcal{L}(\boldsymbol{\theta}_\tau) \leqslant 2C_1\eta\rho^2$.
  If $C_1\eta\rho^2 \leqslant \mathcal{L}(\boldsymbol{\theta}_\tau) \leqslant 2C_1\eta\rho^2$, then $X_\tau = \mathcal{L}(\boldsymbol{\theta}_\tau)$ and $X_{\tau+1} = \mathcal{L}(\boldsymbol{\theta}_{\tau+1})$. Similar to Step I, it is clear that $\boldsymbol{\theta}_{\tau+1} \in \mathbb{B}(K; R)$.

Applying the smoothness, we have:

$$
\begin{aligned}
\mathcal{L}(\boldsymbol{\theta}_{\tau+1}) &= \mathcal{L}\left(\boldsymbol{\theta}_\tau - \eta\nabla\mathcal{L}\left(\boldsymbol{\theta}_\tau + \rho\boldsymbol{v}_\tau\right) - \eta\kappa P(\boldsymbol{\theta}_\tau)\nabla\mathcal{L}\left(\boldsymbol{\theta}_\tau + \rho\boldsymbol{v}_\tau\right)\right) \\
&= \mathcal{L}(\boldsymbol{\theta}_\tau) - \eta\left\langle\nabla\mathcal{L}(\boldsymbol{\theta}_\tau), \nabla\mathcal{L}\left(\boldsymbol{\theta}_\tau + \rho\boldsymbol{v}_\tau\right) + \kappa P(\boldsymbol{\theta}_t)\nabla\mathcal{L}\left(\boldsymbol{\theta}_\tau + \rho\boldsymbol{v}_\tau\right)\right\rangle \\
&\quad + \left(\eta^2\left\|\nabla\mathcal{L}\left(\boldsymbol{\theta}_\tau + \rho\boldsymbol{v}_\tau\right) + \kappa P(\boldsymbol{\theta}_\tau)\nabla\mathcal{L}\left(\boldsymbol{\theta}_\tau + \rho\boldsymbol{v}_\tau\right)\right\|^2\right) \\
&= \mathcal{L}(\boldsymbol{\theta}_\tau) - \eta\left\|\nabla\mathcal{L}(\boldsymbol{\theta}_\tau)\right\|^2 - \eta\rho\left\langle\nabla\mathcal{L}(\boldsymbol{\theta}_\tau), \nabla^2\mathcal{L}(\boldsymbol{\theta}_\tau)\boldsymbol{v}_\tau\right\rangle + \mathcal{O}\left(\eta\rho^2\left\|\nabla\mathcal{L}(\boldsymbol{\theta}_\tau)\right\|\right) \\
&\quad - \eta\kappa\left\langle\nabla\mathcal{L}(\boldsymbol{\theta}_\tau), P(\boldsymbol{\theta}_\tau)\nabla\mathcal{L}(\boldsymbol{\theta}_\tau)\right\rangle - \eta\kappa\rho\left\langle\nabla\mathcal{L}(\boldsymbol{\theta}_\tau), P(\boldsymbol{\theta}_\tau)\nabla^2\mathcal{L}(\boldsymbol{\theta}_\tau)\boldsymbol{v}_t\right\rangle + \mathcal{O}\left(\eta\kappa\rho^2\left\|P(\boldsymbol{\theta}_\tau)\nabla\mathcal{L}(\boldsymbol{\theta}_\tau)\right\|\right) \\
&\quad + \left(\eta^2\left\|\nabla\mathcal{L}\left(\boldsymbol{\theta}_\tau + \rho\boldsymbol{v}_\tau\right) + \kappa P(\boldsymbol{\theta}_\tau)\nabla\mathcal{L}\left(\boldsymbol{\theta}_\tau + \rho\boldsymbol{v}_\tau\right)\right\|^2\right).
\end{aligned}
$$

In the same way as the proof of Claim I, it holds that

$$
\eta\left\|\nabla\mathcal{L}(\boldsymbol{\theta}_\tau)\right\|^2 = \mathcal{O}(\eta^2\rho^2), \quad \eta\rho^2\left\|\nabla\mathcal{L}(\boldsymbol{\theta}_\tau)\right\| = \mathcal{O}(\eta^2\rho^2), \quad \eta\kappa\rho^2\left\|P(\boldsymbol{\theta}_\tau)\nabla\mathcal{L}(\boldsymbol{\theta}_\tau)\right\| = \mathcal{O}\left(\eta^2\rho^3\right),
$$

$$
\eta^2\left\|\nabla\mathcal{L}\left(\boldsymbol{\theta}_\tau + \rho\boldsymbol{v}_\tau\right) + \kappa P(\boldsymbol{\theta}_\tau)\nabla\mathcal{L}\left(\boldsymbol{\theta}_\tau + \rho\boldsymbol{v}_\tau\right)\right\|^2 = \mathcal{O}\left(\eta^2\rho^2\right)
$$

Thus,

$$
\begin{aligned}
&\mathcal{L}(\boldsymbol{\theta}_{\tau+1}) - \mathcal{L}(\boldsymbol{\theta}_\tau) \\
&= -\eta\rho\left\langle\nabla\mathcal{L}(\boldsymbol{\theta}_\tau), \nabla^2\mathcal{L}(\boldsymbol{\theta}_\tau)\boldsymbol{v}_t\right\rangle - \eta\kappa\rho\left\langle\nabla\mathcal{L}(\boldsymbol{\theta}_\tau), P(\boldsymbol{\theta}_\tau)\nabla^2\mathcal{L}(\boldsymbol{\theta}_\tau)\boldsymbol{v}_t\right\rangle + \mathcal{O}(\eta^2\rho^2),
\end{aligned}
$$

which implies:

$$
\begin{aligned}
\left\|X_{\tau+1} - X_\tau - C_2\eta^2\rho^2\right\|_{\psi_2} &\leqslant \left\|\mathcal{L}(\boldsymbol{\theta}_{\tau+1}) - \mathcal{L}(\boldsymbol{\theta}_\tau)\right\|_{\psi_2} + \left\|C_2\eta^2\rho^2\right\|_{\psi_2} \\
&\leqslant \eta\rho\left\|\left\langle\nabla\mathcal{L}(\boldsymbol{\theta}_\tau), \nabla^2\mathcal{L}(\boldsymbol{\theta}_\tau)\boldsymbol{v}_t\right\rangle\right\|_{\psi_2} + \eta\kappa\rho\left\|\left\langle\nabla\mathcal{L}(\boldsymbol{\theta}_\tau), P(\boldsymbol{\theta}_\tau)\nabla^2\mathcal{L}(\boldsymbol{\theta}_\tau)\boldsymbol{v}_t\right\rangle\right\|_{\psi_2} + \mathcal{O}(\eta^2\rho^2) \\
&\leqslant \eta\rho\left\|\nabla^2\mathcal{L}(\boldsymbol{\theta}_\tau)\nabla\mathcal{L}(\boldsymbol{\theta}_\tau)\right\|\left\|\left\langle\frac{\nabla^2\mathcal{L}(\boldsymbol{\theta}_\tau)\nabla\mathcal{L}(\boldsymbol{\theta}_\tau)}{\left\|\nabla^2\mathcal{L}(\boldsymbol{\theta}_\tau)\nabla\mathcal{L}(\boldsymbol{\theta}_\tau)\right\|}, \boldsymbol{v}_t\right\rangle\right\|_{\psi_2} \\
&\quad + \eta\kappa\rho\left\|\nabla^2\mathcal{L}(\boldsymbol{\theta}_\tau)P(\boldsymbol{\theta}_\tau)\nabla\mathcal{L}(\boldsymbol{\theta}_\tau)\right\|\left\|\left\langle\frac{\nabla^2\mathcal{L}(\boldsymbol{\theta}_\tau)P(\boldsymbol{\theta}_\tau)\nabla\mathcal{L}(\boldsymbol{\theta}_\tau)}{\left\|\nabla^2\mathcal{L}(\boldsymbol{\theta}_\tau)P(\boldsymbol{\theta}_\tau)\nabla\mathcal{L}(\boldsymbol{\theta}_\tau)\right\|}, \boldsymbol{v}_t\right\rangle\right\|_{\psi_2} + \mathcal{O}(\eta^2\rho^2) \\
&\overset{\text{Lemma D.5}}{\leqslant} \mathcal{O}\left(\eta^{3/2}\rho^2\left\|\left\langle\frac{\nabla^2\mathcal{L}(\boldsymbol{\theta}_\tau)\nabla\mathcal{L}(\boldsymbol{\theta}_\tau)}{\left\|\nabla^2\mathcal{L}(\boldsymbol{\theta}_\tau)\nabla\mathcal{L}(\boldsymbol{\theta}_\tau)\right\|}, \boldsymbol{v}_t\right\rangle\right\|_{\psi_2}\right) \\
&\quad + \mathcal{O}\left(\eta^{3/2}\rho^2\left\|\left\langle\frac{\nabla^2\mathcal{L}(\boldsymbol{\theta}_\tau)P(\boldsymbol{\theta}_\tau)\nabla\mathcal{L}(\boldsymbol{\theta}_\tau)}{\left\|\nabla^2\mathcal{L}(\boldsymbol{\theta}_\tau)P(\boldsymbol{\theta}_\tau)\nabla\mathcal{L}(\boldsymbol{\theta}_\tau)\right\|}, \boldsymbol{v}_t\right\rangle\right\|_{\psi_2}\right) + \mathcal{O}(\eta^2\rho^2) \\
&\overset{\text{Lemma F.5}}{\leqslant} \mathcal{O}\left(\eta^{3/2}\rho^2/\sqrt{p}\right) + \mathcal{O}(\eta^2\rho^2).
\end{aligned}
$$

With the preparation of Claim I and Claim II, we can use the Azuma-Hoeffding inequality (Lemma F.4 (ii)): for any $Q > 0$, it holds that

$$
\mathbb{P}\left(X_{t+1} - X_{s+1} + (t-s)C_2\eta^2\rho^2 > Q\right) \leqslant 2\exp\left(-\frac{Q^2}{2(t-s)\left(\mathcal{O}(\eta^{3/2}\rho^2/p^{1/2} + \eta^2\rho^2)\right)^2}\right).
$$

As proved in Claim I, $X_{s+1} = \mathcal{L}(\boldsymbol{\theta}_{s+1}) \leqslant \frac{3}{2}C_1\eta\rho^2$ due to $\mathcal{L}(\boldsymbol{\theta}_s) \leqslant C_1\eta\rho^2$. Therefore, by choosing $Q = (t-s)C_2\eta^2\rho^2 - \frac{3}{2}C_1\eta\rho^2 + 2C_1\eta\rho^2 = (t-s)C_2\eta^2\rho^2 + \frac{1}{2}C_1\eta\rho^2$, we have

$$
\begin{aligned}
\mathbb{P}_{t+1,s} &\leqslant \mathbb{P}\left(X_{t+1} \geqslant 2C_1\eta\rho^2\right) \\
&\leqslant \mathbb{P}\left(X_{t+1} - X_{s+1} + (t-s)C_2\eta^2\rho^2 > (t-s)C_2\eta^2\rho^2 + \frac{1}{2}C_1\eta\rho^2\right) \\
&\leqslant 2\exp\left(-\frac{\left((t-s)C_2\eta^2\rho^2 + \frac{1}{2}C_1\eta\rho^2\right)^2}{2(t-s)\left(\mathcal{O}(\eta^{3/2}\rho^2/p^{1/2} + \eta^2\rho^2)\right)^2}\right) \\
&\leqslant 2\exp\left(-\frac{4(t-s)C_2\eta^2\rho^2 \cdot \frac{1}{2}C_1\eta\rho^2}{4(t-s)\left(\mathcal{O}(\eta^3\rho^4/p + \eta^4\rho^4)\right)}\right) \leqslant 2\exp\left(-\Omega\left(\frac{1}{\eta+p^{-1}}\right)\right).
\end{aligned}
$$

Therefore, we obtain the union bound:

$$\mathbb{P}\left(\exists\, t \in [T_\mathrm{I}, T_\mathrm{I} + T_\mathrm{II}], \mathcal{L}(\boldsymbol{\theta}_t) \geqslant 2C_1\eta\rho^2\right) \leqslant \sum_{t=T_I}^{T_\mathrm{I}+T_\mathrm{II}-1}\left(\mathbb{P}_{t+1,t} + \sum_{s=T_\mathrm{I}}^{t-1}\mathbb{P}_{t+1,s}\right)$$

$$\leqslant \sum_{t=T_I}^{T_\mathrm{I}+T_\mathrm{II}-1}\sum_{s=T_\mathrm{I}}^{t-1}\mathbb{P}_{t+1,s} \leqslant T_\mathrm{II}^2 \exp\left(-\Omega\left(\frac{1}{\eta+p^{-1}}\right)\right).$$

Hence, with probability at least $1 - T_\mathrm{II}^2 \exp\left(-\Omega\left(\frac{1}{\eta+p^{-1}}\right)\right)$, for any $t \in [T_\mathrm{I}, T_\mathrm{I} + T_\mathrm{II}]$,

$$\|\boldsymbol{\theta}_t - \Phi(\boldsymbol{\theta}_t)\| \leqslant \sqrt{\frac{2}{\mu}\mathcal{L}(\boldsymbol{\theta}_t)} \leqslant 2\sqrt{\frac{C_1}{\mu}}\sqrt{\eta}\rho = \frac{4\beta_2^{3/2}}{\mu}\sqrt{\eta}\rho = \mathcal{O}(\sqrt{\eta}\rho).$$

### D.2.3  Proof of the Effective Dynamics

We have proved that with high probability at least $1 - T_\mathrm{II}^2 \exp\left(-\Omega\left(\frac{1}{\eta+p^{-1}}\right)\right)$, for any $t \in [T_\mathrm{I}, T_\mathrm{I} + T_\mathrm{II}]$, $\|\boldsymbol{\theta}_t - \Phi(\boldsymbol{\theta}_t)\| = \mathcal{O}(\sqrt{\eta}\rho)$. Then we prove this theorem when the above event occurs.
For any $t \in [T_\mathrm{I}, T_\mathrm{I} + T_\mathrm{II}]$,

$$\|\boldsymbol{\theta}_{t+1} - \boldsymbol{\theta}_t\|$$
$$\leqslant \eta\|\nabla\mathcal{L}(\boldsymbol{\theta}_t + \rho\boldsymbol{v}_t)\| + \eta\kappa\|P(\boldsymbol{\theta}_t)\nabla\mathcal{L}(\boldsymbol{\theta}_t + \rho\boldsymbol{v}_t)\|$$
$$\leqslant \eta\|\nabla\mathcal{L}(\boldsymbol{\theta}_t)\| + \mathcal{O}(\eta\rho) + \eta\kappa\|P(\boldsymbol{\theta}_t)\nabla\mathcal{L}(\boldsymbol{\theta}_t)\| + \eta\kappa\rho\|P(\boldsymbol{\theta}_t)\nabla^2\mathcal{L}(\boldsymbol{\theta}_t)\| + \mathcal{O}(\eta\kappa\rho^2)$$
$$\overset{\text{Lemma D.5}}{\leqslant} \mathcal{O}(\eta^{3/2}\rho) + \mathcal{O}(\eta\rho) + \mathcal{O}(\eta^{3/2}\rho) + \mathcal{O}(\eta^{3/2}\rho^2) + \mathcal{O}(\eta\rho^2) + \mathcal{O}(\eta\rho) = \mathcal{O}(\eta\rho).$$

Then by Taylor's expansion,

$$\Phi(\boldsymbol{\theta}_{t+1}) - \Phi(\boldsymbol{\theta}_t) = \partial\Phi(\boldsymbol{\theta}_t)(\boldsymbol{\theta}_{t+1} - \boldsymbol{\theta}_t) + \mathcal{O}\left(\|\boldsymbol{\theta}_{t+1} - \boldsymbol{\theta}_t\|^2\right)$$
$$= -\eta\partial\Phi(\boldsymbol{\theta}_t)\nabla\mathcal{L}(\boldsymbol{\theta}_t + \rho\boldsymbol{v}_t) - \eta\kappa\partial\Phi(\boldsymbol{\theta}_t)P(\boldsymbol{\theta}_t)\nabla\mathcal{L}(\boldsymbol{\theta}_t + \rho\boldsymbol{v}_t) + \mathcal{O}(\eta^2\rho^2).$$

For the term $\partial\Phi(\boldsymbol{\theta}_t)\nabla\mathcal{L}(\boldsymbol{\theta}_t + \rho\boldsymbol{v}_t)$ and $\partial\Phi(\boldsymbol{\theta}_t)P(\boldsymbol{\theta}_t)\nabla\mathcal{L}(\boldsymbol{\theta}_t + \rho\boldsymbol{v}_t)$, using Taylor's expansion, we have

$$\partial\Phi(\boldsymbol{\theta}_t)\nabla\mathcal{L}(\boldsymbol{\theta}_t + \rho\boldsymbol{v}_t)$$
$$=\partial\Phi(\boldsymbol{\theta}_t)\nabla\mathcal{L}(\boldsymbol{\theta}_t) + \rho\partial\Phi(\boldsymbol{\theta}_t)\nabla^2\mathcal{L}(\boldsymbol{\theta}_t)\boldsymbol{v}_t + \frac{\rho^2}{2}\partial\Phi(\boldsymbol{\theta}_t)\nabla\,\mathrm{Tr}\left(\boldsymbol{v}_t\nabla^2\mathcal{L}(\boldsymbol{\theta}_t)\boldsymbol{v}_t^\top\right) + \mathcal{O}(\rho^3)$$
$$=\partial\Phi(\boldsymbol{\theta}_t)\nabla\mathcal{L}(\boldsymbol{\theta}_t) + \rho\partial\Phi(\boldsymbol{\theta}_t)\nabla^2\mathcal{L}(\boldsymbol{\theta}_t)\boldsymbol{v}_t + \frac{\rho^2}{2}\partial\Phi(\boldsymbol{\theta}_t)\nabla\left(\boldsymbol{v}_t^\top\nabla^2\mathcal{L}(\boldsymbol{\theta}_t)\boldsymbol{v}_t\right) + \mathcal{O}(\rho^3),$$

$$\partial\Phi(\boldsymbol{\theta}_t)P(\boldsymbol{\theta}_t)\nabla\mathcal{L}(\boldsymbol{\theta}_t + \rho\boldsymbol{v}_t)$$
$$=\partial\Phi(\boldsymbol{\theta}_t)P(\boldsymbol{\theta}_t)\nabla\mathcal{L}(\boldsymbol{\theta}_t) + \rho\partial\Phi(\boldsymbol{\theta}_t)P(\boldsymbol{\theta}_t)\nabla^2\mathcal{L}(\boldsymbol{\theta}_t)\boldsymbol{v}_t + \frac{\rho^2}{2}\partial\Phi(\boldsymbol{\theta}_t)P(\boldsymbol{\theta}_t)\nabla\,\mathrm{Tr}\left(\boldsymbol{v}_t\nabla^2\mathcal{L}(\boldsymbol{\theta}_t)\boldsymbol{v}_t^\top\right) + \mathcal{O}(\rho^3)$$
$$=\partial\Phi(\boldsymbol{\theta}_t)P(\boldsymbol{\theta}_t)\nabla\mathcal{L}(\boldsymbol{\theta}_t) + \rho\partial\Phi(\boldsymbol{\theta}_t)P(\boldsymbol{\theta}_t)\nabla^2\mathcal{L}(\boldsymbol{\theta}_t)\boldsymbol{v}_t + \frac{\rho^2}{2}\partial\Phi(\boldsymbol{\theta}_t)P(\boldsymbol{\theta}_t)\nabla\left(\boldsymbol{v}_t^\top\nabla^2\mathcal{L}(\boldsymbol{\theta}_t)\boldsymbol{v}_t\right) + \mathcal{O}(\rho^3).$$

Taking the expectation (about $\boldsymbol{v}_t$), we have

$$\mathbb{E}[\boldsymbol{v}_t] = 0, \quad \mathbb{E}\left[\boldsymbol{v}_t^\top\nabla^2\mathcal{L}(\boldsymbol{\theta}_t)\boldsymbol{v}_t\right] = \frac{\mathrm{Tr}\left(\nabla^2\mathcal{L}(\boldsymbol{\theta}_t)\right)}{p}.$$

Additionally, using Lemma D.2 and Taylor's expansion, we have:

$$\partial\Phi(\boldsymbol{\theta}_t)\nabla\mathcal{L}(\boldsymbol{\theta}_t) = \mathbf{0};$$

$$\partial\Phi(\boldsymbol{\theta}_t) = \partial\Phi(\Phi(\boldsymbol{\theta}_t)) + \|\boldsymbol{\theta}_t - \Phi(\boldsymbol{\theta}_t)\| = \partial\Phi(\Phi(\boldsymbol{\theta}_t)) + \mathcal{O}(\sqrt{\eta}\rho);$$

$$\partial\Phi(\boldsymbol{\theta}_t)P(\boldsymbol{\theta}_t) = \partial\Phi(\Phi(\boldsymbol{\theta}_t))P(\Phi(\boldsymbol{\theta}_t)) + \|\boldsymbol{\theta}_t - \Phi(\boldsymbol{\theta}_t)\| = \partial\Phi(\Phi(\boldsymbol{\theta}_t)) + \mathcal{O}(\sqrt{\eta}\rho);$$

$$\nabla\,\mathrm{Tr}\left(\nabla^2\mathcal{L}(\boldsymbol{\theta}_t)\right) = \nabla\,\mathrm{Tr}\left(\nabla^2\mathcal{L}(\Phi(\boldsymbol{\theta}_t))\right) + \|\boldsymbol{\theta}_t - \Phi(\boldsymbol{\theta}_t)\| = \nabla\,\mathrm{Tr}\left(\nabla^2\mathcal{L}(\Phi(\boldsymbol{\theta}_t))\right) + \mathcal{O}(\sqrt{\eta}\rho);$$

$$\begin{aligned}
\partial\Phi(\boldsymbol{\theta}_t)P(\boldsymbol{\theta}_t)\nabla\mathcal{L}(\boldsymbol{\theta}_t) &= \partial\Phi(\Phi(\boldsymbol{\theta}_t))P(\Phi(\boldsymbol{\theta}_t))\nabla\mathcal{L}(\boldsymbol{\theta}_t) + \mathcal{O}\left(\|\boldsymbol{\theta}_t - \Phi(\boldsymbol{\theta}_t)\|\,\|\nabla\mathcal{L}(\boldsymbol{\theta}_t)\|\right) \\
&= \partial\Phi(\Phi(\boldsymbol{\theta}_t))\nabla\mathcal{L}(\boldsymbol{\theta}_t) + \mathcal{O}\left(\eta\rho^2\right) = \mathbf{0} + \mathcal{O}(\eta\rho^2).
\end{aligned}$$

Combining the results above, we obtain:

$$\begin{aligned}
&\mathbb{E}[\Phi(\boldsymbol{\theta}_{t+1})] \\
=&\Phi(\boldsymbol{\theta}_t) - \eta\partial\Phi(\boldsymbol{\theta}_t)\nabla\mathcal{L}(\boldsymbol{\theta}_t) - \eta\kappa\partial\Phi(\boldsymbol{\theta}_t)P(\boldsymbol{\theta}_t)\nabla\mathcal{L}(\boldsymbol{\theta}_t) \\
&- \frac{\eta\rho^2}{2p}\partial\Phi(\boldsymbol{\theta}_t)\nabla\,\mathrm{Tr}\left(\nabla^2\mathcal{L}(\boldsymbol{\theta}_t)\right) - \frac{\kappa\eta\rho^2}{2p}\partial\Phi(\boldsymbol{\theta}_t)P(\boldsymbol{\theta}_t)\nabla\,\mathrm{Tr}\left(\nabla^2\mathcal{L}(\boldsymbol{\theta}_t)\right) + \mathcal{O}(\eta\rho^3) \\
=&\Phi(\boldsymbol{\theta}_t) + \mathcal{O}\left(\eta^2\kappa\rho^2\right) - \frac{(\kappa+1)\eta\rho^2}{2p}\partial\Phi(\boldsymbol{\theta}_t)\nabla\,\mathrm{Tr}\left(\nabla^2\mathcal{L}(\boldsymbol{\theta}_t)\right) + \mathcal{O}(\eta^{3/2}\rho^3) + \mathcal{O}(\kappa\eta^{3/2}\rho^3) + \mathcal{O}(\eta\rho^3) \\
=&\Phi(\boldsymbol{\theta}_t) + \mathcal{O}\left(\eta^2\kappa\rho^2\right) - \frac{(\kappa+1)\eta\rho^2}{2p}\partial\Phi(\boldsymbol{\theta}_t)\nabla\,\mathrm{Tr}\left(\nabla^2\mathcal{L}(\Phi(\boldsymbol{\theta}_t))\right) + \mathcal{O}(\eta^{3/2}\rho^3) + \mathcal{O}(\kappa\eta^{3/2}\rho^3) + \mathcal{O}(\eta\rho^3) \\
=&\Phi(\boldsymbol{\theta}_t) - (\kappa+1)\eta\rho^2\partial\Phi(\boldsymbol{\theta}_t)\nabla\,\mathrm{Tr}\left(\nabla^2\mathcal{L}(\Phi(\boldsymbol{\theta}_t))/2p\right) + \mathcal{O}(\eta^{3/2}\rho^2).
\end{aligned}$$

# E  Proofs in Section 5.2.2

**Setting E.1.** Consider the empirical risk minimization $\min : \mathcal{L}(\boldsymbol{\theta}) = \frac{1}{n}\sum_{i=1}^{n}\mathcal{L}_i(\boldsymbol{\theta})$, where $\mathcal{L}_i(\boldsymbol{\theta}) = \ell(f_i(\boldsymbol{\theta}), y_i)$ is the loss on the $i$-th data $(\boldsymbol{x}_i, y_i)$. Let $f_i(\cdot)$ and $\ell(\cdot, \cdot)$ be $\mathcal{C}^4$. Suppose all global minimizers interpolate the training dataset, i.e., $\mathcal{L}(\boldsymbol{\theta}^\star) = \min_{\boldsymbol{\theta}}\mathcal{L}(\boldsymbol{\theta})$ implies $f_i(\boldsymbol{\theta}^\star) = y_i$ for all $i \in [n]$. We denote the minima manifold by $\mathcal{M} = \{\boldsymbol{\theta} : f_i(\boldsymbol{\theta}) = y_i, \forall i \in [n]\}$. Moreover, we assume that $\frac{\partial^2\ell(\hat{y},y)}{\partial\hat{y}^2}|_{\hat{y}=y} > 0$ and the feature matrix $(\nabla f_i(\boldsymbol{\theta}), \cdots, \nabla f_n(\boldsymbol{\theta})) \in \mathbb{R}^{p\times n}$ is full-rank at $\boldsymbol{\theta} \in \mathcal{M}$.

**Lemma E.2** (Theorem 5.2 in Wen et al. (2023a)). *Under Setting E.1, Assumption 5.1 holds with $m = n$.*

## E.1  Preliminary Lemmas

Similar to the proofs for Section 5.2.1, we need the following similar preliminary lemmas.

**Lemma E.3.** *Under Setting E.1, for any compact set $K \in \Gamma$, there exist absolute constants $R_1, \mu > 0$ such that*

- *(i) $\overline{\mathbb{B}(K; R_1)} \subset U$;*

- *(ii) $\mathcal{L}_i(\cdot)$ $(i \in [n])$ and $\mathcal{L}(\cdot)$ are $\mu$-PL on $\overline{\mathbb{B}(K; R_1)}$;*

- *(iii) $\inf_{\boldsymbol{\theta}\in\mathbb{B}(K;R_1)}\lambda_n\left(\nabla^2\mathcal{L}(\boldsymbol{\theta})\right) \geqslant \mu$; $\inf_{\boldsymbol{\theta}\in\mathbb{B}(K;R_1)}\lambda_1\left(\nabla^2\mathcal{L}_i(\boldsymbol{\theta})\right) \geqslant \mu$, $\forall i \in [n]$.*

*We further define the following absolute constants on $\overline{\mathbb{B}(K; R)}$:*

$$\beta_2 := \left(\sup_{\boldsymbol{\theta}\in\mathbb{B}(K;R_1)}\left\|\nabla^2\mathcal{L}(\boldsymbol{\theta})\right\|\right) \vee \left(\max_{i\in[n]}\sup_{\boldsymbol{\theta}\in\mathbb{B}(K;R_1)}\left\|\nabla^2\mathcal{L}_i(\boldsymbol{\theta})\right\|\right);$$

$$\beta_3 := \left(\sup_{\boldsymbol{\theta}\in\mathbb{B}(K;R_1)}\left\|\nabla^3\mathcal{L}(\boldsymbol{\theta})\right\|\right) \vee \left(\max_{i\in[n]}\sup_{\boldsymbol{\theta}\in\mathbb{B}(K;R_1)}\left\|\nabla^3\mathcal{L}_i(\boldsymbol{\theta})\right\|\right);$$

$$\nu := \left(\inf_{\boldsymbol{\theta}\in\mathbb{B}(K;R_1)}\lambda_m\left(\nabla^2\mathcal{L}(\boldsymbol{\theta})\right)\right) \wedge \left(\min_{i\in[n]}\inf_{\boldsymbol{\theta}\in\mathbb{B}(K;R_1)}\lambda_1\left(\nabla^2\mathcal{L}_i(\boldsymbol{\theta})\right)\right);$$

$$\zeta_1^\Phi := \sup_{\boldsymbol{\theta}\in\mathbb{B}(K;R_1)}\|\nabla\Phi(\boldsymbol{\theta})\|, \quad \zeta_2^\Phi := \sup_{\boldsymbol{\theta}\in\mathbb{B}(K;R_1)}\left\|\nabla^2\Phi(\boldsymbol{\theta})\right\|.$$

**Lemma E.4** ([Wen et al. (2023a)](#)). *Under Assumption 5.1,*

- *For any $\boldsymbol{\theta} \in U$, $\partial\Phi(\boldsymbol{\theta})\nabla\mathcal{L}(\boldsymbol{\theta}) = \mathbf{0}$.*

- *For any $\boldsymbol{\theta} \in \Gamma$, $\partial\Phi(\boldsymbol{\theta}) = P_{n+1:p}(\nabla^2\mathcal{L}(\boldsymbol{\theta}))$ and $\partial\Phi(\boldsymbol{\theta})\nabla^2\mathcal{L}(\boldsymbol{\theta}) = 0$.*

- *For any $\boldsymbol{\theta} \in \Gamma$, $\partial\Phi(\boldsymbol{\theta})\nabla^2\mathcal{L}_i(\boldsymbol{\theta}) = 0$, $\forall i \in [n]$.*

**Lemma E.5.** *Under Setting E.1, there exists absolute constants $R_2, \zeta_P > 0$ such that for any $\boldsymbol{\theta} \in \mathbb{B}(K; R_2)$,*

$$\left\| P_{n+1:p}(\nabla^2\mathcal{L}(\boldsymbol{\theta})) - P_{n+1:p}(\nabla^2\mathcal{L}(\Phi(\boldsymbol{\theta}))) \right\| \leqslant \zeta_P \left\| \boldsymbol{\theta} - \Phi(\boldsymbol{\theta}) \right\|.$$

**Proof Notations.** Now we introduce some additional useful notations in the proof in this section.

First, we choose $R := (R_1 \wedge R_2)/2$, where $R_1$ is defined in Lemma E.3 and $R_2$ is defined in Lemma E.5. Let $\mu$ be the PL constant on $\mathbb{B}(K; R)$. Moreover, we use the following notations:

$$\beta_2 := \left( \sup_{\boldsymbol{\theta} \in \mathbb{B}(K;R)} \left\| \nabla^2\mathcal{L}(\boldsymbol{\theta}) \right\| \right) \vee \left( \max_{i \in [n]} \sup_{\boldsymbol{\theta} \in \mathbb{B}(K;R)} \left\| \nabla^2\mathcal{L}_i(\boldsymbol{\theta}) \right\| \right);$$

$$\beta_3 := \left( \sup_{\boldsymbol{\theta} \in \mathbb{B}(K;R)} \left\| \nabla^3\mathcal{L}(\boldsymbol{\theta}) \right\| \right) \vee \left( \max_{i \in [n]} \sup_{\boldsymbol{\theta} \in \mathbb{B}(K;R)} \left\| \nabla^3\mathcal{L}_i(\boldsymbol{\theta}) \right\| \right);$$

$$\beta_4 := \left( \sup_{\boldsymbol{\theta} \in \mathbb{B}(K;R)} \left\| \nabla^3\mathcal{L}(\boldsymbol{\theta}) \right\| \right) \vee \left( \max_{i \in [n]} \sup_{\boldsymbol{\theta} \in \mathbb{B}(K;R)} \left\| \nabla^4\mathcal{L}_i(\boldsymbol{\theta}) \right\| \right);$$

$$\nu := \left( \inf_{\boldsymbol{\theta} \in \mathbb{B}(K;R)} \lambda_m\left(\nabla^2\mathcal{L}(\boldsymbol{\theta})\right) \right) \wedge \left( \min_{i \in [n]} \inf_{\boldsymbol{\theta} \in \mathbb{B}(K;R)} \lambda_1\left(\nabla^2\mathcal{L}_i(\boldsymbol{\theta})\right) \right);$$

$$\zeta_\Phi := \sup_{\boldsymbol{\theta} \in \mathbb{B}(K;R)} \left\| \nabla^2\Phi(\boldsymbol{\theta}) \right\|; \quad \zeta_P := \sup_{\boldsymbol{\theta} \in \mathbb{B}(K;R) - \Gamma} \frac{\left\| P_{n+1:p}(\nabla^2\mathcal{L}(\boldsymbol{\theta})) - P_{n+1:p}(\nabla^2\mathcal{L}(\Phi(\boldsymbol{\theta}))) \right\|}{\left\| \boldsymbol{\theta} - \Phi(\boldsymbol{\theta}) \right\|}.$$

$$(16)$$

Ensured by Lemma D.1 and D.3, these quantities are all absolute constants in $(0, +\infty)$. Moreover, without loss of generality, we can assume that $\beta_1, \beta_2, \beta_3, \zeta_\Phi, \zeta_P > 1$ and $\mu \leqslant \nu < 1$.

Then we have the following two lemmas, similar to Lemma D.4 and D.5.

**Lemma E.6.** *For any $\boldsymbol{\theta} \in \mathbb{B}(K; R)$, it holds that:*

- *(para norm v.s. grad norm)* $\mu \left\| \nabla\mathcal{L}(\boldsymbol{\theta}) \right\| \leqslant \left\| \boldsymbol{\theta} - \Phi(\boldsymbol{\theta}) \right\| \leqslant \beta_2 \left\| \nabla\mathcal{L}(\boldsymbol{\theta}) \right\|$; $\mu \left\| \nabla\mathcal{L}_i(\boldsymbol{\theta}) \right\| \leqslant \left\| \boldsymbol{\theta} - \Phi(\boldsymbol{\theta}) \right\| \leqslant \beta_2 \left\| \nabla\mathcal{L}_i(\boldsymbol{\theta}) \right\|$, $\forall i \in [n]$.

- *(grad norm v.s. loss)* $2\mu\mathcal{L}(\boldsymbol{\theta}) \leqslant \left\| \nabla\mathcal{L}(\boldsymbol{\theta}) \right\|^2 \leqslant \frac{2\beta_2^2}{\mu}\mathcal{L}(\boldsymbol{\theta})$; $2\mu\mathcal{L}_i(\boldsymbol{\theta}) \leqslant \left\| \nabla\mathcal{L}_i(\boldsymbol{\theta}) \right\|^2 \leqslant \frac{2\beta_2^2}{\mu}\mathcal{L}_i(\boldsymbol{\theta})$, $\forall i \in [n]$.

- *(loss v.s. para norm)* $\frac{\mu}{2} \left\| \boldsymbol{\theta} - \Phi(\boldsymbol{\theta}) \right\|^2 \leqslant \mathcal{L}(\boldsymbol{\theta}) \leqslant \frac{\beta_2^2}{2\mu} \left\| \boldsymbol{\theta} - \Phi(\boldsymbol{\theta}) \right\|^2$; $\frac{\mu}{2} \left\| \boldsymbol{\theta} - \Phi(\boldsymbol{\theta}) \right\|^2 \leqslant \mathcal{L}_i(\boldsymbol{\theta}) \leqslant \frac{\beta_2^2}{2\mu} \left\| \boldsymbol{\theta} - \Phi(\boldsymbol{\theta}) \right\|^2$, $\forall i \in [n]$.

**Lemma E.7.** *For all $\boldsymbol{\theta} \in \mathbb{B}(K; R)$, $\forall i \in [n]$,*

- $$\left\| P_{n+1:p}(\nabla^2\mathcal{L}(\boldsymbol{\theta}))\nabla^2\mathcal{L}(\boldsymbol{\theta}) \right\|, \left\| P_{n+1:p}(\nabla^2\mathcal{L}(\boldsymbol{\theta}))\nabla^2\mathcal{L}_i(\boldsymbol{\theta}) \right\|, \left\| \partial\Phi(\boldsymbol{\theta})\nabla^2\mathcal{L}_i(\boldsymbol{\theta}) \right\| \leqslant \mathcal{O}\left(\left\| \boldsymbol{\theta} - \Phi(\boldsymbol{\theta}) \right\|\right);$$

- $$\left\| P_{n+1:p}(\nabla^2\mathcal{L}(\boldsymbol{\theta}))\nabla\mathcal{L}(\boldsymbol{\theta}) \right\|, \left\| P_{n+1:p}(\nabla^2\mathcal{L}(\boldsymbol{\theta}))\nabla\mathcal{L}_i(\boldsymbol{\theta}) \right\|, \left\| \partial\Phi(\boldsymbol{\theta})\nabla\mathcal{L}_i(\boldsymbol{\theta}) \right\| \leqslant \mathcal{O}\left(\left\| \boldsymbol{\theta} - \Phi(\boldsymbol{\theta}) \right\|^2\right);$$

- *Let $\rho > 0$ and $\boldsymbol{v} \in \mathbb{S}^{p-1}$. If $\boldsymbol{\theta} + \rho\boldsymbol{v} \in \mathbb{B}(K; R)$, then*
$$\left\| \nabla\mathcal{L}_i(\boldsymbol{\theta} + \rho\boldsymbol{v}) \right\| \leqslant \left\| \nabla\mathcal{L}_i(\boldsymbol{\theta}) \right\| + \rho\beta_2, \forall i \in [n];;$$

$$\left\| P_{n+1:p}(\nabla^2\mathcal{L}(\boldsymbol{\theta}))\nabla\mathcal{L}_i(\boldsymbol{\theta} + \rho\boldsymbol{v}) \right\| \leqslant \mathcal{O}\left(\left\| \boldsymbol{\theta} - \Phi(\boldsymbol{\theta}) \right\|^2\right) + \mathcal{O}\left(\rho \left\| \boldsymbol{\theta} - \Phi(\boldsymbol{\theta}) \right\|\right) + \frac{\rho^2\beta_3}{2}, \forall i \in [n];$$

**Lemma E.8** (Lemma H.9 in Wen et al. (2023a))**.** *For any absolute constant $C > 0$, there exist absolute constant $C_1, C_2 > 0$ such that: if $\boldsymbol{\theta}_t \in \mathbb{B}(K; R)$ and $C_1 \eta \rho \leqslant \|\boldsymbol{\theta}_t - \Phi(\boldsymbol{\theta}_t)\| \leqslant C\rho$, then it holds that:*

$$\mathbb{E}_{i_t}\left[\left\|\boldsymbol{\theta}_{t+1/2} - \Phi(\boldsymbol{\theta}_{t+1/2})\right\|\right] \leqslant \|\boldsymbol{\theta}_t - \Phi(\boldsymbol{\theta}_t)\| - C_2 \eta \rho.$$

**Lemma E.9** (Lemma H.10 in Wen et al. (2023a))**.** *For any absolute constant $C > 0$, there exists absoulte constant $C_3$ such that: if $\boldsymbol{\theta}_t \in \mathbb{B}(K; R)$ and $\|\boldsymbol{\theta}_t - \Phi(\boldsymbol{\theta}_t)\| \leqslant C\rho$, then we have that*

$$\left|\left\|\boldsymbol{\theta}_{t+1/2} - \Phi(\boldsymbol{\theta}_{t+1/2})\right\| - \|\boldsymbol{\theta}_t - \Phi(\boldsymbol{\theta}_t)\|\right| \leqslant C_3 \eta \rho.$$

Now we fix the positive number $C = 1$ and use the absolute constants $C_2, C_3$, defined in Lemma E.8 and Lemma E.9.

## E.2 Proof of Theorem 5.6

### E.2.1 Proof of Moving Near Minimizers for SAM-IRE

This proof is similar to the proof for Section 5.2.1.

Under the conditions in Theorem 5.6, the update rule of IRE on standard SAM is

$$
\begin{aligned}
\boldsymbol{\theta}_{t+1} = & \boldsymbol{\theta}_t - \eta \nabla \mathcal{L}_{i_t}\left(\boldsymbol{\theta}_t + \rho \frac{\nabla \mathcal{L}_{i_t}(\boldsymbol{\theta}_t)}{\|\nabla \mathcal{L}_{i_t}(\boldsymbol{\theta}_t)\|}\right) \\
& - \eta \kappa P_{n+1:p}\left(\nabla^2 \mathcal{L}(\boldsymbol{\theta}_t)\right) \nabla \mathcal{L}_{i_t}\left(\boldsymbol{\theta}_t + \rho \frac{\nabla \mathcal{L}_{i_t}(\boldsymbol{\theta}_t)}{\|\nabla \mathcal{L}_{i_t}(\boldsymbol{\theta}_t)\|}\right), \quad \text{where } i_t \sim \mathbb{U}([n]).
\end{aligned}
$$

Let the $\kappa$ in IRE satisfy

$$\kappa \leqslant 1/\rho.$$

Additionally, we fix a constant $\alpha \in (0, 1)$ in the proof.

For simplicity, we denote

$$\boldsymbol{v}_t := \frac{\nabla \mathcal{L}_{i_t}(\boldsymbol{\theta}_t)}{\|\nabla \mathcal{L}_{i_t}(\boldsymbol{\theta}_t)\|}, \quad P(\boldsymbol{\theta}_t) := P_{n+1:p}\left(\nabla^2 \mathcal{L}(\boldsymbol{\theta}_t)\right);$$

and

$$
\begin{aligned}
\boldsymbol{\theta}_{t+1/2} = & \boldsymbol{\theta}_t - \nabla \mathcal{L}_{i_t}\left(\boldsymbol{\theta}_t + \rho \boldsymbol{v}_t\right); \\
\boldsymbol{\theta}_{t+1} = & \boldsymbol{\theta}_{t+1/2} - \kappa \eta P(\boldsymbol{\theta}_t) \nabla \mathcal{L}_{i_t}\left(\boldsymbol{\theta}_t + \rho \boldsymbol{v}_t\right).
\end{aligned}
$$

Additionally, we denote

$$\mathbb{P}_{t+1,t} := \mathbb{P}\left(\|\boldsymbol{\theta}_{t+1} - \Phi(\boldsymbol{\theta}_{t+1})\| \geqslant 2C\eta^{1-\alpha}\rho \,\middle|\, \|\boldsymbol{\theta}_t - \Phi(\boldsymbol{\theta}_t)\| < C\eta^{1-\alpha}\rho\right),$$

$$
\begin{aligned}
\mathbb{P}_{t+1,s} := & \mathbb{P}\Big(\|\boldsymbol{\theta}_{t+1} - \Phi(\boldsymbol{\theta}_{t+1})\| \geqslant 2C\eta^{1-\alpha}\rho; \\
& \forall \tau \in [s+1, t], C\eta^{1-\alpha}\rho \leqslant \|\boldsymbol{\theta}_\tau - \Phi(\boldsymbol{\theta}_\tau)\| < 2C\eta^{1-\alpha}\rho \,\Big|\, \|\boldsymbol{\theta}_s - \Phi(\boldsymbol{\theta}_s)\| < C\sqrt{\eta}\rho\Big), \quad s \in [T_{\mathrm{I}}, t-1].
\end{aligned}
$$

Then the following bound holds naturally:

$$\mathbb{P}\left(\exists\, t \in [T_{\mathrm{I}}, T_{\mathrm{I}} + T_{\mathrm{II}}], \|\boldsymbol{\theta}_t - \Phi(\boldsymbol{\theta}_t)\| \geqslant 2C\eta^{1-\alpha}\rho\right) \leqslant \sum_{t=T_{\mathrm{I}}}^{T_{\mathrm{I}}+T_{\mathrm{II}}-1}\left(\mathbb{P}_{t+1,t} + \sum_{s=T_{\mathrm{I}}}^{t-1} \mathbb{P}_{t+1,s}\right).$$

- Step I. Bounding $\mathbb{P}_{t+1,t}$.
  From $\|\boldsymbol{\theta}_t - \Phi(\boldsymbol{\theta}_t)\| = \mathcal{O}(\eta^{1-\alpha}\rho)$, thus

$$\|\boldsymbol{\theta}_t + \rho \boldsymbol{v}_t - \Phi(\boldsymbol{\theta}_t)\| \leqslant \|\boldsymbol{\theta}_t - \Phi(\boldsymbol{\theta}_t)\| + \rho = \mathcal{O}(\eta^{1-\alpha}\rho) + \mathcal{O}(\rho) < R,$$

which means $\boldsymbol{\theta}_t + \rho \boldsymbol{v}_t \in \mathbb{B}(K; R)$. Using Lemma E.7 and $\kappa \leqslant 1/\rho$, we can estimate:

$$\left\| \boldsymbol{\theta}_{t+1} - \boldsymbol{\theta}_{t+1/2} \right\| = \eta\kappa \left\| \nabla P(\boldsymbol{\theta}_t) \nabla \mathcal{L}_{i_t} (\boldsymbol{\theta}_t + \rho \boldsymbol{v}_t) \right\|$$

$$\leqslant \eta\kappa \left( \left\| \nabla P(\boldsymbol{\theta}_t) \nabla \mathcal{L}_{i_t}(\boldsymbol{\theta}_t) \right\| + \rho \left\| \nabla P(\boldsymbol{\theta}_t) \nabla^2 \mathcal{L}_{i_t}(\boldsymbol{\theta}_t) \right\| + \frac{\beta_3}{2}\rho^2 \right)$$

$$\leqslant \eta\kappa \left( \eta\rho^2 + \eta^{1-\alpha}\rho^2 + \frac{\beta_3}{2}\rho^2 \right) \leqslant \beta_3 \eta\kappa\rho^2 \leqslant \frac{C_2}{2(1 + \zeta_1^\Phi)}\eta\rho,$$

$$\left\| \boldsymbol{\theta}_{t+1} - \Phi(\boldsymbol{\theta}_{t+1}) - (\boldsymbol{\theta}_{t+1/2} - \Phi(\boldsymbol{\theta}_{t+1/2})) \right\|$$

$$\leqslant (1 + \zeta_1^\Phi) \left\| \boldsymbol{\theta}_{t+1} - \boldsymbol{\theta}_{t+1/2} \right\| \leqslant \frac{C_2}{2}\eta\rho.$$

Then we have the following bound:

$$\left\| \boldsymbol{\theta}_{t+1} - \Phi(\boldsymbol{\theta}_{t+1}) \right\|$$

$$\leqslant \left\| \boldsymbol{\theta}_t - \Phi(\boldsymbol{\theta}_t) \right\| + \left\| \boldsymbol{\theta}_{t+1/2} - \Phi(\boldsymbol{\theta}_{t+1/2}) - (\boldsymbol{\theta}_t - \Phi(\boldsymbol{\theta}_t)) \right\|$$

$$+ \left\| \boldsymbol{\theta}_{t+1} - \Phi(\boldsymbol{\theta}_{t+1}) - (\boldsymbol{\theta}_{t+1/2} - \Phi(\boldsymbol{\theta}_{t+1/2})) \right\|$$

$$\overset{\text{Lemma E.9}}{\leqslant} C_1 \eta^{1-\alpha}\rho + \mathcal{O}(\eta\rho) + \left\| \boldsymbol{\theta}_{t+1} - \Phi(\boldsymbol{\theta}_{t+1}) - (\boldsymbol{\theta}_{t+1/2} - \Phi(\boldsymbol{\theta}_{t+1/2})) \right\|$$

$$\leqslant C_1 \eta^{1-\alpha}\rho + \mathcal{O}(\eta\rho) + \mathcal{O}(\eta\rho) < \frac{3C_1}{2}\eta^{1-\alpha}\rho.$$

Thus, we obtain

$$\mathbb{P}_{t+1,t} = \mathbb{P}\left( \left\| \boldsymbol{\theta}_{\tau+1} - \Phi(\boldsymbol{\theta}_{\tau+1}) \right\| \geqslant 2C\eta^{1-\alpha}\rho \,\middle|\, \left\| \boldsymbol{\theta}_t - \Phi(\boldsymbol{\theta}_t) \right\| < C\eta^{1-\alpha}\rho \right) = 0.$$

- **Step II. Bounding $\mathbb{P}_{t+1,s}$ for $s \in [T_\mathrm{I}, t-1]$.**
  We prove this step under the condition $\left\| \boldsymbol{\theta}_s - \Phi(\boldsymbol{\theta}_s) \right\| < C\eta^{1-\alpha}\rho$. Define a process $\{X_\tau\}_{\tau=s}^{t+1}$:
  $X_{s+1} = \left\| \boldsymbol{\theta}_{s+1} - \Phi(\boldsymbol{\theta}_{s+1}) \right\|$,

$$X_{\tau+1} = \begin{cases} \left\| \boldsymbol{\theta}_{\tau+1} - \Phi(\boldsymbol{\theta}_{\tau+1}) \right\|, & \text{if } C\eta^{1-\alpha}\rho \leqslant X_\tau = \left\| \boldsymbol{\theta}_\tau - \Phi(\boldsymbol{\theta}_\tau) \right\| \leqslant 2C\eta^{1-\alpha}\rho \\ X_\tau - C_2\eta\rho/2, & \text{else} \end{cases}.$$

It is clear that

$$\mathbb{P}_{t+1,s} \leqslant \mathbb{P}\left( X_{t+1} \geqslant 2C\eta^{1-\alpha}\rho \right).$$

Then our key step is to prove the following two claims about the process $\{X_\tau\}$.

- Claim I. $X_\tau - C_2\tau\eta\rho/2$ is a super-martingale. From the definition of $X_\tau$, we only need to prove that if $C\eta^{1-\alpha}\rho \leqslant X_\tau = \left\| \boldsymbol{\theta}_\tau - \Phi(\boldsymbol{\theta}_\tau) \right\| \leqslant 2C\eta^{1-\alpha}\rho$, then $\mathbb{E}\left\| \boldsymbol{\theta}_{\tau+1} - \Phi(\boldsymbol{\theta}_{\tau+1}) \right\| \leqslant \left\| \boldsymbol{\theta}_\tau - \Phi(\boldsymbol{\theta}_\tau) \right\| - C_2\eta\rho/2$.
  If $C\eta^{1-\alpha}\rho \leqslant X_\tau = \left\| \boldsymbol{\theta}_\tau - \Phi(\boldsymbol{\theta}_\tau) \right\| \leqslant 2C\eta^{1-\alpha}\rho$, similar to Step I, it holds that $\boldsymbol{\theta}_{\tau+1} \in \mathbb{B}(K; R)$ and $\left\| \boldsymbol{\theta}_{t+1} - \Phi(\boldsymbol{\theta}_{\tau+1}) - (\boldsymbol{\theta}_{\tau+1/2} - \Phi(\boldsymbol{\theta}_{\tau+1/2})) \right\| \leqslant \frac{C_2}{2}\eta\rho$. Moreover,

$$\left\| \boldsymbol{\theta}_{\tau+1} - \Phi(\boldsymbol{\theta}_{\tau+1}) \right\|$$

$$\leqslant \left\| \boldsymbol{\theta}_{\tau+1/2} - \Phi(\boldsymbol{\theta}_{\tau+1/2}) \right\| + \left\| \boldsymbol{\theta}_{\tau+1} - \Phi(\boldsymbol{\theta}_{\tau+1}) - (\boldsymbol{\theta}_{\tau+1/2} - \Phi(\boldsymbol{\theta}_{\tau+1/2})) \right\|$$

$$\leqslant \left\| \boldsymbol{\theta}_{\tau+1/2} - \Phi(\boldsymbol{\theta}_{\tau+1/2}) \right\| + \frac{C_2}{2}\eta\rho.$$

Taking the expectation and using Lemma E.8, we have

$$\mathbb{E}\left\| \boldsymbol{\theta}_{\tau+1} - \Phi(\boldsymbol{\theta}_{\tau+1}) \right\| \leqslant \mathbb{E}\left\| \boldsymbol{\theta}_{\tau+1/2} - \Phi(\boldsymbol{\theta}_{\tau+1/2}) \right\| + \frac{C_2}{2}\eta\rho$$

$$\leqslant \left\| \boldsymbol{\theta}_\tau - \Phi(\boldsymbol{\theta}_\tau) \right\| - C_2\eta\rho + \frac{C_2}{2}\eta\rho = \left\| \boldsymbol{\theta}_\tau - \Phi(\boldsymbol{\theta}_\tau) \right\| - \frac{C_2}{2}\eta\rho.$$

- Claim II. $X_{\tau+1} - X_\tau + C_2\eta\rho/2$ is $\mathcal{O}(\eta\rho)$-bounded. From the definition of $X_\tau$, we only need to prove for the case $C\eta^{1-\alpha}\rho \leqslant X_\tau = \|\boldsymbol{\theta}_\tau - \Phi(\boldsymbol{\theta}_\tau)\| \leqslant 2C\eta^{1-\alpha}\rho$.

  If $C\eta^{1-\alpha}\rho \leqslant X_\tau = \|\boldsymbol{\theta}_\tau - \Phi(\boldsymbol{\theta}_\tau)\| \leqslant 2C\eta^{1-\alpha}\rho$, we have $\boldsymbol{\theta}_{\tau+1} \in \mathbb{B}(K;R)$ and $\|\boldsymbol{\theta}_{t+1} - \Phi(\boldsymbol{\theta}_{t+1}) - (\boldsymbol{\theta}_{t+1/2} - \Phi(\boldsymbol{\theta}_{t+1/2}))\| \leqslant \frac{C_2}{2}\eta\rho$. Combining this result and Lemma E.9, we have

$$\left| \|\boldsymbol{\theta}_{\tau+1} - \Phi(\boldsymbol{\theta}_{\tau+1})\| - \|\boldsymbol{\theta}_\tau - \Phi(\boldsymbol{\theta}_\tau)\| \right|$$

$$\leqslant \left| \|\boldsymbol{\theta}_{\tau+1} - \Phi(\boldsymbol{\theta}_{\tau+1})\| - \|\boldsymbol{\theta}_{\tau+1/2} - \Phi(\boldsymbol{\theta}_{\tau+1/2})\| \right| + \left| \|\boldsymbol{\theta}_{\tau+1/2} - \Phi(\boldsymbol{\theta}_{\tau+1/2})\| - \|\boldsymbol{\theta}_\tau - \Phi(\boldsymbol{\theta}_\tau)\| \right|$$

$$\leqslant \left\| (\boldsymbol{\theta}_{\tau+1} - \Phi(\boldsymbol{\theta}_{\tau+1})) - (\boldsymbol{\theta}_{\tau+1/2} - \Phi(\boldsymbol{\theta}_{\tau+1/2})) \right\| + \left| \|\boldsymbol{\theta}_{\tau+1/2} - \Phi(\boldsymbol{\theta}_{\tau+1/2})\| - \|\boldsymbol{\theta}_\tau - \Phi(\boldsymbol{\theta}_\tau)\| \right|$$

$$\leqslant \frac{C_2}{2}\eta\rho + C_3\eta\rho = \mathcal{O}(\eta\rho).$$

With the preparation of Claim I and Claim II, we can use the Azuma-Hoeffeding inequality: for any $Q > 0$, it holds that

$$\mathbb{P}\left(X_{t+1} - X_{s+1} + (t-s)C_2\eta\rho/2 > Q\right) \leqslant 2\exp\left(-\frac{Q^2}{2(t-s)\mathcal{O}(\eta^2\rho^2)}\right).$$

As proved in Claim I, $X_{s+1} = \|\boldsymbol{\theta}_{s+1} - \Phi(\boldsymbol{\theta}_{s+1})\| \leqslant \frac{3}{2}C\eta^{1-\alpha}\rho$ due to $\|\boldsymbol{\theta}_s - \Phi(\boldsymbol{\theta}_s)\| \leqslant C\eta^{1-\alpha}\rho$. Therefore, by choosing $Q = (t-s)C_2\eta\rho/2 - \frac{3}{2}C\eta^{1-\alpha}\rho + 2C\eta^{1-\alpha}\rho = (t-s)C_2\eta\rho/2 + C\eta^{1-\alpha}\rho/2$, we have

$$\mathbb{P}_{t+1,s} \leqslant \mathbb{P}\left(X_{t+1} \geqslant 2C\eta^{1-\alpha}\rho\right)$$

$$\leqslant \mathbb{P}\left(X_{t+1} - X_{s+1} + (t-s)C_2\eta\rho/2 > (t-s)\frac{C_2}{2}\eta\rho + \frac{C}{2}\eta^{1-\alpha}\rho\right)$$

$$\leqslant 2\exp\left(-\frac{\left((t-s)C_2\eta\rho/2 + C\eta^{1-\alpha}\rho/2\right)^2}{2(t-s)\mathcal{O}(\eta^2\rho^2)}\right)$$

$$\leqslant 2\exp\left(-\frac{(t-s)C_2\eta\rho \cdot C\eta^{1-\alpha}\rho}{4(t-s)\mathcal{O}(\eta^2\rho^2)}\right) \leqslant 2\exp\left(-\Omega\left(\frac{1}{\eta^\alpha}\right)\right).$$

Therefore, we obtain the union bound:

$$\mathbb{P}\left(\exists\, t \in [T_{\mathrm{I}}, T_{\mathrm{I}} + T_{\mathrm{II}}], \|\boldsymbol{\theta}_t - \Phi(\boldsymbol{\theta}_t)\| \geqslant 2C\eta^{1-\alpha}\rho\right) \leqslant \sum_{t=T_{\mathrm{I}}}^{T_{\mathrm{I}}+T_{\mathrm{II}}-1}\left(\mathbb{P}_{t+1,t} + \sum_{s=T_{\mathrm{I}}}^{t-1}\mathbb{P}_{t+1,s}\right)$$

$$\leqslant \sum_{t=T_I}^{T_{\mathrm{I}}+T_{\mathrm{II}}-1}\sum_{s=T_{\mathrm{I}}}^{t-1}\mathbb{P}_{t+1,s} \leqslant T_{\mathrm{II}}^2\exp\left(-\Omega\left(1/\eta^\alpha\right)\right).$$

### E.2.2 Proof of the Effective Dynamics

By our proof above, with probability at least $1 - T_{\mathrm{II}}^2\exp\left(-\Omega\left(1/\eta^\alpha\right)\right)$, for any $t \in [T_{\mathrm{I}}, T_{\mathrm{I}} + T_{\mathrm{II}}]$, $\|\boldsymbol{\theta}_t - \Phi(\boldsymbol{\theta}_t)\| = \mathcal{O}(\eta^{1-\alpha}\rho)$. Then we prove this theorem when the above event occurs.

Due to $\|\boldsymbol{\theta}_t - \Phi(\boldsymbol{\theta}_t)\| = \mathcal{O}(\eta^{1-\alpha}\rho)$, we have:

$$\|\boldsymbol{\theta}_{t+1} - \boldsymbol{\theta}_t\|$$
$$\leqslant \eta\|\nabla\mathcal{L}_{i_t}(\boldsymbol{\theta}_t + \rho\boldsymbol{v}_t)\| + \eta\kappa\|P(\boldsymbol{\theta}_t)\nabla\mathcal{L}_{i_t}(\boldsymbol{\theta}_t + \rho\boldsymbol{v}_t)\|$$
$$\overset{\text{Lemma E.7}}{\leqslant} \eta\|\nabla\mathcal{L}_{i_t}(\boldsymbol{\theta}_t)\| + \mathcal{O}(\eta\rho) + \mathcal{O}(\eta\kappa\rho^2) \leqslant \mathcal{O}(\eta\rho).$$

Then by Taylor's expansion,

$$\Phi(\boldsymbol{\theta}_{t+1}) - \Phi(\boldsymbol{\theta}_t) = \partial\Phi(\boldsymbol{\theta}_t)(\boldsymbol{\theta}_{t+1} - \boldsymbol{\theta}_t) + \mathcal{O}\left(\|\boldsymbol{\theta}_{t+1} - \boldsymbol{\theta}_t\|^2\right)$$

$$= -\eta\partial\Phi(\boldsymbol{\theta}_t)\nabla\mathcal{L}_{i_t}(\boldsymbol{\theta}_t + \rho\boldsymbol{v}_t) - \eta\kappa\partial\Phi(\boldsymbol{\theta}_t)P(\boldsymbol{\theta}_t)\nabla\mathcal{L}_{i_t}(\boldsymbol{\theta}_t + \rho\boldsymbol{v}_t) + \mathcal{O}(\eta^2\rho^2).$$

For the term $\partial\Phi(\boldsymbol{\theta}_t)\nabla\mathcal{L}_{i_t}(\boldsymbol{\theta}_t + \rho\boldsymbol{v}_t)$ and $\partial\Phi(\boldsymbol{\theta}_t)P(\boldsymbol{\theta}_t)\nabla\mathcal{L}_{i_t}(\boldsymbol{\theta}_t + \rho\boldsymbol{v}_t)$, using Taylor's expansion and Lemma E.7, we have

$$\partial\Phi(\boldsymbol{\theta}_t)\nabla\mathcal{L}_{i_t}(\boldsymbol{\theta}_t + \rho\boldsymbol{v}_t)$$

$$
\begin{aligned}
=&\partial\Phi(\boldsymbol{\theta}_t)\nabla\mathcal{L}_{i_t}(\boldsymbol{\theta}_t) + \rho\partial\Phi(\boldsymbol{\theta}_t)\nabla^2\mathcal{L}_{i_t}(\boldsymbol{\theta}_t)\boldsymbol{v}_t + \frac{\rho^2}{2}\partial\Phi(\boldsymbol{\theta}_t)\nabla\operatorname{Tr}\left(\boldsymbol{v}_t\nabla^2\mathcal{L}_{i_t}(\boldsymbol{\theta}_t)\boldsymbol{v}_t^\top\right) + \mathcal{O}(\rho^3)\\
=&\partial\Phi(\boldsymbol{\theta}_t)\nabla\mathcal{L}_{i_t}(\boldsymbol{\theta}_t) + \rho\partial\Phi(\boldsymbol{\theta}_t)\nabla^2\mathcal{L}_{i_t}(\boldsymbol{\theta}_t)\boldsymbol{v}_t + \frac{\rho^2}{2}\partial\Phi(\boldsymbol{\theta}_t)\nabla\left(\boldsymbol{v}_t^\top\nabla^2\mathcal{L}_{i_t}(\boldsymbol{\theta}_t)\boldsymbol{v}_t\right) + \mathcal{O}(\rho^3)\\
=&\mathcal{O}(\|\boldsymbol{\theta}_t - \Phi(\boldsymbol{\theta}_t)\|^2) + \rho\partial\Phi(\Phi(\boldsymbol{\theta}_t))\nabla^2\mathcal{L}_{i_t}(\Phi(\boldsymbol{\theta}_t))\frac{\nabla\mathcal{L}_{i_t}(\Phi(\boldsymbol{\theta}_t))}{\|\nabla\mathcal{L}_{i_t}(\Phi(\boldsymbol{\theta}_t))\|} + \mathcal{O}(\rho\|\boldsymbol{\theta}_t - \Phi(\boldsymbol{\theta}_t)\|)\\
&+ \frac{\rho^2}{2}\partial\Phi(\Phi(\boldsymbol{\theta}_t))\nabla\left(\frac{\nabla\mathcal{L}_{i_t}(\Phi(\boldsymbol{\theta}_t))}{\|\nabla\mathcal{L}_{i_t}(\Phi(\boldsymbol{\theta}_t))\|}^\top \nabla^2\mathcal{L}_{i_t}(\Phi(\boldsymbol{\theta}_t))\frac{\nabla\mathcal{L}_{i_t}(\Phi(\boldsymbol{\theta}_t))}{\|\nabla\mathcal{L}_{i_t}(\Phi(\boldsymbol{\theta}_t))\|}\right)\\
&+ \mathcal{O}(\rho^2\|\boldsymbol{\theta}_t - \Phi(\boldsymbol{\theta}_t)\|) + \mathcal{O}(\rho^3)\\
=&\frac{\rho^2}{2}\partial\Phi(\Phi(\boldsymbol{\theta}_t))\nabla\lambda_1\left(\nabla^2\mathcal{L}_{i_t}(\Phi(\boldsymbol{\theta}_t))\right) + \mathcal{O}(\eta^{1-\alpha}\rho^2 + \rho^3),
\end{aligned}
$$

and

$$
\begin{aligned}
&\partial\Phi(\boldsymbol{\theta}_t)P(\boldsymbol{\theta}_t)\nabla\mathcal{L}_{i_t}(\boldsymbol{\theta}_t + \rho\boldsymbol{v}_t)\\
=&\partial\Phi(\boldsymbol{\theta}_t)P(\boldsymbol{\theta}_t)\nabla\mathcal{L}_{i_t}(\boldsymbol{\theta}_t) + \rho\partial\Phi(\boldsymbol{\theta}_t)P(\boldsymbol{\theta}_t)\nabla^2\mathcal{L}_{i_t}(\boldsymbol{\theta}_t)\boldsymbol{v}_t\\
&+ \frac{\rho^2}{2}\partial\Phi(\boldsymbol{\theta}_t)P(\boldsymbol{\theta}_t)\nabla\operatorname{Tr}\left(\boldsymbol{v}_t\nabla^2\mathcal{L}_{i_t}(\boldsymbol{\theta}_t)\boldsymbol{v}_t^\top\right) + \mathcal{O}(\rho^3)\\
=&\mathcal{O}(\|\boldsymbol{\theta}_t - \Phi(\boldsymbol{\theta}_t)\|^2) + \mathcal{O}(\rho\|\boldsymbol{\theta}_t - \Phi(\boldsymbol{\theta}_t)\|) + \frac{\rho^2}{2}\partial\Phi(\boldsymbol{\theta}_t)P(\boldsymbol{\theta}_t)\nabla\left(\boldsymbol{v}_t^\top\nabla^2\mathcal{L}_{i_t}(\boldsymbol{\theta}_t)\boldsymbol{v}_t\right) + \mathcal{O}(\rho^3)\\
=&\mathcal{O}(\eta^{1-\alpha}\rho^2) + \frac{\rho^2}{2}\partial\Phi(\Phi(\boldsymbol{\theta}_t))\nabla\left(\frac{\nabla\mathcal{L}_{i_t}(\Phi(\boldsymbol{\theta}_t))}{\|\nabla\mathcal{L}_{i_t}(\Phi(\boldsymbol{\theta}_t))\|}^\top \nabla^2\mathcal{L}_{i_t}(\Phi(\boldsymbol{\theta}_t))\frac{\nabla\mathcal{L}_{i_t}(\Phi(\boldsymbol{\theta}_t))}{\|\nabla\mathcal{L}_{i_t}(\Phi(\boldsymbol{\theta}_t))\|}\right)\\
&+ \mathcal{O}(\rho^2\|\boldsymbol{\theta}_t - \Phi(\boldsymbol{\theta}_t)\|) + \mathcal{O}(\rho^3)\\
=&\frac{\rho^2}{2}\partial\Phi(\Phi(\boldsymbol{\theta}_t))\nabla\lambda_1\left(\nabla^2\mathcal{L}_{i_t}(\Phi(\boldsymbol{\theta}_t))\right) + \mathcal{O}(\eta^{1-\alpha}\rho^2 + \rho^3),.
\end{aligned}
$$

Combining the results above, we obtain:

$$
\Phi(\boldsymbol{\theta}_{t+1}) = \Phi(\boldsymbol{\theta}_t) - (1+\kappa)\frac{\eta\rho^2}{2}\partial\Phi(\Phi(\boldsymbol{\theta}_t))\nabla\lambda_1\left(\nabla^2\mathcal{L}_{i_t}(\Phi(\boldsymbol{\theta}_t))\right) + \mathcal{O}(\kappa\eta^{2-\alpha}\rho^2 + \kappa\eta\rho^3).
$$

Additionally, taking the expectation, we have:

$$
\mathbb{E}_{i_t}\left[\partial\Phi(\Phi(\boldsymbol{\theta}_t))\nabla\lambda_1\left(\nabla^2\mathcal{L}_{i_t}(\Phi(\boldsymbol{\theta}_t))\right)\right] = \partial\Phi(\Phi(\boldsymbol{\theta}_t))\nabla\operatorname{Tr}\left(\nabla^2\mathcal{L}(\Phi(\boldsymbol{\theta}_t))\right).
$$

Therefore, we obtain

$$
\begin{aligned}
\mathbb{E}_{i_t}\left[\Phi(\boldsymbol{\theta}_{t+1})\right] =&\Phi(\boldsymbol{\theta}_t) - (1+\kappa)\frac{\eta\rho^2}{2}\partial\Phi(\Phi(\boldsymbol{\theta}_t))\nabla\operatorname{Tr}\left(\nabla^2\mathcal{L}(\Phi(\boldsymbol{\theta}_t))\right) + \mathcal{O}(\kappa\eta^{2-\alpha}\rho^2 + \kappa\eta\rho^3)\\
=&\Phi(\boldsymbol{\theta}_t) - (1+\kappa)\frac{\eta\rho^2}{2}\partial\Phi(\Phi(\boldsymbol{\theta}_t))\nabla\operatorname{Tr}\left(\nabla^2\mathcal{L}(\Phi(\boldsymbol{\theta}_t))\right) + h.o.t..
\end{aligned}
$$

# F Useful Inequalities

**Definition F.1** ($\mu$-PL). Let $\mu > 0$ be a constant. A function $\mathcal{L}$ is $\mu$-PL in a set $U$ iff $\|\nabla\mathcal{L}(\boldsymbol{\theta})\|^2 \geqslant 2\mu(\mathcal{L}(\boldsymbol{\theta}) - \inf_{\boldsymbol{\theta}\in U}\mathcal{L}(\boldsymbol{\theta})), \forall\boldsymbol{\theta}\in U$.

**Lemma F.2** (Weyl Theorem). *Let $\boldsymbol{A}, \boldsymbol{B} \in \mathbb{R}^{p\times p}$ be symmetric with eigenvalues $\lambda_1 \geqslant \cdots \geqslant \lambda_p$ and $\mu_1 \geqslant \cdots \geqslant \mu_p$ respectively, then for any $k \in [p]$, it holds that*

$$
|\lambda_k - \mu_k| \leqslant \|\boldsymbol{A} - \boldsymbol{B}\|.
$$

**Lemma F.3** (Davis-Kahan $\sin(\Theta)$ theorem). *Let $\boldsymbol{A}, \boldsymbol{B} \in \mathbb{R}^{p \times p}$ be symmetric matrices. Denote their orthogonal decomposition as $\boldsymbol{A} = \boldsymbol{E}_1 \boldsymbol{\Lambda}_1 \boldsymbol{E}_1^\top + \boldsymbol{E}_2 \boldsymbol{\Lambda}_2 \boldsymbol{E}_2^\top$ and $\boldsymbol{B} = \boldsymbol{F}_1 \boldsymbol{\Gamma}_1 \boldsymbol{F}_1^\top + \boldsymbol{F}_2 \boldsymbol{\Gamma}_2 \boldsymbol{F}_2^\top$ with $(\boldsymbol{E}_1, \boldsymbol{E}_2)$ and $(\boldsymbol{D}_1, \boldsymbol{D}_2)$ orthogonal. If the eigenvalues in $\boldsymbol{\Lambda}_1$ are contained in an interval $(a, b)$, and the eigenvalues of $\boldsymbol{\Gamma}_2$ are excluded from the interval $(a - \delta, b + \delta)$ for some $\delta > 0$, then for any unitarily invariant norm $\|\cdot\|_\star$,*

$$\left\| \boldsymbol{F}_2^\top \boldsymbol{E}_1 \right\|_\star \leqslant \frac{\left\| \boldsymbol{F}_2^\top (\boldsymbol{A} - \boldsymbol{B}) \boldsymbol{E}_1 \right\|_\star}{\delta}.$$

**Lemma F.4** (Azuma-Hoeffding Inequality). *Suppose $\{X_n\}_{n \in \mathbb{N}}$ is a super-martingale.*

- *(i) (Bounded martingale difference). If $-\alpha \leqslant X_{i+1} - X_i \leqslant \beta$, then for any $n, t > 0$, we have:*

$$\mathbb{P}\left( X_n - X_0 \geqslant t \right) \leqslant 2 \exp\left( -\frac{t^2}{2n(\alpha + \beta)^2} \right).$$

- *(ii) (Sub-Gaussian martingale difference). If $X_{i+1} - X_i$ is $\sigma_i^2$-sub-Gaussian, then for any $n, t > 0$, we have:*

$$\mathbb{P}\left( X_n - X_0 \geqslant t \right) \leqslant 2 \exp\left( -\frac{t^2}{2 \sum_{i=1}^{n} \sigma_i^2} \right).$$

**Lemma F.5.** *Let $\boldsymbol{v} \in \mathbb{R}^p$. Let $\boldsymbol{g} \sim \mathcal{N}(\boldsymbol{0}, \boldsymbol{I})$. Then there exists an absolute constant $c > 0$ such that for any $t > 0$,*

$$\mathbb{P}\left( \left| \left\langle \frac{\boldsymbol{v}}{\|\boldsymbol{v}\|}, \frac{\boldsymbol{g}}{\|\boldsymbol{g}\|} \right\rangle \right| \geqslant t \right) \leqslant 4 e^{-cpt^2}.$$

*Proof of Lemma F.5.* From P54 in Vershynin (2018), there exists an absolute constant $c > 0$ such that for any $t > 0$, $\mathbb{P}\left( \frac{|\langle \boldsymbol{e}_1, \boldsymbol{g} \rangle|}{\|\boldsymbol{g}\|} \geqslant t \right) \leqslant 4 e^{-cpt^2}$. Without loss of generality, we can assume $\boldsymbol{v} \neq \boldsymbol{0}$. Then we have:

$$\mathbb{P}\left( \left| \left\langle \frac{\boldsymbol{v}}{\|\boldsymbol{v}\|}, \frac{\boldsymbol{g}}{\|\boldsymbol{g}\|} \right\rangle \right| \geqslant \frac{t}{\sqrt{p}} \right) = \mathbb{P}\left( \left| \frac{\langle \boldsymbol{e}_1, \boldsymbol{g} \rangle}{\|\boldsymbol{g}\|} \right| \geqslant t \right) \leqslant 4 e^{-cpt^2}.$$

$\square$

**Lemma F.6.** $\|\boldsymbol{A}\boldsymbol{B}\|_{\mathrm{F}} \leqslant \|\boldsymbol{A}\| \|\boldsymbol{B}\|_{\mathrm{F}}$.

