# OpenReview forum: "Improving Generalization and Convergence by Enhancing Implicit Regularization"
_NeurIPS.cc/2024/Conference — NeurIPS 2024 poster_

### Official Review · Reviewer_Jrgs · 2024-07-05

**Soundness:** 3
**Presentation:** 3
**Contribution:** 3
**Rating:** 6
**Confidence:** 4

**Summary:**

This paper proposes a new optimization scheme for deep learning.  The idea is to periodically estimate the Hessian diagonal and use a larger learning rate on the smaller-curvature parameters.   The paper argues that this algorithm enhances the implicit curvature regularization of the optimization algorithm while not degrading stability (which is what would happen if you used a larger learning rate on all parameters).  The paper first motivates the scheme by analyzing a toy setting.  The paper then shows empirically that adding this scheme to SGD and to SAM results in better generalizing and lower-curvature solutions, where curvature is quantified using Hessian trace.   The paper then shows that adding this scheme to Adam accelerates the _optimization_ of Llama transformers, and leaves an explanation why to future work.  Finally, the paper theoretically analyzes this scheme for SAM in the manifold-of-global-minima setting, and proves that this scheme accelerates the drift towards flatter regions of the manifold.

**Strengths:**

I think the idea is novel, and the experiments are promising, so this paper could stimulate significant research in the future.

**Weaknesses:**

One weakness is that the motivation comes from the stylized example in section 2, which may be unrealistic for general deep learning optimization.  For example, the idea of "sharp coordinates" only makes sense in this stylized example.

**Questions:**

Shouldn't this algorithm also speed up _optimization_?  The algorithm uses a larger learning rate on some coordinates, which should also have an optimization effect.  (This would not be visible in the analysis setting of manifold-of-global-minima because no optimization is occurring in that setting.)

How would you define IRE at the most abstract level?  If we define an abstract version of preconditioned gradient descent as $\theta_{t+1} = \theta_t - P g_t$ (where vanilla GD with learning rate $\eta$ corresponds to $P = \eta I$), is the idea that it's better to use $P_2$ rather than $P_1$ provided that $P_2 \succeq P_1$ and both are locally stable i.e. $\lambda_{\max}(P H) \le 2$ for $P \in $ {$P_1, P_2$}?

**Limitations:**

Yes

---

> ### Author Rebuttal · Authors · 2024-08-04
>
> We thank the reviewer for the appreciation of our work and insightful comments. Below, we address the reviewer’s questions in detail.
>
> - **W1.** One weakness is that the motivation comes from the stylized example in section 2, which may be unrealistic for general deep learning optimization. For example, the idea of "sharp coordinates" only makes sense in this stylized example.
>
>   **Response:** We thank the reviewer for this insightful question.
>
>   - **Clarification.** We clarify that the idea behind IRE is to distinguish between "sharp/flat directions" rather than "sharp/flat coordinates". The  use of a diagonal Hessian in Section 2  is only for simplification purpose. In fact,  the theoretical analysis in Section 5 considers a more general setting  without assuming a diagonal Hessian (see Assumption 5.1).
>
>   - **Practical Considerations.** In the practical implementation described in Section 3, we indeed use the diagonal Hessian to approximate the full Hessian, aiming to reduce computational costs. The diagonal Hessian is commonly acknowledged as a reasonable approximation of the full Hessian in deep neural networks and is frequently employed in the design of deep learning optimizers, as noted in references [1][2]. For a more detailed discussion, please refer to lines 156-162 in our manuscript.
>
> - **Q1.** Shouldn't this algorithm also speed up optimization? The algorithm uses a larger learning rate on some coordinates, which should also have an optimization effect. (This would not be visible in the analysis setting of manifold-of-global-minima because no optimization is occurring in that setting.)
>
>   **Response:** We thank the reviewer for this insightful comment and will include a   discussion on this issue in our revised version. The motivation of IRE is to speed up the sharpness reduction, which only requires to increase learning rate along completely flat (zero-curvature) directions. However, practical implementation may also increase  learning rate along directions with small but non-zero curvatures, which can further speed up loss convergence. However, explaining why  this approach can provide a significant acceleration is  unclear.
>
> - **Q2.** How would you define IRE at the most abstract level? If we define an abstract version of preconditioned gradient descent as $\theta_{t+1}=\theta_t-Pg_t$ (where vanilla GD with learning rate $\eta$ corresponds to $P=\eta I$), is the idea that it's better to use $P_2$ rather than $P_1$ provided that $P_2\succeq P_1$ and both are locally stable i.e. $\lambda_{\max}(PH)<2$ for $P\in\{P_1,P_2\}$?
>
>   **Response:** We are grateful to the reviewer for sharing this high-level perspective, which can help  explain the acceleration of loss convergence observed with IRE. Given a base optimizer with update $\theta_{t+1}=\theta_t-P_1g_t$, we aim to improve it by selecting a better $P_2=QP_1$, while ensuring $\lambda_{\max}(P_2 H)<2$ to maintain training stability. Let $P_{flat}$ be the projection matrix into the flat (zero-curvature) directions. In our IRE approach, we choose $P_2=(I+\mu P_{flat}) P_1$, which meets the condition $P_2\succeq P_1$ mentioned by the reviewer. However, we remark that our original motivation of  this specific $P_2$ is to accelerate effective dynamics on the manifold of global minima, thereby accelerating the sharpness reduction.
>
>
> [1] George et al. Fast approximate natural gradient descent in a Kronecker-factored eigenbasis. (NeurIPS 2018)
>
> [2] Liu et al. Sophia: A Scalable Stochastic Second-order Optimizer for Language Model Pre-training. (ICLR 2024)

---

> > ### Author Response · Authors · 2024-08-12
> >
> > Thanks again for your valuable time and effort in reviewing our work!
> >
> > We are wondering if our responses address your questions or concerns.
> >
> > We are happy to try to address any other comments in the time remaining.

---

> > > ### Comment · Reviewer_Jrgs · 2024-08-12
> > > **reviewer response**
> > >
> > > Thanks, I don't have any more questions.

---

### Official Review · Reviewer_QWSd · 2024-07-13

**Soundness:** 2
**Presentation:** 2
**Contribution:** 2
**Rating:** 6
**Confidence:** 3

**Summary:**

This paper proposes an Implicit Regularization Enhancement (IRE) framework to accelerate the convergence of optimization algorithms towards flat minima in deep learning, thereby improving generalization and convergence. The key idea behind IRE is to decouple the dynamics along flat and sharp directions in the optimization landscape. It speeds up the dynamics along flat directions while keeping the dynamics in sharp directions unchanged. This allows IRE to substantially boost the implicit sharpness reduction of the base optimizer without hurting training stability.

The authors provide a practical way to efficiently incorporate IRE with generic base optimizers without introducing significant computational overhead. Extensive experiments across image classification and language modeling tasks demonstrate that IRE consistently improves the generalization performance of base optimizers like SGD, Adam, and SAM. Surprisingly, IRE also achieves a 2x speedup compared to a well-tuned AdamW optimizer in pre-training Llama language models of various sizes.

Furthermore, the paper provides theoretical guarantees showing that IRE can achieve a substantial acceleration over the base SAM algorithm in minimizing the trace of the Hessian matrix, which measures the flatness of the loss landscape.

In summary, the key contributions are:

1. The IRE framework that enhances implicit regularization by decoupling and accelerating optimization dynamics along flat directions
2. A practical and efficient implementation of IRE that can be incorporated into generic base optimizers
3. Extensive empirical results showing IRE improves generalization and speeds up convergence across vision and language tasks
4. Theoretical analysis proving IRE can substantially accelerate the sharpness reduction of the SAM optimizer

**Strengths:**

## Originality
The key idea behind the proposed Implicit Regularization Enhancement (IRE) framework - decoupling the optimization dynamics along flat and sharp directions to accelerate convergence to flat minima - appears to be quite novel and creative. To the best of my knowledge, this specific approach has not been explored in the optimization and generalization literature before. The authors motivate IRE through an intuitive illustrative example, showing how selectively increasing the learning rate along flat directions can speed up the implicit regularization. Adapting this core idea into a practical algorithm that can be efficiently incorporated into existing optimizers is also a notable contribution.

However, the paper could benefit from a more detailed discussion comparing and contrasting IRE with other approaches that aim to improve sharpness-aware minimization, such as those reducing the computational cost of SAM (e.g., Kwon et al., 2021; Liu et al., 2022). This would help further highlight the novelty and uniqueness of IRE.

## Quality
The paper presents a solid mix of conceptual explanations, practical algorithm design, extensive experiments, and theoretical analysis. The illustrative example in Section 2 provides a clear and intuitive understanding of the mechanism behind IRE. The practical implementation of IRE in Section 3, particularly the efficient approximation of the projection operator using diagonal Hessian estimates, demonstrates the authors' attention to computational efficiency.

The experimental results on image classification and language modeling tasks are extensive and convincing, showing consistent improvements in generalization performance across various models, datasets, and base optimizers. The surprising 2x speedup over AdamW in Llama pre-training is particularly impressive and warrants further investigation.

The theoretical analysis in Section 5, proving the substantial acceleration of IRE over SAM in minimizing the trace of Hessian, adds rigor to the empirical findings. The proofs leverage reasonable assumptions and provide non-asymptotic guarantees.

However, the paper could be strengthened by providing more insights and discussions on the potential limitations and failure cases of IRE. For example, are there scenarios where the diagonal Hessian approximation might be less effective? How sensitive is IRE to the choice of hyperparameters (e.g., $\lambda$ and $\gamma$)?

## Clarity
Overall, the paper is well-structured and clearly written. The main ideas, algorithms, and results are presented in a logical flow, making it easy for readers to follow. The use of figures (e.g., Figure 1) and illustrative examples enhances the clarity of the exposition. The mathematical notations and definitions are introduced appropriately and used consistently throughout the paper.

One area that could be improved is the description of the experimental setup and implementation details. While the paper provides references to the appendix for more details, including some key information (e.g., hyperparameter tuning ranges, model architectures) in the main text would make the experiments more self-contained and easier to interpret.

## Significance
The proposed IRE framework has the potential to make a significant impact in the field of deep learning optimization and generalization. By providing a principled and efficient way to accelerate convergence to flat minima, IRE can lead to models with better generalization performance and faster training times. The consistent improvements demonstrated across a range of vision and language tasks suggest that IRE could be widely applicable.

Moreover, the theoretical analysis of IRE's acceleration over SAM opens up new avenues for understanding and improving sharpness-aware optimization. The non-asymptotic guarantees provide a solid foundation for further theoretical investigations.

The 2x speedup achieved by IRE in Llama pre-training is particularly significant, given the computational challenges in training large language models. If these speedups can be reliably reproduced and scaled to larger models and datasets, IRE could meaningfully contribute to advancing the state-of-the-art in language model training.

To fully assess the significance of IRE, it would be valuable to see more comparisons with other state-of-the-art optimizers and regularization techniques. Additionally, evaluating the downstream performance of models trained with IRE (e.g., fine-tuning Llama on benchmarks) would provide a more comprehensive understanding of its impact.

**Weaknesses:**

### Comparison with Related Works
While the paper presents a novel approach to enhancing implicit regularization, it could benefit from a more detailed comparison with related works. The authors should discuss how IRE differs from and improves upon other techniques that aim to accelerate convergence to flat minima or reduce the computational cost of sharpness-aware minimization (e.g., Kwon et al., 2021; Liu et al., 2022). This would help to better highlight the novelty and advantages of IRE.

### Limitations and Failure Cases
The paper could be strengthened by providing a more in-depth discussion of the potential limitations and failure cases of IRE. For example:
- Are there scenarios where the diagonal Hessian approximation might be less effective or lead to suboptimal results?
- How sensitive is IRE to the choice of hyperparameters (e.g., $\lambda$ and $\gamma$)? Is there a risk of instability or divergence if these hyperparameters are not tuned properly?
- Are there any particular types of models, datasets, or tasks where IRE may not provide significant benefits or even hurt performance?

Addressing these questions would help readers better understand the scope and applicability of IRE.

### Experimental Setup and Implementation Details
The description of the experimental setup and implementation details could be improved. While the appendix contains additional information, it would be helpful to include key details in the main text, such as:
- The specific hyperparameter tuning ranges for $\lambda$ and $\gamma$ used in the experiments
- The architectures of the models used (e.g., ResNet and ViT variants)
- The data augmentation and preprocessing techniques applied

Including these details would make the experiments more self-contained and easier to interpret and reproduce.

### Comparisons with State-of-the-Art Optimizers
To fully demonstrate the significance of IRE, it would be valuable to include comparisons with a broader range of state-of-the-art optimizers and regularization techniques. While the paper shows consistent improvements over SGD, AdamW, and SAM, it would be informative to see how IRE performs compared to other recent approaches, such as:
- Adaptive gradient methods like Adagrad (Duchi et al., 2011), Adam (Kingma and Ba, 2014), and their variants
- Second-order optimization methods like K-FAC (Martens and Grosse, 2015) and Shampoo (Gupta et al., 2018)
- Other regularization techniques like weight decay, dropout, and label smoothing

These comparisons would provide a more comprehensive understanding of IRE's performance and potential advantages over existing methods.

### Downstream Performance Evaluation
While the paper demonstrates impressive speedups in Llama pre-training, it would be informative to evaluate the downstream performance of the models trained with IRE. For example, fine-tuning the pre-trained Llama models on benchmark tasks like language understanding, question answering, or text generation would provide insights into the practical impact of IRE on model quality and generalization.

However, given the computational cost and time constraints of the rebuttal period, it may not be feasible to conduct extensive downstream evaluations. In this case, the authors could discuss this limitation and propose it as a direction for future work.

### Theoretical Analysis
The theoretical analysis in Section 5 provides valuable guarantees for IRE's acceleration over SAM. However, the paper could benefit from a more intuitive explanation of the key assumptions and their implications. For example:
- Discussing the practical significance of Assumption 5.1 (manifold of minimizers) and how it relates to the empirical observations in deep learning
- Providing a high-level interpretation of the non-asymptotic bounds and their dependence on the hyperparameters (e.g., $\eta$, $\rho$, $\lambda$)

Additionally, exploring the theoretical connections between IRE and other optimization techniques (e.g., momentum, adaptive methods) could provide further insights into its behavior and potential extensions.

**Questions:**

1. **Comparison with related works:**
   - Question: How does IRE differ from and improve upon other techniques that aim to accelerate convergence to flat minima or reduce the computational cost of sharpness-aware minimization, such as those proposed by Kwon et al. (2021) and Liu et al. (2022)?
   - Suggestion: Provide a more detailed discussion comparing and contrasting IRE with these related approaches to better highlight the novelty and advantages of IRE.

2. **Limitations and failure cases:**
   - Questions: Are there scenarios where the diagonal Hessian approximation might be less effective or lead to suboptimal results? How sensitive is IRE to the choice of hyperparameters (e.g., $\lambda$ and $\gamma$)? Are there any particular types of models, datasets, or tasks where IRE may not provide significant benefits or even hurt performance?
   - Suggestion: Include a more in-depth discussion of the potential limitations and failure cases of IRE to help readers better understand its scope and applicability.

3. **Experimental setup and implementation details:**
   - Question: What are the specific hyperparameter tuning ranges for $\lambda$ and $\gamma$ used in the experiments, the architectures of the models (e.g., ResNet and ViT variants), and the data augmentation and preprocessing techniques applied?
   - Suggestion: Include these key details in the main text to make the experiments more self-contained and easier to interpret and reproduce.

4. **Comparisons with state-of-the-art optimizers:**
   - Question: How does IRE perform compared to other recent optimization approaches, such as adaptive gradient methods (e.g., Adagrad, Adam), second-order methods (e.g., K-FAC, Shampoo), and other regularization techniques (e.g., weight decay, dropout, label smoothing)?
   - Suggestion: Include comparisons with a broader range of state-of-the-art optimizers and regularization techniques to comprehensively understand IRE's performance and potential advantages.

5. **Downstream performance evaluation:**
   - Question: How does the downstream performance of models trained with IRE compare to those trained with other optimizers when fine-tuned on benchmark tasks like language understanding, question answering, or text generation?
   - Suggestion: If feasible within the rebuttal period, evaluate the downstream performance of the pre-trained Llama models to provide insights into the practical impact of IRE on model quality and generalization. If not feasible, discuss this limitation and propose it as a direction for future work.

6. **Theoretical analysis:**
   - Questions: What is the practical significance of Assumption 5.1 (manifold of minimizers) and how does it relate to the empirical observations in deep learning? Can you provide a high-level interpretation of the non-asymptotic bounds and their dependence on the hyperparameters (e.g., $\eta$, $\rho$, $\lambda$)?
   - Suggestion: Provide more intuitive explanations of the key assumptions and their implications in the theoretical analysis. Additionally, explore the theoretical connections between IRE and other optimization techniques (e.g., momentum, adaptive methods) to provide further insights into its behavior and potential extensions.

Addressing these questions and incorporating the suggestions in the rebuttal or a revised version of the paper would help to strengthen the work and provide a more comprehensive understanding of the proposed IRE framework, its novelty, effectiveness, and impact.

**Limitations:**

1. **Limitations:**
   - Add a dedicated subsection discussing the assumptions made in the theoretical analysis, potential scenarios where IRE may not provide benefits or even hurt performance, and the sensitivity of IRE to hyperparameter choices.

2. **Potential negative societal impact:**
   - Include a subsection discussing the environmental cost of training LLMs, the potential misuse of LLMs for generating harmful content, and the possible widening of the gap between well-resourced and under-resourced research groups due to IRE's computational advantages.
   - Propose mitigation strategies or areas for future research to address these concerns.

3. **Reproducibility and transparency:**
   - Provide clear instructions for reproducing the experiments, make the code and pre-trained models publicly available, and discuss any limitations or challenges in reproducibility.

---

> ### Author Rebuttal · Authors · 2024-08-06
>
> We appreciate the reviewer's recognition of our work and helpful comments. Below, we offer detailed responses to the reviewer’s questions:
>
> - **W\&Q1. Comparison with related works.**
>
>   **Response:** We thank the reviewer for this question and will provide a more detailed comparison with related algorithms in the revised version. First, we clarify that the mechanism behind IRE in accelerating sharpness reduction is fundamentally different from that of SAM. Moreover, IRE can be integrated with generic base optimizers, including SGD, Adam, and SAM. In contrast, the SAM variants, including Kwon et al. (2021) and Liu et al. (2022), are specifically designed to improve SAM.
>
>
> - **W\&Q2. Limitations and failure cases.**
>
>   **Response:** Thank the reviewer for the insightful question. We acknowledge  that diagonal Hessian may not always provide accurate second-order information for identifying the flat (zero-curvature) directions. Recall that IRE iterates as $\theta_{t+1} = \theta_t - \eta (1+\kappa P) g_t$, where $P$ represents the projection to flat directions. When $\kappa=0$, it recovers the base optimizer. For problems where  diagonal Hessian provides relatively accurate second-order information,  a large $\kappa$ can be used. Conversely, for problems where diagonal Hessian is less informative, we can set $\kappa$ to be small or even to zero. This approach allows to control over the influence of whether the diagonal Hessian approximation is effective. This idea is analogous to the damping technique adopted in Newton's methods with an approximate Hessian.   Specifically, we tune $\kappa\in$\{1,2\}  for CNNs, $\kappa\in$\{2,3,4\} for Llama pre-training, and $\kappa\in$\{20,50\} for ViTs.
>
> - **W\&Q3. Experimental setup and implementation details.**
>
>   **Response:** Thank the reviewer for this question. For image classification tasks, the models including ResNets and ViT-T/S follows the ones in Muller et al. (2023). For LLM pre-training, we use the Llama models from HuggingFace. For image data pre-processing, we apply the default normalization technique: *transforms.Normalize(mean, std)*. The search ranges of hyperparameters, as well as the data augmentation methods are specified in Appendix B. In the revised version, we will carefully incorporate all these experimental details in the main text.
>
>
> - **W\&Q4. Comparisons with state-of-the-art optimizers.**
>
>   **Response:** Thank the reviewer for this question. We clarify that IRE is a generic framework for boosting optimizer's implicit bias and can be integrated with general optimizers. Due to space and resource limit, we have only compared IRE with the popular algorithm including SGD, SAM and AdamW. Specifically, we show that IRE can improve the generalization in image classification task and convergence of AdamW in LLM pre-training. As for other optimizers mentioned by the reviewer, Adagrad has been replaced by Adam(W) in most fields; second-order algorithms like K-FAC and Shampoo is computational expensive and it is less  popular in training large models.
>
>   About regularization, IRE is proposed to enhance implicit regularization and thus, orthogonal to those explicit regularization, including weight decay, dropout and label smoothing. It is worth mentioning that our experiments incorporate various regularization techniques such as weight decay and label smoothing. Please refer to the Appendix for more details.
>
> - **W\&Q5. Downstream performance evaluation.**
>
>   **Response:** Thank the reviewer for this constructive comment. It is valuable to explore the performance of the pretrained models in downstream tasks. However, due to the limited time for response, we could not fully evaluate the downstream tasks for pretrained models, but we conducted preliminary experiments to explore this issue.
>   - Notably, "Liu et al. (2023) Same pretraining loss..." points out a strong correlation between the flatness and downstream task performance among models. Specifically, it implies that *for models with the same pre-training loss, flatter solutions yield better performance on downstream tasks.*
>   - Inspired by this observation, we evaluated the flatness, measured by the trace of Hessian, of models pre-trained by AdamW and AdmIRE. Due to time constraints, we only focus on the experiments in Fig.3 (left), i.e., training Llama (60M) on wiki-103 dataset using either AdamW or AdmIRE. The experimental results, shown in the **attached PDF** in our **Response to All Reviewers**, demonstrate that AdmIRE not only achieves the same loss in *only half the iterations* required by AdamW, but also the solutions found by AdmIRE are *significantly flatter* than that found by AdamW.
>
>   In the revised version, we will include a comprehensive evaluation of downstream tasks.
>
>
> - **W\&Q6. Theoretical analysis.**
>
>   **Response:** Thanks the reviewer for the question.
>   - We clarify that Assumption 5.1 essentially only assume: 1) the loss $\mathcal{L}(\cdot)$ is  $C^4$ smooth and 2) the global minima are connected. For over-parametrized models, this connectivity assumption on the minima manifold has been empirically verified in works such as Draxler et al. (2018) and Garipov et al. (2018), and theoretically supported in Cooper (2018). We refer to lines 267-270 of our manuscript for more details.
>   - Our main theoretical results are explained in lines 286-369, where we provide an intuitive interpretation: IRE accelerates the "effective dynamics" of base optimizers in flat directions, achieving faster reduction of sharpness; moreover, IRE maintains stability by not affecting movement along sharp directions. This dual effect ensures IRE can improve generalization performance without compromising training stability.
>   - As for other optimization techniques, our experiments have incorporated momentum, weight decay, and the adaptive method (AdamW). However, these techniques poses new challenges to theoretical analysis, which we leave as future work.

---

> > ### Author Response · Authors · 2024-08-12
> >
> > Thanks again for your valuable time and effort in reviewing our work!
> >
> > We are wondering if our responses and new experiments address your questions or concerns.
> >
> > We are happy to try to address any other comments in the time remaining.

---

> > > ### Comment · Reviewer_QWSd · 2024-08-13
> > > **Thanks for the response**
> > >
> > > Thanks for responding to my questions; they generally address my concerns. I don't have any further questions, and I still support the work's acceptance; I'm keeping my original positive score to reflect my assessment.

---

### Official Review · Reviewer_69nS · 2024-07-14

**Soundness:** 3
**Presentation:** 3
**Contribution:** 2
**Rating:** 6
**Confidence:** 4

**Summary:**

The authors propose IRE to enhance the implicit regularization of base optimizers, thereby improving the generalization and convergence in deep learning. IRE decouples the dynamics of flat and sharp directions, reducing sharpness along flat directions while maintaining stability in sharp directions. The paper provides theoretical evidence that IRE can substantially expedite convergence towards flat minima in SAM.

**Strengths:**

1. IRE's ability to integrate with existing optimizers without major modifications makes it easily adoptable in current systems.
Performance Improvement: Empirical results show that IRE enhances the generalization capabilities of popular optimizers across multiple tasks and datasets.
2. In the pre-training of large language models, IRE has demonstrated a significant acceleration in convergence speed.
3. The paper offers a theoretical foundation for IRE's effectiveness in minimizing the trace of the Hessian, reinforcing its practical applications.

**Weaknesses:**

1. The improvements of IRE in Table 1 and 7 are not as significant as expected. It makes one wonder about its usefulness for CNN networks and its oversensitivity to hyperparameters
2. A big concern for me is that the experiment in Figure 3 does not appear to have converged yet, the rate of convergence in the early part of an experiment does not equate to the rate of convergence throughout the training process, as well as a more detailed analysis of the performance of the model after convergence should have been added to demonstrate that the final position of convergence is good.
3. Judging from the code, the cost of each training step is at least twice that of SGD.

**Questions:**

1. Could IRE help in the finetuning phase of LLMs, Including both convergence speed and convergence position properties?

**Limitations:**

1. The author did not provide information on the computing resources they used.

---

> ### Author Rebuttal · Authors · 2024-08-06
>
> We appreciate the reviewer’s recognition of our work and helpful comments. Below, we offer detailed responses to the reviewers questions.
>
> - **W1.** The improvements of IRE in Table 1 and 7 are not as significant as expected. It makes one wonder about its usefulness for CNN networks and its oversensitivity to hyperparameters.
>
>   **Response:** We thank the reviewer for this insightful comment.
>   - **CNNs vs ViTs.**  It is important to emphasize that **CNNs are designed  with a strong image prior**, which limits the potential for further improvement through regularization. Consequently, it is not unexpected that the improvement achieved by IRE for CNNs is less significant. This aligns  with  observations in previous regularization techniques, such as ASAM [1] and SAM-ON [2], which similarly shows limited improvements for CNNs. For instance, ASAM improves SAM by only 0.26\% points when training WRN-28-10 on CIFAR-100, and SAM-ON improves SAM by just 0.17\% points when training ResNet-50 on ImageNet. In contrast, ViTs, which have a much weaker image prior, benefit more from regularization, as shown in Table 3.
>   - **Sensitivity to hyperparameters.** 1) For image classification tasks, the optimal  hyperparameters for IRE do vary depending the dataset and model. Nevertheless, Table 7 demonstrates that  IRE **consistently** improves performance even without tuning the hyper-parameters, although the improvement is modest. 2) In contrast, when training Llama models, IRE with a fixed hyperparameter $\gamma=0.6$ performs effectively across varying model and data sizes, as shown in Fig.3.
>
> - **W2.** A big concern for me is that the experiment in Figure 3 does not appear to have converged yet, the rate of convergence in the early part of an experiment does not equate to the rate of convergence throughout the training process, as well as a more detailed analysis of the performance of the model after convergence should have been added to demonstrate that the final position of convergence is good.
>
>   **Response:** We thank the reviewer for raising this concern.
>   - **Loss convergence in LLM pre-training.** It is important to note that in LLM pre-training, the training loss never decrease significantly due to the huge size of  dataset. For example, as reported in [3], the official Llama pre-training achieves only  a final loss around 1.55, despite the model size being 65B.
>   - **Final loss v.s. performance.** In LLM pre-training, it is widely observed that  the performance on downstream tasks is predominantly determined by the final pre-training loss and has little correlation with other pre-training components such as model types. We refer to [4] for a detailed analysis of this issue. Therefore,  the primary focus in LLM pre-training is to reduce the final loss  as much as possible, given specific computational and data resources.
>   - **Final loss.** Our final loss aligns with  expectations based on our model scale. For instance, for our baseline, our LLama (229M) was trained on openwebtext by AdamW for 100k steps, resulting in a final loss of 2.835. Fig.1(d) in [5] show that GPT models of varying sizes were trained on openwebtext by AdamW for 100k steps, achieving losses of 2.915 for GPT-small (125M) and 2.695 for GPT-middle (355M). Our loss of 2.835  for Llama (229M) (the baseline) is consistent with these results. Notably, our proposed AdmIRE achieves the same loss of 2.835 with only 50k iterations, which is half the number of iterations  required by AdamW.
>
> - **W3.** Judging from the code, the cost of each training step is at least twice that of SGD.
>
>   **Response:** We  clarify that the cost of our IRE takes twice that of base optimizers only during the steps of estimating the diagonal Hessian and updating the mask. However, these steps are triggered only once every $K$ steps (see Alg. 1). In all our experiments, we set $K=10$ (see l.168-l.172), so the average cost of IRE is only **1.1 times** that of the base optimizer. Table 4 further verify this estimate empirically.
>
> - **Q1.** Could IRE help in the finetuning phase of LLMs, Including both convergence speed and convergence position properties?
>
>   **Response:** We thank the reviewer for this constructive question. To address this, we have conducted a new Supervised Fine-Tuning (SFT) experiment. Specifically, we finetune pretrained Llama2-7B with LoRA on Stanford's Alpaca dataset. The results, shown in the **attached PDF** in our **Global Response to All Reviewers**, demonstrate that IRE can **accelerate the convergence in SFT**. This is consistent with the advantage of IRE in pretraining. Unfortunately, we do not have enough time to evaluate the performance of the convergent models on downstream tasks. In the revised version, we will include complete results for SFT.
>
>
> - **L1.** The author did not provide information on the computing resources they used.
>
>   **Response:** Thank you for this reminder. For Section 4.1 (image classification), the experiments on Cifar-10/100 were conducted using a single A800 GPU, while the experiments on ImageNet were conducted using 4 A800 GPUs. Details regarding the computing resources for the experiments in Section 4.2 are provided in Appendix B (l.664 and l.678).
>
>
> [1] Kwon et al. ASAM: Adaptive Sharpness-Aware Minimization for Scale-Invariant Learning of Deep Neural Networks. (ICML 2021)
>
> [2] Muller et al. Normalization Layers Are All That Sharpness-Aware Minimization Needs. (NeurIPS 2023)
>
> [3] Touvron et al., LLaMA: Open and Efficient Foundation Language Models. (2023)
>
> [4] Du et al. Understanding Emergent Abilities of Language Models from the Loss Perspective. (2024)
>
> [5] Liu et al. Sophia: A Scalable Stochastic Second-order Optimizer for Language Model Pre-training. (ICLR 2024)

---

> > ### Author Response · Authors · 2024-08-12
> >
> > Thanks again for your valuable time and effort in reviewing our work!
> >
> > We are wondering if our responses and new experiments address your questions or concerns.
> >
> > We are happy to try to address any other comments in the time remaining.

---

### Author Rebuttal · Authors · 2024-08-06

### **Global Response to All Reviewers.**

- We express our sincere gratitude to all reviewers for appreciating our results, i.e,
  - A novel algorithm framework (IRE), which can improve both generalization and optimization, by enhancing the implicit regularization.
  - Experimentally, IRE consistently improves generalization performance for image classification. Remarkably, IRE achieves a $2\times$ speed-up in pre-training Llama models.
  - Theoretically, we demonstrate that IRE substantially accelerate convergence towards flat minima in sharpness-aware minimization (SAM).

- We also thank appreciate the reviewers for their valuable comments and suggestions for improving our paper. In our revised version, we will correct all typos, provide complete experimental settings and results, and incorporate the discussions with the reviewers.

- **New experiments.** To further explore the performance of IRE in various settings, we have conducted 3 new experiments. As suggested by Reviewer 69nS and QWSd, we examined:
  -  (i) Whether IRE can accelerate the convergence of AdamW in the finetuning phase;
  - (ii) During the pre-training phase, whether the solution found by AdmIRE has better properties than that found by AdamW.

  Additionally, to supplement our results on generalization, we  conducted an additional experiment:
  - (iii) Whether IRE can improve the generalization performance of AdamW when training ViT on ImageNet.

  All of these results are reported in the **attached PDF**, which further verifies remarkable performance of IRE under more settings.

- We have addressed each concern raised by the reviewers through separate responses provided below.

---

### Decision · Program_Chairs · 2024-09-25

**Decision:**

Accept (poster)

**Comment:**

This paper proposes a method for decoupling the optimization dynamics arising from sharp and flat directions, amplifying the dynamics along flat directions. This allows the optimization algorithm to navigate along flat directions faster without jeopardizing the optimization's stability due to sharp directions. The resulting algorithm is shown to improve both generalization performance and convergence speed, supported by empirical and theoretical evidence.

The paper received a detailed review, in which reviewers found the paper well-written and the proposed idea original with the potential for a significant impact. The reviewers also raised questions and concerns about the computational cost of the algorithm's steps, related works, empirical assessment, and the relevance of the stylized example in section 2 to deep learning. The authors' response to this feedback was clear and complete enough that reviewers did not need to ask follow-up questions. All reviewers rated the paper as accept. In agreement with them, I believe this paper has an interesting idea with potential impact, and I recommend acceptance.